# Spectro-Riemannian Graph Neural Networks

**Karish Grover**[1,†]**, Haiyang Yu**[2]**, Xiang Song**[3]**, Qi Zhu**[3]**, Han Xie**[3]**,
Vassilis N. Ioannidis**[3]**, Christos Faloutsos**[1]
[1]Carnegie Mellon University, [2]Texas A&M University, [3]Amazon
{karishg,christos}@cs.cmu.edu, {haiyang}@tamu.edu,
{xiangsx,qzhuamzn,hanxie,ivasilei}@amazon.com

## Abstract

Can integrating *spectral* and *curvature* signals unlock new potential in graph representation learning? Non-Euclidean geometries, particularly Riemannian manifolds such as hyperbolic (negative curvature) and spherical (positive curvature), offer powerful inductive biases for embedding complex graph structures like scale-free, hierarchical, and cyclic patterns. Meanwhile, spectral filtering excels at processing signal variations across graphs, making it effective in homophilic and heterophilic settings. Leveraging both can significantly enhance the learned representations. To this end, we propose *Spectro-Riemannian Graph Neural Networks* (CUSP) – the *first* graph representation learning paradigm that unifies both CUrvature (geometric) and SPectral insights. CUSP is a mixed-curvature spectral GNN that learns spectral filters to optimize node embeddings in *products* of constant-curvature manifolds (hyperbolic, spherical, and Euclidean). Specifically, CUSP introduces three novel components: (a) *Cusp Laplacian*, an extension of the traditional graph Laplacian based on Ollivier-Ricci curvature, designed to capture the curvature signals better; (b) *Cusp Filtering*, which employs multiple Riemannian graph filters to obtain cues from various bands in the eigenspectrum; and (c) *Cusp Pooling*, a hierarchical attention mechanism combined with a curvature-based positional encoding to assess the relative importance of differently *curved* substructures in our graph. Empirical evaluation across eight homophilic and heterophilic datasets demonstrates the superiority of CUSP in node classification and link prediction tasks, with a gain of up to $5.3\%$ over state-of-the-art models. The code is available at: https://github.com/amazon-science/cusp.

## 1 Introduction

Graph representation learning has garnered significant research interest in recent years, owing to its fundamental relevance in domains such as natural language processing (Mihalcea & Radev, 2011), biology (Zhang et al., 2021a), and social network analysis (Grover et al., 2022). Recent advances have rigorously examined constant *curvature* spaces to learn distortion-free graph representations, as they provide suitable inductive biases for particular structures while avoiding the intrinsic problems of very high dimensionality. For example, hyperbolic space (negative curvature) is optimal for hierarchical tree-like graphs (Chami et al., 2019), while spherical geometry (positive curvature) is best suited for cyclic graphs (Gu et al., 2019).The idea behind all Riemannian GNNs is that optimal embeddings are achieved when the underlying manifold's curvature aligns with the graphs' discrete curvature. To model real-world graphs with complex topologies that cannot be adequately represented within a single constant-curvature manifold, various mixed-curvature GNNs have been proposed, that operate across multiple manifolds (Zhu et al., 2020).

Despite their success in managing complex graph topologies, existing mixed-curvature GNNs still have significant limitations (Figure 1 (b)). **(a)** L1 (*Low-pass filtering bias*): State-of-the-art mixed-curvature Riemannian GNNs, such as $\kappa$GCN (Bachmann et al., 2020) and $\mathcal{Q}$GCN (Xiong et al., 2022), inherently mimic low-pass spectral filters. These models are adaptations of the Euclidean GCN (Kipf & Welling, 2016) to Riemannian manifolds, which are known to operate under the homophily assumption (low-pass filters) predominantly. Consequently, their performance degrades on

---

†The work was done during Karish Grover's internship at Amazon, US.

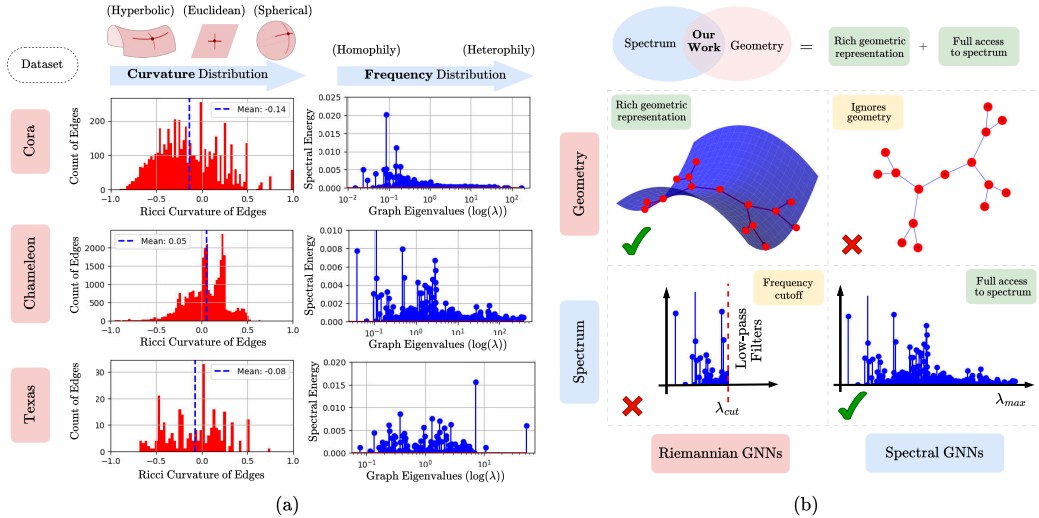

Figure 1: **Motivation behind CUSP**. **(a)** Diversity of curvatures ($-$ve to $+$ve) and frequencies in homophilic (Cora) and heterophilic (Texas, Chameleon) real-world graphs (Section 5). **(b)** Spectral GNNs overlook the curvature, whereas Riemannian GNNs are restricted by a cut-off frequency. CUSP aims for the best of both – a rich geometric representation + full access to the spectrum.

graphs with varying degrees of heterophily. **(b)** L2 (*Task-specific relevance of curvature*): Different datasets and tasks may emphasize more on different curvatures within a graph (Figure 1 (a)). For example, the task of community detection focuses on clustered nodes typically associated with a positive curvature (Tian et al., 2023). In contrast, fake news detection on social media graphs prioritizes tree-like cascade structures characterized by a negative curvature (Grover et al., 2022). Current models do not adapt curvatures to tasks. **(c)** L3 (*Lack of geometry-equivariance in spectral GNNs*): Spectral GNNs like BernNet (He et al., 2021) and GPRGNN (Chien et al., 2020) incorporate flexible graph filters to learn representations based on different parts of the eigenspectrum but assume a flat Euclidean manifold, ignoring the underlying geometry. As the geometry of the graph changes, these filters should adapt to reflect the changing geometric properties, which is currently not the case.

To bridge these gaps, we introduce CUSP, a mixed-curvature spectral GNN that operates on a product manifold composed of multiple constant-curvature spaces and computes *geometrically* and *spectrally* parameterized graph filters. We begin by introducing the **(a)** *Cusp Laplacian*, a curvature-aware Laplacian operator for graphs, inspired by the equation of heat flow (Thanou et al., 2017) and the discrete Ollivier-Ricci curvature (Ollivier, 2007). At the core of our approach is **(b)** *Cusp Filtering*, where we propose a filter bank consisting of multiple mixed-curvature graph filters to ensure that our GNN captures information from multiple bands in the eigenspectrum. Using the Cusp Laplacian, we extend the generalized PageRank (GPR)-based spectral GNN (Chien et al., 2020) to mixed-curvature spaces by incorporating operations based on the $\kappa$-stereographic model of Riemannian geometry (Bachmann et al., 2020). We chose GPRGNN as the spectral backbone of CUSP because of its ability to capture node features and topological graph signals simultaneously. Lastly, we introduce the **(c)** *Cusp Pooling* mechanism, complemented by functional curvature embeddings based on Bochner's Theorem (Xu et al., 2020), to weigh differently *curved* substructures, enhancing its ability to model real-world graphs with diverse geometric and spectral properties. We perform an extensive empirical evaluation of CUSP for Node Classification (NC) and Link Prediction (LP) tasks, on eight real-world datasets. CUSP records state-of-the-art performance, with a gain of up to 5.3%. We summarize our main contributions as follows:

- To the best of our knowledge, this is the **first attempt** towards a graph learning paradigm that seamlessly integrates both *geometry* and *spectral* cues.

- We introduce a curvature-aware *Cusp Laplacian* operator, design a mixed-curvature spectral graph filtering framework, *Cusp Filtering*, and propose a curvature embedding method using classical harmonic analysis and a hierarchical attention mechanism called *Cusp Pooling*.

- We conduct extensive experimentation on eight real-world benchmarking datasets, featuring homophilic and heterophilic graphs, for node classification and link prediction tasks.

## 2  RELATED WORKS

**Riemannian Geometry in Graph Neural Networks**. Graph Neural Networks (GNNs) have set new benchmarks for tasks like node classification and link prediction. Recently, non-Euclidean (Riemannian) spaces – particularly hyperbolic (Sala et al., 2018) and spherical (Liu et al., 2017; Wilson et al., 2014) geometries – have garnered attention for their ability to produce less distorted representations, aligning well with hierarchical and cyclic data structures, respectively. Several approaches have emerged in this context. (a) *Single Manifold GNNs*: GNNs such as HGAT (Zhang et al., 2021b), HGCN (Chami et al., 2019), and HVAE (Sun et al., 2021) have demonstrated state-of-the-art performance on tree-like or hierarchical graphs by learning representations in hyperbolic space. (b) *Mixed-Curvature GNNs*: To model more complex topologies (for example, a tree branching from a cyclic graph), mixed-curvature GNNs have been proposed. Gu et al. (2019) pioneered this direction by embedding graphs in a product manifold combining spherical, hyperbolic, and Euclidean spaces. Building on this, models like $\kappa$-GCN (Bachmann et al., 2020) and Q-GCN (Xiong et al., 2022) extended the GCN architecture (Kipf & Welling, 2016) to constant-curvature spaces using the $\kappa$-stereographic model and pseudo-Riemannian manifolds, respectively. More recently, Sun et al. (2022) proposed a mixed-curvature GNN for self-supervised learning, while FPS-T (Cho et al., 2023) generalized the Graph Transformer (Min et al., 2022) to operate across multiple manifolds.

**Spectral Graph Neural Networks**. Spectral GNNs employ spectral graph filters (Liao et al., 2024) to process graph data. These models either use fixed filters, as seen in APPNP (Gasteiger et al., 2018) and GNN-LF/HF (Zhu et al., 2021), or learnable filters, as demonstrated by ChebyNet (Defferrard et al., 2016) and GPRGNN (Chien et al., 2020), which approximate polynomial filters using Chebyshev polynomials and generalized PageRank, respectively. BernNet (He et al., 2021) expresses filtering operations through Bernstein polynomials. However, many of these methods focus primarily on low-frequency components of the eigenspectrum, potentially overlooking important information from other frequency bands – particularly in heterophilic graphs. Models like GPRGNN and BernNet address this by exploring the entire spectrum, performing well across both homophilic and heterophilic graphs. GPRGNN stands out among them because it can express several polynomial filters and incorporate node features and topological information. Despite these advances, current mixed-curvature and spectral GNNs face significant limitations (L1, L2, L3) that constrain their performance. To the best of our knowledge, this work is the first to unify *geometric* and *spectral* information within a single model. Before presenting the architecture of CUSP, we introduce some key preliminary concepts in the following section.

## 3  PRELIMINARIES

We study graphs $\mathcal{G} = (\mathcal{V}, \mathcal{E}, \mathbf{A})$, where $\mathcal{V}$ is a finite set of $|\mathcal{V}| = n$ vertices, $\mathcal{E}$ is a set of edges and $\mathbf{A} \in \mathbb{R}^{n \times n}$ is a weighted graph adjacency matrix. The nodes are associated with the node feature matrix $\mathbf{F} \in \mathbb{R}^{n \times d_f}$ ($d_f$ is the feature node dimension). A graph signal $\mathbf{f} : \mathcal{V} \to \mathbb{R}$ on the nodes of the graph may be regarded as a vector $\mathbf{f} \in \mathbb{R}^n$ where $f_i$ is the value of $\mathbf{f}$ at the $i^{th}$ node. An essential operator in spectral graph analysis is the graph Laplacian $\mathbf{L} = \mathbf{D} - \mathbf{A} \in \mathbb{R}^{n \times n}$ where $\mathbf{D} \in \mathbb{R}^{n \times n}$ is the diagonal degree matrix with $\mathbf{D}_{ii} = \sum_j \mathbf{A}_{ij}$ (Kipf & Welling, 2016). The normalized Laplacian is defined as $\mathbf{L}_n = \mathbf{I} - \mathbf{A}_n$ where $\mathbf{A}_n = \mathbf{D}^{-1/2}\mathbf{A}\mathbf{D}^{-1/2}$ is the normalized adjacency matrix, and $\mathbf{I} \in \mathbb{R}^{n \times n}$ is the identity matrix. As $\mathbf{L}$ is a real symmetric positive semidefinite matrix, it has a complete set of orthonormal eigenvectors $\mathbf{U} = \left[ \{\mathbf{u}_l\}_{l=0}^{n-1} \right] \in \mathbb{R}^{n \times n}$, and their associated ordered real nonnegative eigenvalues $\left[ \{\lambda_l\}_{l=0}^{n-1} \right] \in \mathbb{R}^n$, identified as the *frequencies* of the graph. The Laplacian can be diagonalized as $\mathbf{L} = \mathbf{U}\mathbf{\Lambda}\mathbf{U}^\top$ where $\mathbf{\Lambda} = \mathbf{diag}(\left[ \{\lambda_l\}_{l=0}^{n-1} \right]) \in \mathbb{R}^{n \times n}$.

**Riemannian Geometry** (Do Carmo & Flaherty Francis, 1992). A smooth *manifold* $\mathcal{M}$ generalizes the notion of *surface* to higher dimensions. Each point $\mathbf{x} \in \mathcal{M}$ is associated with a *tangent space* $\mathcal{T}_\mathbf{x}\mathcal{M}$, which is locally Euclidean. On tangent space $\mathcal{T}_\mathbf{x}\mathcal{M}$, the *Riemannian metric*, $g_\mathbf{x}(\cdot, \cdot) : \mathcal{T}_\mathbf{x}\mathcal{M} \times \mathcal{T}_\mathbf{x}\mathcal{M} \to \mathbb{R}$, defines an inner product so that geometric notions (like distance, angle, etc.) can be induced. The pair $(\mathcal{M}, g)$ is called a *Riemannian manifold*. For $\mathbf{x} \in \mathcal{M}$, the *exponential map* at $\mathbf{x}$, $\exp_\mathbf{x}(\mathbf{v}) : \mathcal{T}_\mathbf{x}\mathcal{M} \to \mathcal{M}$, projects the vector $\mathbf{v} \in \mathcal{T}_\mathbf{x}\mathcal{M}$ onto the manifold $\mathcal{M}$, and the *logarithmic map*, $\log_\mathbf{x}(\mathbf{y}) : \mathcal{M} \to \mathcal{T}_\mathbf{x}\mathcal{M}$, projects the vector $\mathbf{y} \in \mathcal{M}$ back to the tangent space $\mathcal{T}_\mathbf{x}\mathcal{M}$. The Riemannian metric defines a *curvature* ($\kappa$) at each point on the manifold, indicating how the space is curved. There are three canonical types: positively curved *Spherical* ($\mathbb{S}$) space ($\kappa > 0$), negatively curved *Hyperbolic* ($\mathbb{H}$) space ($\kappa < 0$), and flat *Euclidean* ($\mathbb{E}$) space ($\kappa = 0$).

**Product Manifolds** (Gu et al., 2019). Consider $q$ constant-curvature manifolds $\{\mathcal{M}_i^{\kappa_i, d_i}\}_{i=1}^q$ with dimension $d_i$ and curvature $\kappa_i$. Then, the product manifold is defined as the Cartesian product $\mathbb{P} = \mathcal{M}_1^{\kappa_1, d_1} \times \mathcal{M}_2^{\kappa_2, d_2} \cdots \times \mathcal{M}_q^{\kappa_q, d_q}$, with total dimension $\sum_{i=1}^q d_i$. Each $\mathcal{M}_i \in \{\mathbb{H}, \mathbb{S}, \mathbb{E}\}$ is known as a *component* space, and the decomposition $\mathbb{P} = \times_{i=1}^q \mathcal{M}_i^{\kappa_i, d_i}$ is called the *signature* of $\mathbb{P}$.

$\kappa-$**Stereographic Model** (Bachmann et al., 2020). In this work, we adopt the $\kappa$-**Stereographic Model** to define Riemannian algebraic operations across both positively and negatively curved spaces within a unified framework. This model eliminates the need for separate mathematical formulations for different geometries. In particular, $\mathcal{M}_\kappa^d$ is the stereographic sphere model for spherical manifold ($\kappa > 0$), while it is the Poincaré ball model (Ungar, 2001) for hyperbolic manifold ($\kappa < 0$). More mathematical details and intuitions have been discussed in Appendix 7.2.4.

**Ollivier-Ricci Curvature** (ORC). Discrete data like graphs lack manifold structure (hence, no *curvature*). Several discrete analogs of manifold curvature have been defined, which satisfy properties similar to *curvature*. ORC (Ollivier, 2007) is a graph discrete analog of *Ricci curvature* (Tanno, 1988) and is defined by transport along an edge of the network, between neighborhoods of the vertex. In an unweighted graph, for a hyperparameter $\delta \in [0, 1]$, we endow each node's ($x$) neighbourhood ($\mathcal{N}(x)$) with a probability measure, $m_x^\delta(z) := \frac{1-\delta}{|\mathcal{N}(x)|} \; \forall z \in \mathcal{N}(x)$, and $m_x^\delta(z) = \delta$ when $z = x$, and analogously for $m_y^\delta(z)$. ORC for an edge $(x, y)$ is then defined w.r.t. the Wasserstein-1 distance, $\mathbf{W}_1$ (Piccoli & Rossi, 2016), between these measures, i.e., $\widetilde{\kappa}(x, y) := 1 - \frac{\mathbf{W}_1(m_x^\delta, m_y^\delta)}{d_\mathcal{G}(x, y)}$. $d_\mathcal{G}(x, y)$ is the shortest graph distance between nodes $x$ and $y$. The Ollivier-Ricci curvature $\widetilde{\kappa}(x)$ for a node $x$ is defined as the average curvature of its adjacent edges i.e. $\widetilde{\kappa}(x) = \frac{1}{|\mathcal{N}(x)|} \sum_{z \in \mathcal{N}(x)} \widetilde{\kappa}(x, z)$.

**Generalized PageRanks** (GPR). Generalized PageRank methods originated in the context of unsupervised graph clustering, where they demonstrated notable improvements over the classical Personalized PageRank (Kloumann et al., 2017; Li et al., 2019). The core idea behind GPRs is as follows: starting with a seed node $s \in \mathcal{V}$ (vertex set) within a graph cluster, an initial feature vector $\mathbf{H}^{(0)} \in \mathbb{R}^{n \times 1}$ is set, where $\mathbf{H}_v^{(0)} = \delta_{vs}$ (i.e., 1 for the seed node and 0 for all others). The GPR score is then defined as $\sum_{k=0}^\infty \gamma_k \widetilde{\mathbf{A}}_n^k \mathbf{H}^{(0)} = \sum_{k=0}^\infty \gamma_k \mathbf{H}^{(k)}$, where $\gamma_k \in \mathbb{R}$ are the GPR weights that control the importance of higher-order neighbors. This iterative process propagates the feature information throughout the graph. Clustering is performed by locally thresholding the GPR scores. Refer to the Appendix for more details on ORC (7.2.2), product manifolds (7.2.3), and GPR (7.4).

## 4 PROPOSED METHOD: CUSP

In this section, we present a comprehensive overview of the architecture of CUSP, as shown in Figure 2. We start by introducing and deriving the *Cusp Laplacian* (Section 4.1). Building on this, we propose *Cusp Filtering*, the core component of our approach, which is a GPR-based, mixed-curvature spectral graph filtering network (Section 4.2). Next, we introduce a curvature embedding technique grounded in classical harmonic analysis (Section 4.3), which acts as the positional encoding in the hierarchical attention-based *Cusp Pooling* mechanism (Section 4.4). Throughout this paper, we use $\kappa$ to denote the continuous manifold curvature and $\widetilde{\kappa}$ for the Ollivier-Ricci curvature.

### 4.1 CUSP LAPLACIAN

To effectively incorporate geometric insights into spectral graph learning, we introduce the *Cusp Laplacian*, a curvature-aware Laplacian operator. We begin by examining the concept of heat flow on a graph (Weber, 2008). Suppose $\psi$ describes a temperature distribution across a graph, where $\psi(x)$ is the temperature at vertex $x$. According to Newton's law of cooling (He, 2024), the heat transferred from node $x$ to node $y$ is proportional to $\psi(x) - \psi(y)$ if nodes $x$ and $y$ are connected (if they are not

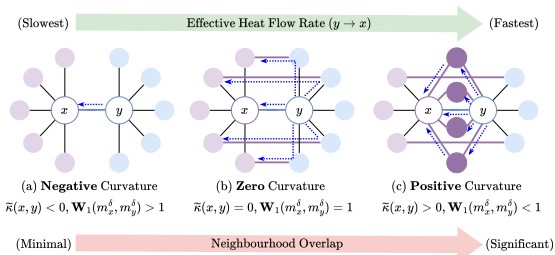

(a) **Negative** Curvature
$\widetilde{\kappa}(x, y) < 0, \mathbf{W}_1(m_x^\delta, m_y^\delta) > 1$

(b) **Zero** Curvature
$\widetilde{\kappa}(x, y) = 0, \mathbf{W}_1(m_x^\delta, m_y^\delta) = 1$

(c) **Positive** Curvature
$\widetilde{\kappa}(x, y) > 0, \mathbf{W}_1(m_x^\delta, m_y^\delta) < 1$

Figure 2: Consider heat diffusion from $y \to x$. If $\widetilde{\kappa}(x, y) < 0$, there is a single path for the heat to diffuse from $y \to x$. When $\widetilde{\kappa}(x, y) \geq 0$, heat can effectively diffuse through other paths from $y \to x$ (*dotted*).

connected, no heat is transferred). Consequently, the heat diffusion equation on the graph can be expressed as $\frac{d\psi}{dt} = -\beta \sum_y \mathbf{A}_{xy}(\psi(x) - \psi(y))$, where $\beta$ is a constant of proportionality and $\mathbf{A}$ denotes the adjacency matrix of the graph. Further insight can be gained by considering Fourier's law of thermal conductance (Liu, 1990), which states that heat flow is inversely proportional to the resistance to heat transfer. In this context, we leverage the Ollivier-Ricci curvature (`ORC`) to define the notion of resistance between nodes. Implicitly, `ORC` measures the transportation cost ($\mathbf{W}_1(:,:)$) between the neighborhoods of two nodes, reflecting the effort required to transport mass between these neighborhoods (Bauer et al., 2011). We interpret this transportation cost as the *resistance* between nodes.

The vital takeaway here is that $-$ *Heat flow between two nodes in a graph is influenced by the underlying Ollivier-Ricci curvature (`ORC`) distribution.* The diffusion rate is faster on an edge with positive curvature (low resistance), and slower on an edge with negative curvature (high resistance). Intuitively, if the neighborhoods of two nodes overlap significantly (Figure 2(c)), the transportation cost between them is low ($\mathcal{R}_{xy}^{res} = \frac{\mathbf{W}_1(m_x^\delta, m_y^\delta)}{d_{\mathcal{G}}(x,y)} < 1$), resulting in a positive curvature value for the edge connecting these nodes. In this scenario, messages (analogously, *heat*) can be transmitted efficiently between the neighborhoods. Conversely, if the neighborhoods have little overlap (Figure 2(a)), the transportation cost is high, leading to a negative curvature value ($\mathcal{R}_{xy}^{res} > 1$), and the edge acts as a bottleneck, impeding effective message-passing. As a result, the heat diffusion process is influenced by the underlying curvature (resistance).

**Definition 1.** *The Cusp Laplacian operator takes the form* $-$

$$\widetilde{\mathbf{L}}\psi(x) = \sum_{y \sim x} \bar{w}_{xy} \left(\psi(x) - \psi(y)\right) = \sum_{y \sim x} e^{\frac{-1}{1-\widetilde{\kappa}(x,y)}} \left(\psi(x) - \psi(y)\right), \tag{1}$$

*where $x \sim y$ denotes adjacency between nodes $x$ and $y$, $\bar{w}_{xy} = e^{\frac{-1}{1-\widetilde{\kappa}(x,y)}}$ represents the curvature-based weight between nodes $x$ and $y$, and $\widetilde{\kappa}(x,y)$ is the discrete Ollivier-Ricci curvature between $x$ and $y$. The function $\psi : V \to \mathbb{R}$ is defined on the vertex set $V$ of the graph. In the matrix form, we can write, $\widetilde{\mathbf{L}} = \widetilde{\mathbf{D}} - \widetilde{\mathbf{A}}$, where $\widetilde{\mathbf{D}}$ and $\widetilde{\mathbf{A}}$ are the degree and adjacency matrices based on $\bar{w}_{xy}$.*

Refer to Appendix 7.3 for the derivation of Definition 1. The curvature-based Laplacian operator allow us to incorporate geometric cues into the spectral perspective, which are otherwise not captured by the traditional graph Laplacian.

## 4.2 CUSP FILTERING

Now, we will talk about how we create a filter bank (i.e. multiple graph filters) to fuse information from different parts of the eigenspectrum and learn node representations on a product manifold $\mathbb{P}$.

**Product manifold construction**. $\mathbb{P}$ can have multiple hyperbolic or spherical components with distinct *learnable* curvatures (parameters). This enables us to be representative of a wider range of curvatures. However, we only need one Euclidean space, since the Cartesian product of Euclidean space is $\mathbb{E}^{d_{(e)}} = \times_{i=1}^{j} \mathbb{E}_i^{d_{(i)}}$ such that $\sum_{i=1}^{j} d_{(i)} = d_{(e)}$. This is not the case with $\mathbb{H}$ or $\mathbb{S}$ (Eg. *Torus* i.e. $\mathbb{S}^1 \times \mathbb{S}^1$ is topologically distinct from *Sphere* i.e. $\mathbb{S}^2$). Thus, we can represent the product manifold *signature*, $\mathbb{P}^{d_{\mathcal{M}}} = \times_{q=1}^{\mathcal{Q}} \mathcal{M}_q^{\kappa_{(q)},d_{(q)}} = (\times_{h=1}^{\mathcal{H}} \mathbb{H}_h^{\kappa_{(h)},d_{(h)}}) \times (\times_{s=1}^{\mathcal{S}} \mathbb{S}_s^{\kappa_{(s)},d_{(s)}}) \times \mathbb{E}^{d_{(e)}}$ with total dimension $d_{\mathcal{M}} = \sum_{h=1}^{\mathcal{H}} d_{(h)} + \sum_{s=1}^{\mathcal{S}} d_{(s)} + d_{(e)}$. We use a simple combinatorial construction of the mixed-curvature space, induced by the cartesian product (Sun et al., 2022). In Section 5 and Appendix 7.2.3 we describe how we heuristically identify the signature of $\mathbb{P}^{d_{\mathcal{M}}}$.

**Extending PageRank GNN to $\mathbb{P}$**: We choose GPRGNN (Chien et al., 2020) as the spectral backbone of `CUSP` (See Appendix 7.4 for more background on GPRGNN), becasue it jointly optimizes node feature and topological information extraction, which is different from other spectral GNNs like ChebyNet (Defferrard et al., 2016). The `GPR` weights automatically adjust to the node label pattern (homophilic or heterophilic). Given the node feature matrix $\mathbf{F} \in \mathbb{R}^{n \times d_f}$ and normalized symmetric *Cusp adjacency* matrix $\widetilde{\mathbf{A}}_n = \widetilde{\mathbf{D}}^{\frac{-1}{2}} \widetilde{\mathbf{A}} \widetilde{\mathbf{D}}^{\frac{-1}{2}}$, we first extract hidden state features for

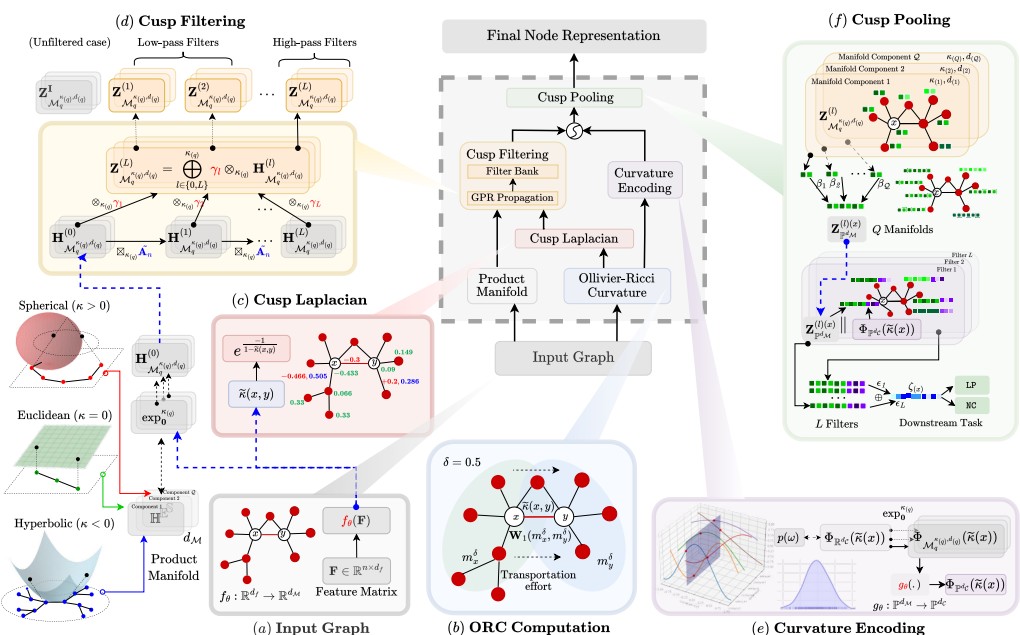

Figure 3: **Architecture of `CUSP`**. The input graph **(a)** is used to construct the *Cusp Laplacian* ($\widetilde{\mathbf{L}}$), based on the Ollivier-Ricci curvature **(b)**, as illustrated in **(c)**. The computed edge `ORC` (red), node `ORC` (Green), and curvature-based weights (blue) have been highlighted in the input graph. With $\widetilde{\mathbf{L}}$, **(d)** *Cusp Filtering* introduces multiple graph filters to capture different parts of the eigenspectrum. Each node receives a curvature positional encoding $\Phi$ in **(e)** as part of the **(f)** *Cusp Pooling* mechanism, which computes the relative importance of different filters and manifold components.

each node and then use `GPR` to propagate them. The process can be mathematically described as:

$$\mathbf{H}^{(0)}_{\mathcal{M}_q^{\kappa_{(q)}, d_{(q)}}} = \mathbf{exp}_0^{\kappa_{(q)}}\left(f_\theta(\mathbf{F})\right) \quad ; \mathbf{H}^{(l)}_{\mathcal{M}_q^{\kappa_{(q)}, d_{(q)}}} = \widetilde{\mathbf{A}}_n \boxtimes_{\bar{\kappa}_{(q)}} \mathbf{H}^{(l-1)}_{\mathcal{M}_q^{\kappa_{(q)}, d_{(q)}}} \tag{2}$$

$$\mathbf{Z}^{(L)}_{\mathcal{M}_q^{\kappa_{(q)}, d_{(q)}}} = \bigoplus_{l \in \{0,L\}}^{\kappa_{(q)}} \gamma_l \otimes_{\kappa_{(q)}} \mathbf{H}^{(l)}_{\mathcal{M}_q^{\kappa_{(q)}, d_{(q)}}} \quad ; \mathbf{Z}^{(L)}_{\mathbb{P}^{d_\mathcal{M}}} = ||_{q=1}^{\mathcal{Q}} \mathbf{Z}^{(L)}_{\mathcal{M}_q^{\kappa_{(q)}, d_{(q)}}} \tag{3}$$

where $f_\theta(.) : \mathbb{R}^{d_f} \to \mathbb{R}^{d_\mathcal{M}}$ represents a neural network with parameter set $\{\theta\}$ that generates the hidden state features of dimension $d^\mathcal{M}$. Here, $\mathbf{exp}_0^\kappa : \mathbb{R}^{d_\mathcal{M}} \to \mathcal{M}^{\kappa, d_\mathcal{M}}$ is the exponential map (Section 3) to transform the euclidean node features $\mathbf{F}$ to the component manifold $\mathcal{M}$, and $||$ is the attentional concatenation operator (discussed in Section 4.4). $\oplus_\kappa$, $\otimes_\kappa$ and $\boxtimes_\kappa$ denote *mobius* addition, $\kappa$-right-matrix-multiplication and $\kappa$-left-matrix-multiplication respectively (Appendix 7.2.4). These operations generalize vector addition and multiplication on $\kappa$-stereographic model. The GPR weights $\gamma_l$ are trained together with $\{\theta\}$ in an end-to-end fashion. The final mixed-curvature node embeddings after attention can be represented as $\mathbf{Z}^{(L)}_{\mathbb{P}^{d_\mathcal{M}}} \in \mathbb{P}^{n \times d_\mathcal{M}}$, such that $\sum_{q=1}^{\mathcal{Q}} d_{(q)} = d_\mathcal{M}$.

**Filter Bank**. The `GPR` component of the network may be viewed as a polynomial graph filter (See Appendix 7.4). Let $\widetilde{\mathbf{A}}_n = \mathbf{U}\boldsymbol{\Lambda}\mathbf{U}^T$ be the eigenvalue decomposition of $\widetilde{\mathbf{A}}_n$. Then, the polynomial graph filter equals $\sum_{l=0}^{L} \gamma_l \widetilde{\mathbf{A}}_n^l = \mathbf{U}g_{\gamma,L}(\boldsymbol{\Lambda})\mathbf{U}^T$, where $g_{\gamma,L}(\boldsymbol{\Lambda})$ is applied element-wise and $g_{\gamma,L}(\lambda) = \sum_{l=0}^{L} \gamma_l \lambda^l$. If one allows $\gamma_l$ to be negative and learnt adaptively, the graph filter will pass relevant high frequencies. Consequently, `CUSP` performs exceptionally well on heterophilic graphs.

**Theorem 1** (Informal (Chien et al., 2020))**.** *If $\gamma_l \geq 0 \ \forall l \in \{0, 1, ..., L\}$, $\sum_{l=0}^{L} \gamma_l = 1$ and $\exists l' > 0$ such that $\gamma_{l'} > 0$, then $g_{\gamma,L}(\cdot)$ is a low-pass graph filter. Also, if $\gamma_l = (-\alpha)^l, \alpha \in (0,1)$ and $L$ is large enough, then $g_{\gamma,L}(\cdot)$ is a high-pass graph filter.*

We encourage the reader to refer to Chien et al. (2020) for a detailed discussion on how different initializations of the `GPR` weights, as mentioned in the theorem above, assist in designing low-pass and high-pass filters. We present a proof of Theorem 1 in Appendix 7.4. Motivated by the limitation

`L1`, instead of a single filter, we construct a filter bank to focus on different parts of the eigenspectrum (to assist in both heterophilic and homophilic graphs). In an attempt to be representative of the higher order filters, the proposed filterbank is: $\Omega_{\mathbb{P}^{d}\mathcal{M}} = \left[\mathbf{Z}^{\mathbf{I}}_{\mathbb{P}^{d}\mathcal{M}}, \mathbf{Z}^{(1)}_{\mathbb{P}^{d}\mathcal{M}}, \mathbf{Z}^{(2)}_{\mathbb{P}^{d}\mathcal{M}}, \ldots, \mathbf{Z}^{(L)}_{\mathbb{P}^{d}\mathcal{M}}\right]$. Here, $\mathbf{Z}^{\mathbf{I}}_{\mathbb{P}^{d}\mathcal{M}}$ is the unfiltered case, where we pass the identity matrix $\mathbf{I}$ instead of $\widetilde{\mathbf{A}}_n$ for `GPR` propagation.

## 4.3 Functional Curvature Encoding

Recall our motivation that `CUSP` must be able to pay more attention to differently curved substructures in our model, depending on different tasks and datasets, while learning the final node representations. Specifically, our goal is to obtain a continuous functional mapping $\Phi : \mathbb{K}_{\mathbb{P}} \to \mathbb{P}^{d_C}$ from curvature domain to the $d_C$-dimensional product space to serve as positional encoding in our attention mechanism (Section 4.4). We approximate this in two steps, by considering the Ollivier-Ricci (`ORC`) discretization of curvature on our graph and using the Bochner's theorem (Moeller et al., 2016) from classical harmonic analysis. First, we construct a translation-invariant Euclidean curvature encoding map, and then map it to the product manifold. Without loss of generality, we assume that the curvature domain can be represented by the interval: $\mathbb{K} = [-1, 1]$, where $-1$ to $1$ is the *typical* range[1] of `ORC` in the observed data. Formally, we define the **Curvature Kernel** $\mathcal{K}_{\mathbb{R}} : \mathbb{K} \times \mathbb{K} \to \mathbb{R}$ with $\mathcal{K}_{\mathbb{R}}(\widetilde{\kappa}_a, \widetilde{\kappa}_b) := \left\langle \Phi_{\mathbb{R}^{d_C}}(\widetilde{\kappa}_a), \Phi_{\mathbb{R}^{d_C}}(\widetilde{\kappa}_b) \right\rangle$ and $\mathcal{K}_{\mathbb{R}}(\widetilde{\kappa}_a, \widetilde{\kappa}_b) = \Psi_{\mathbb{R}}(\widetilde{\kappa}_a - \widetilde{\kappa}_b)$, $\forall \widetilde{\kappa}_a, \widetilde{\kappa}_b \in \mathbb{K}$ for some $\Psi_{\mathbb{R}} : [-2, 2] \to \mathbb{R}$. The intuition behind choosing such a kernel for curvature values, lies in the fact that when comparing $\widetilde{\kappa}_a$ and $\widetilde{\kappa}_b$ we are only concerned with the relative difference between the two values. According to the Bochner's theroem (Moeller et al., 2016), a continuous, translation-invariant kernel $\mathcal{K}(\mathbf{x}, \mathbf{y}) = \Psi(\mathbf{x} - \mathbf{y})$ on $\mathbb{R}^d$ is positive definite if $\Psi$ is the Fourier transform of a non-negative probability measure on $\mathbb{R}$. Our kernel is translation-invariant, since $\mathcal{K}_{\mathbb{R}}(\widetilde{\kappa}_a + \widetilde{c}, \widetilde{\kappa}_b + \widetilde{c}) = \Psi_{\mathbb{R}}((\widetilde{\kappa}_a + \widetilde{c}) - (\widetilde{\kappa}_b + \widetilde{c})) = \mathcal{K}_{\mathbb{R}}(\widetilde{\kappa}_a, \widetilde{\kappa}_b)$ for any constant $\widetilde{c}$. The following theorems defines the Euclidean encoding and it's corresponding mapping to the product space.

**Definition 2.** *Let $\mathbb{R}^{d_C}$ be the ambient Euclidean space for a node's curvature encoding. The functional curvature encoding $\Phi_{\mathbb{R}^{d_C}} : \mathbb{K} \to \mathbb{R}^{d_C}$ for curvature $\widetilde{\kappa}(x) \in \mathbb{K}$ of node $x$, is defined as:*

$$\Phi_{\mathbb{R}^{d_C}}(\widetilde{\kappa}(x)) = \sqrt{\frac{1}{d_C}}\left[\cos(\omega_1\widetilde{\kappa}(x)), \sin(\omega_1\widetilde{\kappa}(x)), \ldots, \cos(\omega_{d_C}\widetilde{\kappa}(x)), \sin(\omega_{d_C}\widetilde{\kappa}(x))\right], \quad (4)$$

*where $\omega_1, \ldots, \omega_{d_C} \overset{i.i.d}{\sim} p(\omega)$ are sampled from a distribution $p(\omega)$. The corresponding mapping to the product manifold $\mathbb{P}^{d_C}$, is defined as:*

$$\Phi_{\mathbb{P}^{d_C}}(\widetilde{\kappa}(x)) = g_\theta\left(\|_{q=1}^{\mathcal{Q}} \mathbf{exp}_{\mathbf{0}}^{\kappa_{(q)}}(\Phi_{\mathbb{R}^{d_C}}(\widetilde{\kappa}(x)))\right) = g_\theta\left(\|_{q=1}^{\mathcal{Q}} \Phi_{\mathcal{M}_q^{\kappa_{(q)}, d_{(q)}}}(\widetilde{\kappa}(x))\right), \quad (5)$$

*where $\mathbf{exp}_{\mathbf{0}}^{\kappa_{(q)}} : \mathbb{R}^{d_C} \to \mathcal{M}_q^{\kappa_{(q)}, d_{(q)}}$ denotes the exponential map on the $q^{th}$ component manifold with curvature $\kappa_{(q)}$, $\|$ is the concatenation operator and $g_\theta : \mathbb{P}^{d_\mathcal{M}} \to \mathbb{P}^{d_C}$ is a Riemannian projector.*

It is easy to show that $\left\langle \Phi_{d_C}(\widetilde{\kappa}_a), \Phi_{d_C}(\widetilde{\kappa}_b) \right\rangle \approx \mathcal{K}(\widetilde{\kappa}_a, \widetilde{\kappa}_b)$. The unknown distribution $p(\omega)$ is estimated using the inverse cumulative distribution function (CDF) transformation as in Xu et al. (2020). Next, we prove the translation invariance of the curvature kernel in the product manifold setting.

**Theorem 2.** *The mixed-curvature kernel $\mathcal{K}_{\mathbb{P}}(\widetilde{\kappa}_a, \widetilde{\kappa}_b) := \left\langle \Phi_{\mathbb{P}^{d_C}}(\widetilde{\kappa}_a), \Phi_{\mathbb{P}^{d_C}}(\widetilde{\kappa}_b) \right\rangle$ is translation invariant, i.e. $\mathcal{K}_{\mathbb{P}}(\widetilde{\kappa}_a, \widetilde{\kappa}_b) = \Psi_{\mathbb{P}}(\widetilde{\kappa}_a - \widetilde{\kappa}_b)$.*

Please refer to Appendix 7.5 for the detailed proofs (derivations) of Definition 2 and Theorem 2. We employ $\Phi_{\mathbb{P}^{d_C}}(\widetilde{\kappa}(x))$ as the `ORC`-based positional encoding for each node $x$ in the final pooling layer, enabling it to incorporate local curvature information and compute the task-specific relevance of substructures with varying curvatures within the graph.

## 4.4 Cusp Pooling

The importance of constant-curvature component spaces depends on the downstream task. To this end, we propose a hierarchical attention mechanism called *Cusp Pooling*. We perform attentional

---

[1] `ORC` can theoretically vary from $-\infty$ to $1$, but in practice, its values usually lie within $[-1, 1]$ for the majority of graphs. For extreme scenarios, such as very sparse or highly clustered graphs, values may fall outside this range. In such instances, `ORC` is normalized to $[-1, 1]$.

concatenation to fuse constant-curvature representations across component spaces so as to learn mixed-curvature representations in the product space, weighing them appropriately. Specifically, we lift component encodings to the common tangent space, where we can perform the usual euclidean operations, and compute their centorid by the mean pooling. Then, we model the importance of a component by the position of the embedding relative to the centorid, parameterized by $\boldsymbol{\theta}$.

$$\boldsymbol{\mu}^{(L)} = \frac{1}{\mathcal{Q}} \sum_{q=1}^{\mathcal{Q}} \mathbf{log_0}^{\kappa_{(q)}} \big( \mathbf{W}_q \otimes_{\kappa_{(q)}} \mathbf{Z}^{(L)}_{\mathcal{M}_q^{\kappa_{(q)}, d_{(q)}}} \big) \quad \text{(Centroid using linear transformation } \mathbf{W}_q \text{)} \quad (6)$$

$$\tau_q = \sigma \big( \boldsymbol{\theta}^\top \big( \mathbf{log_0}^{\kappa_{(q)}} \big( \mathbf{W}_q \otimes_{\kappa_{(q)}} \mathbf{Z}^{(L)}_{\mathcal{M}_q^{\kappa_{(q)}, d_{(q)}}} \big) - \boldsymbol{\mu}^{(L)} \big) \big) \quad \text{(Relative importance of } \mathcal{M}_q \text{)} \quad (7)$$

Finally, with the learnable attentional weights $\beta_q = e^{\tau_q} / \sum_{q=0}^{Q} (e^{\tau_q})$, we perform attentional concatenation. Further, as motivated earlier, for every node $x$, we fuse the curvature embedding as positional encoding:

$$\mathbf{Z}^{(L)(x)}_{\mathbb{P}^d_\mathcal{M}} = \Big\|_{q=1}^{\mathcal{Q}} \Big( \beta_q \otimes_{\kappa_{(q)}} \mathbf{Z}^{(L)(x)}_{\mathcal{M}_q^{\kappa_{(q)}, d_{(q)}}} \Big) \quad ; \boldsymbol{\zeta}^{(L)}_{(x)} = \mathbf{Z}^{(L)(x)}_{\mathbb{P}^d_\mathcal{M}} \Big\| \Phi_{\mathbb{P}^d_\mathcal{M}} \big( \widetilde{\kappa}(x) \big) \quad (8)$$

We weigh different filters in our filter bank to yield the final node representation $\boldsymbol{\zeta}_{(x)}$ for $x$ as $\boldsymbol{\zeta}_{(x)} = \sum_{l=1}^{L} \epsilon_l \boldsymbol{\zeta}^{(l)}_{(x)}$. Here, $\boldsymbol{\zeta} \in \mathbb{P}^{n \times (d_\mathcal{M} + d_\mathcal{C})}$ contains the final node embeddings, and $\epsilon_l$ is a learnable parameter to weigh the importance of different filters in our bank. These node embeddings are then used for the final downstream tasks of node classification and link prediction. In the following section, we lay out the empirical results to validate the efficacy of CUSP.

## 5 EXPERIMENTATION

**Datasets**. We evaluate CUSP on the Node Classification (NC) and Link Prediction (LP) tasks using eight benchmark datasets. These include **(a) Homophilic** datasets such as *(i) Citation networks* – Cora, Citeseer and PubMed (Sen et al., 2008; Yang et al., 2016), and **(b) Heterophilic** datasets, which comprise *(i) Wikipedia graphs* – Chameleon and Squirrel (Rozemberczki et al., 2021), *(ii) Actor co-occurrence network* (Tang et al., 2009), and *(iii) Webpage graphs* from WebKB [2] – Texas and Cornell. The dataset statistics and their respective homophily ratios are detailed in Table 1. Refer to Appendix 7.2.1 for the discrete curvature and eigenspectrum distributions of these datasets.

**Baselines**. To ensure a fair comparison, we evaluate CUSP against three types of baselines: **(a) Spatial–Euclidean**, including traditional methods such as GCN (Kipf & Welling, 2016), GAT (Veličković et al., 2017), and GraphSAGE (Hamilton et al., 2017). **(b) Spatial–Riemannian**, comprising

| Dataset | Cora | Citeseer | PubMed | Chameleon | Squirrel | Actor | Texas | Cornell |
|---|---|---|---|---|---|---|---|---|
| Classes | 7 | 6 | 5 | 5 | 5 | 5 | 5 | 5 |
| Features | 1433 | 3703 | 500 | 2325 | 2089 | 932 | 1703 | 1703 |
| Nodes | 2708 | 3327 | 19717 | 2277 | 5201 | 7600 | 183 | 183 |
| Edges | 5278 | 4552 | 44324 | 31371 | 198353 | 26659 | 279 | 277 |
| $\mathcal{H}$ | 0.825 | 0.718 | 0.792 | 0.247 | 0.217 | 0.215 | 0.057 | 0.301 |

Table 1: Data statistics and homophily ratio ($\mathcal{H}$).

*(i) Constant-curvature models* like HGCN (Chami et al., 2019) and HGAT (Zhang et al., 2021b), and *(ii) Mixed-curvature GNNs* such as $\kappa$GCN (Bachmann et al., 2020), $\mathcal{Q}$GCN (Xiong et al., 2022), and SelfMGNN (Sun et al., 2022). **(c) Spectral**, including ChebyNet (Defferrard et al., 2016), Bern-Net (He et al., 2021), GPRGNN (Chien et al., 2020), and FiGURe (Ekbote et al., 2024) (More details in Appendix 7.6.2). There are no existing methods that integrate both spectral and geometric signals.

**Experimental Settings**. For the transductive LP task, we randomly split edges into $85\%/5\%/10\%$ for training, validation and test sets, while for transductive NC task, we use the $60\%/20\%/20\%$ split. The results are averaged over 10 random splits and their $95\%$ confidence intervals are reported. We report the AUC-ROC and F1 Score metrics for LP and NC respectively. For computing ORC (in *Cusp Laplacian*), we use $\delta = 0.5$, i.e. equal probability mass is retained by the node and distributed among it's neighbors. We adopt the ORC implementation from Ni et al. (2019), and use the *Sinkhorn* algorithm (Sinkhorn & Knopp, 1967) to approximate the $\mathbf{W}_1$ distance. See Appendix 7.2.2 for more computational details and complexity analysis. Next, we adopt the implementation of the $\kappa$-stereographic product manifold from Geoopt library [3]. We heuristically determine the *signature* of our manifold $\mathbb{P}$ (i.e. component manifolds) using the discrete ORC curvature of the

---

[2] http://www.cs.cmu.edu/afs/cs.cmu.edu/project/theo-11/www/wwkb
[3] https://github.com/geoopt/geoopt

| Baseline | Cora | Citeseer | PubMed | Chameleon | Actor | Squirrel | Texas | Cornell | Av. ΔGain |
|---|---|---|---|---|---|---|---|---|---|
| GCN | $75.21_{\pm 0.28}$ | $67.30_{\pm 1.05}$ | $83.75_{\pm 0.07}$ | $61.16_{\pm 0.23}$ | $31.12_{\pm 0.96}$ | $43.06_{\pm 0.33}$ | $75.61_{\pm 0.07}$ | $67.72_{\pm 1.19}$ | 11.95 |
| GAT | $76.70_{\pm 0.13}$ | $66.23_{\pm 0.85}$ | $82.83_{\pm 0.22}$ | $63.10_{\pm 0.77}$ | $32.65_{\pm 0.23}$ | $43.90_{\pm 0.01}$ | $76.09_{\pm 0.77}$ | $74.01_{\pm 0.01}$ | 10.63 |
| SAGE | $71.88_{\pm 0.91}$ | $70.01_{\pm 0.64}$ | $81.09_{\pm 0.13}$ | $59.99_{\pm 0.89}$ | $36.73_{\pm 0.01}$ | $41.11_{\pm 1.16}$ | $77.11_{\pm 0.45}$ | $69.91_{\pm 0.24}$ | 11.59 |
| HGCN | $78.50_{\pm 0.14}$ | $69.55_{\pm 0.39}$ | $83.72_{\pm 0.21}$ | $60.18_{\pm 0.57}$ | $35.89_{\pm 0.29}$ | $39.93_{\pm 0.35}$ | $88.11_{\pm 1.12}$ | $72.88_{\pm 1.15}$ | 8.97 |
| HGAT | $77.12_{\pm 0.01}$ | $70.12_{\pm 0.92}$ | $84.02_{\pm 0.19}$ | $62.43_{\pm 0.59}$ | $35.12_{\pm 0.27}$ | $41.78_{\pm 0.37}$ | $85.56_{\pm 1.10}$ | $73.12_{\pm 0.18}$ | 8.91 |
| $\kappa$GCN | $78.71_{\pm 1.37}$ | $68.14_{\pm 0.34}$ | $85.18_{\pm 0.52}$ | $62.12_{\pm 0.49}$ | $34.57_{\pm 0.24}$ | $43.04_{\pm 0.31}$ | $85.03_{\pm 0.63}$ | $86.36_{\pm 0.64}$ | 7.06 |
| $\mathcal{Q}$GCN | $79.64_{\pm 0.38}$ | $71.15_{\pm 1.11}$ | $84.76_{\pm 0.13}$ | $61.83_{\pm 1.01}$ | $32.24_{\pm 0.65}$ | $46.65_{\pm 0.90}$ | $82.76_{\pm 0.07}$ | $83.90_{\pm 0.71}$ | 7.20 |
| SelfMGNN | $80.19_{\pm 0.60}$ | $70.91_{\pm 0.38}$ | $82.81_{\pm 0.34}$ | $64.97_{\pm 0.54}$ | $38.99_{\pm 0.23}$ | $49.79_{\pm 0.29}$ | $90.92_{\pm 0.65}$ | $85.01_{\pm 0.69}$ | 4.62 |
| ChebyNet | $71.09_{\pm 0.91}$ | $66.67_{\pm 0.38}$ | $83.83_{\pm 0.42}$ | $59.96_{\pm 1.51}$ | $38.02_{\pm 0.01}$ | $45.67_{\pm 0.11}$ | $79.08_{\pm 0.96}$ | $71.33_{\pm 1.04}$ | 10.61 |
| BernNet | $73.34_{\pm 0.53}$ | $62.12_{\pm 2.09}$ | $82.15_{\pm 0.13}$ | $62.03_{\pm 0.12}$ | $33.55_{\pm 0.24}$ | $42.81_{\pm 0.66}$ | $75.11_{\pm 0.14}$ | $65.56_{\pm 1.02}$ | 12.98 |
| GPRGNN | $79.49_{\pm 0.31}$ | $67.61_{\pm 0.38}$ | $84.07_{\pm 0.09}$ | $65.09_{\pm 0.43}$ | $37.43_{\pm 1.09}$ | $47.51_{\pm 0.23}$ | $88.34_{\pm 0.09}$ | $87.21_{\pm 0.70}$ | 5.58 |
| FiGURe | $80.01_{\pm 0.09}$ | $71.26_{\pm 0.41}$ | $83.89_{\pm 0.11}$ | $67.18_{\pm 0.02}$ | $38.31_{\pm 0.36}$ | $48.71_{\pm 1.02}$ | $86.66_{\pm 0.62}$ | $85.01_{\pm 0.68}$ | 4.94 |
| CUSP | $\mathbf{83.45_{\pm 0.15}}$ | $\mathbf{74.21_{\pm 0.02}}$ | $\mathbf{87.99_{\pm 0.45}}$ | $\mathbf{70.23_{\pm 0.61}}$ | $\mathbf{43.91_{\pm 0.11}}$ | $\mathbf{52.98_{\pm 0.25}}$ | $\mathbf{94.03_{\pm 0.72}}$ | $\mathbf{92.31_{\pm 0.09}}$ | 0.0 |
| ΔImp. | 3.26% | 2.95% | 2.81% | 4.05% | 5.32% | 3.19% | 3.11% | 5.10% | |

Table 3: Performance comparision of CUSP with baselines for NC task (Mean F1 Score $\pm$ 95% confidence interval). **First**, Second and Third best performing models are highlighted. **Av. Δ Gain** represents the average gain of CUSP over the model in that row, averaged across the different datasets. **ΔImp.** implies the % improvement of CUSP over the second best performing baseline.

input graph. The key idea is that the underlying curvature of the manifold must align with ORC. However, this discussion, along with how we initialise the learnable curvatures for the component manifolds, has been reserved for the Appendix 7.6.4. For all experiments, we choose the total manifold dimension as $d_{\mathcal{M}} = 48$ and learning rate as $4\text{e-}3$. We use the filter bank $\Omega_{\mathbb{P}^{d_{\mathcal{M}}}} = \left[\mathbf{Z}^{\mathbf{I}}_{\mathbb{P}^{d_{\mathcal{M}}}}, \mathbf{Z}^{(1)}_{\mathbb{P}^{d_{\mathcal{M}}}}, \mathbf{Z}^{(2)}_{\mathbb{P}^{d_{\mathcal{M}}}}, \dots, \mathbf{Z}^{(L)}_{\mathbb{P}^{d_{\mathcal{M}}}}\right]$, with $L = 10$. For the GPR weights, we experiment with different initializations, $\alpha \in \{0.1, 0.3, 0.5, 0.9\}$. We list the hyperparameter settings in Appendix 7.6.5.

**Analysis**. Tables 3 and 8 present a comparative performance analysis of CUSP against baseline models for the NC and LP tasks respectively. **(a)** CUSP consistently outperforms all baseline models across both homophilic and heterophilic datasets, achieving an improvement up to 5.32% in F1-score for NC, up to 5.11% increase in ROC-AUC score for LP. **(b)** Riemannian baselines perform well for homophilic datasets, while performing poorly in heterophilic cases (Evidence of limitation L1). **(c)** While spectral baselines like ChebyNet and BernNet perform poorly on heterophilic tasks because they act as *low-pass*, owing to the fixed filters, FiGURe performs well across all tasks because it uses a filter bank to handle different bands in the eigenspectrum. **(d)** Mixed-curvature baselines ($\kappa$GCN and $\mathcal{Q}$GCN) outperform constant-curvature GNNs (HGCN and HGAT) because they capture the complex graph geometry (Evidence of limitation L2).

Table 4 presents the learnt filter weights ($\epsilon$), highlighting the distinct preferences of homophilic and heterophilic datasets. Homophilic datasets, such as Citeseer and PubMed, emphasize low-pass filters, with the highest weights assigned to lower-order filters, $\mathbf{Z}^{(2)}$ and $\mathbf{Z}^{(3)}$, at 45.94% and 47.97% respectively. In contrast,

| CUSP | Cora | Citeseer | PubMed | Chameleon | Actor | Squirrel | Texas | Cornell |
|---|---|---|---|---|---|---|---|---|
| $\mathbb{H}^{24} \times \mathbb{S}^{24}$ | $82.50_{\pm 0.18}$ | $73.20_{\pm 0.16}$ | $87.20_{\pm 0.26}$ | $69.42_{\pm 0.01}$ | $42.70_{\pm 0.22}$ | $52.00_{\pm 0.20}$ | $81.97_{\pm 0.24}$ | $87.20_{\pm 0.02}$ |
| $(\mathbb{H}^8)^2 \times (\mathbb{S}^8)^2 \times \mathbb{E}^{16}$ | $82.80_{\pm 0.20}$ | $73.50_{\pm 0.18}$ | $85.50_{\pm 0.28}$ | $69.80_{\pm 0.03}$ | $43.06_{\pm 0.19}$ | $50.76_{\pm 0.18}$ | $91.60_{\pm 0.23}$ | $91.60_{\pm 0.24}$ |
| $\mathbb{H}^8 \times \mathbb{S}^8 \times \mathbb{E}^{32}$ | $82.70_{\pm 0.32}$ | $73.40_{\pm 0.17}$ | $86.40_{\pm 0.29}$ | $69.60_{\pm 0.07}$ | $40.73_{\pm 0.20}$ | $51.20_{\pm 0.19}$ | $\mathbf{94.03_{\pm 0.72}}$ | $\mathbf{92.31_{\pm 0.09}}$ |
| $\mathbb{H}^{16} \times (\mathbb{S}^{16})^2$ | $81.83_{\pm 0.20}$ | $72.72_{\pm 0.13}$ | $85.90_{\pm 0.60}$ | $\mathbf{70.23_{\pm 0.61}}$ | $42.50_{\pm 0.21}$ | $\mathbf{52.98_{\pm 0.25}}$ | $92.50_{\pm 0.22}$ | $83.11_{\pm 0.36}$ |
| $(\mathbb{H}^{16})^2 \times \mathbb{E}^{16}$ | $81.90_{\pm 0.17}$ | $72.50_{\pm 0.15}$ | $\mathbf{87.99_{\pm 0.45}}$ | $65.30_{\pm 0.28}$ | $\mathbf{43.91_{\pm 0.11}}$ | $51.10_{\pm 0.24}$ | $93.20_{\pm 0.19}$ | $91.20_{\pm 0.45}$ |
| $\mathbb{H}^{24} \times \mathbb{E}^{24}$ | $81.60_{\pm 0.16}$ | $72.30_{\pm 0.14}$ | $87.50_{\pm 0.64}$ | $67.50_{\pm 0.23}$ | $43.50_{\pm 0.16}$ | $48.30_{\pm 0.23}$ | $91.45_{\pm 0.61}$ | $91.39_{\pm 0.98}$ |
| $\mathbb{S}^{24} \times \mathbb{E}^{24}$ | $80.80_{\pm 0.41}$ | $71.80_{\pm 0.19}$ | $80.90_{\pm 0.31}$ | $69.20_{\pm 0.21}$ | $36.44_{\pm 0.23}$ | $51.83_{\pm 0.21}$ | $90.99_{\pm 0.21}$ | $90.00_{\pm 0.10}$ |
| $\mathbb{H}^{16} \times \mathbb{S}^{16} \times \mathbb{E}^{16}$ | $\mathbf{83.45_{\pm 0.15}}$ | $\mathbf{74.21_{\pm 0.02}}$ | $85.80_{\pm 0.27}$ | $70.17_{\pm 0.17}$ | $43.20_{\pm 0.18}$ | $52.71_{\pm 0.17}$ | $93.80_{\pm 0.18}$ | $91.80_{\pm 0.21}$ |
| $(\mathbb{S}^8)^2 \times \mathbb{E}^{32}$ | $80.50_{\pm 0.30}$ | $71.50_{\pm 0.18}$ | $79.30_{\pm 0.32}$ | $68.18_{\pm 0.20}$ | $37.14_{\pm 0.24}$ | $51.60_{\pm 0.22}$ | $92.80_{\pm 0.22}$ | $90.80_{\pm 0.53}$ |
| $(\mathbb{H}^{16})^3$ | $81.00_{\pm 0.19}$ | $72.00_{\pm 0.16}$ | $87.70_{\pm 0.92}$ | $68.11_{\pm 0.25}$ | $43.70_{\pm 0.17}$ | $45.67_{\pm 0.25}$ | $88.15_{\pm 0.25}$ | $85.01_{\pm 0.51}$ |

Table 2: Performance comparison of CUSP with different manifold signatures for Node Classification (NC). Best performing *signatures* are in **Bold**, and cases with a large decline in performance because of manifold mismatch are in Blue.

heterophilic datasets like Actor and Cornell favor high-pass filters, attributing 40.09% and 28.56% to higher-order filters, $\mathbf{Z}^{(5)}$ and $\mathbf{Z}^{(9)}$. In the next section, we perform extensive ablation studies to evaluate the effectiveness of all components of CUSP.

## 5.1 ABLATION STUDY

■ **Impact of product manifold *signatures***. Different datasets leverage substructures with varying curvatures as inductive biases for downstream tasks, leading to differing performance across product manifold signatures. Table 5 shows the learned curvature and weight ($\beta$) for the best-performing

| Dataset | I | $\mathbf{Z}^{(1)}$ | $\mathbf{Z}^{(2)}$ | $\mathbf{Z}^{(3)}$ | $\mathbf{Z}^{(4)}$ | $\mathbf{Z}^{(5)}$ | $\mathbf{Z}^{(6)}$ | $\mathbf{Z}^{(7)}$ | $\mathbf{Z}^{(8)}$ | $\mathbf{Z}^{(9)}$ | $\mathbf{Z}^{(10)}$ |
|---|---|---|---|---|---|---|---|---|---|---|---|
| Cora | **0.1729** | 0.1586 | 0.0695 | 0.0438 | 0.0446 | 0.0221 | 0.1525 | 0.0526 | 0.0711 | 0.0739 | 0.1385 |
| Citeseer | 0.0478 | 0.0506 | **0.4594** | 0.0183 | 0.0765 | 0.0187 | 0.1093 | 0.0755 | 0.0686 | 0.0232 | 0.0522 |
| PubMed | 0.1666 | 0.0342 | 0.0132 | **0.4795** | 0.1116 | 0.0289 | 0.0057 | 0.0153 | 0.0564 | 0.0442 | 0.0444 |
| Chameleon | 0.0232 | 0.1679 | 0.0536 | 0.1696 | 0.0121 | 0.0137 | 0.0392 | **0.2178** | 0.1356 | 0.0140 | 0.1532 |
| Actor | 0.0360 | 0.0753 | 0.0332 | 0.0337 | 0.0101 | **0.4009** | 0.2419 | 0.0096 | 0.1155 | 0.0089 | 0.0347 |
| Squirrel | 0.0306 | 0.0796 | 0.0719 | **0.2562** | 0.0351 | 0.0423 | 0.0415 | 0.1322 | 0.1050 | 0.1105 | 0.0951 |
| Texas | 0.1052 | 0.0308 | 0.0664 | 0.0490 | 0.0766 | 0.0274 | 0.0995 | 0.0402 | 0.1741 | 0.0639 | **0.2667** |
| Cornell | 0.1159 | 0.0302 | 0.0671 | 0.0434 | 0.0747 | 0.0249 | 0.0966 | 0.0358 | 0.1691 | **0.2856** | 0.0568 |

| Dataset | Signature |
|---|---|
| Cora | $\mathbb{H}^{16}(-0.36, 0.34) \times \mathbb{S}^{16}(+0.68, 0.29) \times \mathbb{E}^{16}(0, 0.37)$ |
| Citeseer | $\mathbb{H}^{16}(-0.62, 0.36) \times \mathbb{S}^{16}(+.55, 0.49) \times \mathbb{E}^{16}(0, 0.15)$ |
| PubMed | $\mathbb{H}^{16}(-0.91, 0.40) \times \mathbb{H}^{16}(-0.37, 0.36) \times \mathbb{E}^{16}(0, 0.24)$ |
| Chameleon | $\mathbb{H}^{16}(-0.21, 0.22) \times \mathbb{S}^{16}(+0.13, 0.29) \times \mathbb{S}^{16}(+0.59, 0.49)$ |
| Actor | $\mathbb{H}^{16}(-0.87, 0.39) \times \mathbb{H}^{16}(-0.53, 0.36) \times \mathbb{E}^{16}(0, 0.25)$ |
| Squirrel | $\mathbb{H}^{16}(-0.28, 0.14) \times \mathbb{S}^{16}(+0.31, 0.39) \times \mathbb{S}^{16}(+0.67, 0.53)$ |
| Texas | $\mathbb{H}^{8}(-0.45, 0.32) \times \mathbb{S}^{8}(+0.34, 0.23) \times \mathbb{E}^{32}(0, 0.45)$ |
| Cornell | $\mathbb{H}^{8}(-0.50, 0.14) \times \mathbb{S}^{8}(+0.23, 0.19) \times \mathbb{E}^{32}(0, 0.67)$ |

Table 4: Learned filter weights (NC) for the top-performing split, distinguishing between homophilic (favoring low-pass filters) and heterophilic (favoring high-pass filters). **First**, second, and third highest filter weights are highlighted.

Table 5: Learning results (NC) of CUSP for the best performing product signature − *manifold* (dim) (curvature, weight).

configurations. Table 2 highlights performance across multiple *signatures* (which are heuristically determined as discussed in Appendix 7.6.4), with large degradation (marked in blue) when there is a mismatch between manifold and graph curvature. For example, Squirrel, which has a predominantly positively curved structure (Figure 7.2.1), performs poorly (drop of $\sim 6.5\%$) on signatures dominated by negative curvature, such as $\mathbb{S}^{24} \times \mathbb{E}^{24}$ and $(\mathbb{H}^{16})^3$.

■ **Impact of mixed-curvature space**. Table 6 outlines the comparison of CUSP with its Euclidean variant $\text{CUSP}_{euc}$, where all representations are learned in Euclidean space, omitting *Cusp pooling*. On average, $\text{CUSP}_{euc}$ experiences a $\sim 3.2\%$ performance drop across all datasets for the NC task, highlighting the importance of capturing mixed-curvature signals. Notably, the performance degradation is more pronounced on heterophilic datasets ($\sim 4\%$) compared to homophilic datasets ($\sim 2\%$). This suggests that CUSP's integration of spectral and geometric properties is crucial, particularly for heterophilic data where capturing the eigenspectrum is more relevant.

■ **Impact of Cusp Laplacian**. To assess the effectiveness of the *Cusp Laplacian*, we conducted an ablation study by replacing the Cusp Laplacian-based adjacency matrix $\widetilde{\mathbf{A}}$, with the standard graph adjacency matrix (for Cusp filtering) in $\text{CUSP}_{lap}$. This resulted in a performance drop of $1.57\%$ (avg.) across all datasets. The relatively modest degradation highlights the Cusp Laplacian's role in capturing curvature information.

| Dataset | CUSP | $\text{CUSP}_{euc}$ | $\text{CUSP}_{lap}$ | $\text{CUSP}_{enc}$ | $\text{CUSP}_{pool}$ | $\text{CUSP}_{fil}$ |
|---|---|---|---|---|---|---|
| Cora | **83.45**$_{\pm 0.15}$ | 80.78$_{\pm 0.13}$ | 81.95$_{\pm 0.34}$ | 82.61$_{\pm 0.25}$ | 80.11$_{\pm 0.22}$ | 81.20$_{\pm 0.29}$ |
| Citeseer | **74.21**$_{\pm 0.02}$ | 72.50$_{\pm 0.19}$ | 72.73$_{\pm 0.30}$ | 71.95$_{\pm 0.28}$ | 73.10$_{\pm 0.30}$ | 72.87$_{\pm 0.27}$ |
| PubMed | **87.99**$_{\pm 0.45}$ | 85.23$_{\pm 0.25}$ | 86.67$_{\pm 0.10}$ | 87.11$_{\pm 0.12}$ | 86.23$_{\pm 0.10}$ | 87.29$_{\pm 0.09}$ |
| Chameleon | **70.23**$_{\pm 0.61}$ | 66.47$_{\pm 0.56}$ | 68.12$_{\pm 0.35}$ | 67.83$_{\pm 0.31}$ | 68.47$_{\pm 0.32}$ | 66.12$_{\pm 0.34}$ |
| Actor | **43.91**$_{\pm 0.11}$ | 39.03$_{\pm 0.09}$ | 43.03$_{\pm 0.27}$ | 41.12$_{\pm 0.28}$ | 40.03$_{\pm 0.27}$ | 38.81$_{\pm 0.29}$ |
| Squirrel | **52.98**$_{\pm 0.25}$ | 49.92$_{\pm 0.36}$ | 51.39$_{\pm 0.35}$ | 50.92$_{\pm 0.35}$ | 51.03$_{\pm 0.11}$ | 48.13$_{\pm 0.17}$ |
| Texas | **94.03**$_{\pm 0.72}$ | 90.15$_{\pm 0.61}$ | 92.15$_{\pm 0.52}$ | 93.15$_{\pm 0.55}$ | 92.15$_{\pm 0.60}$ | 90.27$_{\pm 0.62}$ |
| Cornell | **92.31**$_{\pm 0.09}$ | 89.46$_{\pm 0.13}$ | 90.46$_{\pm 0.17}$ | 91.06$_{\pm 0.28}$ | 90.46$_{\pm 0.07}$ | 89.73$_{\pm 0.09}$ |
| **Avg. $\Delta$ Gain** | 0 | 3.19 | 1.57 | 1.67 | 2.19 | 3.08 |

Table 6: Ablation study (NC) results on benchmark datasets. **Av. $\Delta$ Gain** represents the average gain of CUSP over the model in that column, averaged across the different datasets.

■ **Impact of the filter bank, curvature encoding and Cusp pooling**. $\text{CUSP}_{fil}$ replaces the filter bank with a single learnable filter, as a result the performance degrades largely on heterophilic datasets ($\sim 4\%$), while the degradation in performance is not that large on homophilic datasets ($\sim 2.5\%$). Further, we remove the curvature-based positional encoding in $\text{CUSP}_{enc}$ and replace the Cusp pooling mechanism with simple concatenation in $\text{CUSP}_{pool}$. We observe a consistent decline in performance across both these ablations. Owing to spatial limitations, we present the experimental results for the LP task in Appendix 7.6.3.

## 6 CONCLUSION

In this paper, we propose to unify *Spectral* and *Curvature* signals in a graph for learning optimal graph representations, aiming to inspire further research in this area. CUSP introduces a graph learning paradigm parameterized by spectral filters on a mixed-curvature product manifold. We propose a new curvature-informed *Cusp Laplacian* operator to capture the underlying geometry, use it to define a novel GPR-based spectral filter-bank (*Cusp Filtering*) and introduce an attention-based pooling mechanism to fuse representations from multiple mixed-curvature graph filters (*Cusp Pooling*). CUSP outperforms the state-of-the-art baselines for node classification and link prediction over homophilic and heterophilic datasets, highlighting the efficacy of combining spectrum and curvature (geometry), in learning graph representations.

ACKNOWLEDGEMENTS

We would like to sincerely thank Prof. Geoffrey J. Gordon (ggordon@cs.cmu.edu) from Carnegie Mellon University for his invaluable feedback and comments throughout multiple iterations of this paper, which greatly helped improve its quality.

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

# Table of Contents      **14**

# 7 APPENDIX

## 7.1 NOTATION TABLE

| Notation | Reference |
|---|---|
| $\mathbb{H}$ | Hyperbolic manifold |
| $\mathbb{S}$ | Spherical manifold |
| $\mathbb{E}$ | Euclidean manifold |
| $\mathbb{P}^{d_{\mathcal{M}}}$ | Product manifold of dimension $d_{\mathcal{M}}$ |
| $\kappa$ | Continous manifold curvature |
| $\widetilde{\kappa}$ | Ollivier-Ricci Curvature (ORC) |
| $\widetilde{\kappa}(x, y)$ | ORC of edge $\{x, y\}$ |
| $\widetilde{\kappa}(x)$ | ORC of node $x$ |
| $m_x^{\delta}$ | Probability mass assigned to node $x$ for ORC computation |
| $\delta_{ij}$ | Kroneckner delta function |
| $\mathcal{N}(x)$ | Neighbourhood set of node $x$ |
| $x \sim y$ | This implies that $x$ and $y$ are adjacent nodes |
| $\delta$ | ORC neighbourhood weighting parameter |
| $\mathbf{W}_1(.)$ | Wasserstein-1 distance |
| $d_{\mathcal{G}}(x, y)$ | Shortest path (graph distance) between nodes $x$ and $y$ on graph $\mathcal{G}$ |
| $\mathcal{M}_i^{\kappa_i, d_i}$ | Constant-curvature manifold with dimension $d_i$ and curvature $\kappa_i$. $\mathcal{M}_i \in \{\mathbb{H}, \mathbb{S}, \mathbb{E}\}$ |
| $\widetilde{\mathbf{L}}, \widetilde{\mathbf{D}}, \widetilde{\mathbf{A}}$ | Cusp Laplacian operator; $\widetilde{\mathbf{L}} = \widetilde{\mathbf{D}} - \widetilde{\mathbf{A}}$ |
| $\widetilde{\mathbf{L}}_n, \widetilde{\mathbf{A}}_n$ | Normalized Cusp Laplacian and Adjacency matrices; $\widetilde{\mathbf{A}}_n = \widetilde{\mathbf{D}}^{\frac{-1}{2}} \widetilde{\mathbf{A}} \widetilde{\mathbf{D}}^{\frac{-1}{2}}$ |
| $\mathbf{L}, \mathbf{D}, \mathbf{A}$ | Traditional graph Laplacian, Adjacency and Degree matrices; $\mathbf{L} = \mathbf{D} - \mathbf{A}$ |
| $d_f$ | Input graph feature dimension |
| $d_{\mathcal{M}}$ | Total dimension of the product manifold |
| $d_{\mathcal{C}}$ | Total dimension of the curvature embedding |
| $\mathbf{F} \in \mathbb{R}^{n \times d_f}$ | Input feature matrix |
| $\exp_0^{\kappa} : \mathbb{R}^{d_{\mathcal{M}}} \to \mathcal{M}^{\kappa}$ | Exponential map, to map from tangent plane (Euclidean) to the product manifold |
| $\oplus_{\kappa}$ | *Mobius* addition |
| $\otimes_{\kappa}$ | $\kappa$-right-matrix-multiplication |
| $\boxtimes_{\kappa}$ | $\kappa$-left-matrix-multiplication |
| $\boldsymbol{\zeta}_{(x)}$ | Final node representation for node $x$ |
| $\epsilon_l$ | Learnable weight of $l^{th}$ filter |
| $\boldsymbol{\zeta} \in \mathbb{P}^{n \times (d_{\mathcal{M}} + d_{\mathcal{C}})}$ | Matrix containing final node embeddings |
| $\beta_q$ | Learnable weight of $q^{th}$ component manifold |
| $\tau_q$ | Relative importance of manifold $\mathcal{M}_q$ in Cusp Pooling |
| $L$ | Total number of filters |
| $\mathcal{Q}$ | Total number of components in product manifold |
| $l$ | Used to denote the $l^{th}$ filter |
| $\mathbf{Z}_{\mathbb{P}^{d_{\mathcal{M}}}}^{(L)(x)}$ | The GPR-based node representation for filter with $L$ layers, on manifold $\mathbb{P}^{d_{\mathcal{M}}}$ |
| $\kappa_{(q)}$ | Manifold curvature of $q^{th}$ component in product manifold |
| $d_{(q)}$ | Manifold dimension of $q^{th}$ component in product manifold |
| $\mathbf{H}_{\mathcal{M}_q^{\kappa_{(q)}, d_{(q)}}}^{(l)}$ | Hidden state representation after $l$ layers of GPR, on $q^{th}$ manifold component with curvature $\kappa_{(q)}$ and dimension $d_{(q)}$ |
| $\gamma_l$ | GPR weight for $l^{th}$ layer in the filter, while propagation GPR score. |
| $\alpha$ | Initialising parameter for GPR |
| Signature | $\mathbb{P}^{d_{\mathcal{M}}} = \times_{q=1}^{\mathcal{Q}} \mathcal{M}_q^{\kappa_{(q)}, d_{(q)}} = \left(\times_{h=1}^{\mathcal{H}} \mathbb{H}_h^{\kappa_{(h)}, d_{(h)}}\right) \times \left(\times_{s=1}^{\mathcal{S}} \mathbb{S}_s^{\kappa_{(s)}, d_{(s)}}\right) \times \mathbb{E}^{d_{(e)}}$ |
| $\psi : V \to \mathbb{R}$ | A function defined on vertex set $V$ |
| $\Phi_{\mathbb{P}^{d_{\mathcal{C}}}}(\widetilde{\kappa}(x))$ | Curvature encoding on product manifold |
| $\mathcal{K}_{\mathbb{P}}(\widetilde{\kappa}_a, \widetilde{\kappa}_b)$ | $\mathcal{K}_{\mathbb{P}}(\widetilde{\kappa}_a, \widetilde{\kappa}_b) := \left\langle \Phi_{\mathbb{P}^{d_{\mathcal{C}}}}(\widetilde{\kappa}_a), \Phi_{\mathbb{P}^{d_{\mathcal{C}}}}(\widetilde{\kappa}_b) \right\rangle$ is the curvature kernel |
| $\lambda_i$ | $i^{th}$ eigenvalue of $\widetilde{\mathbf{A}}_n$ |
| $\widetilde{\lambda}_i$ | $i^{th}$ eigenvalue of $\widetilde{\mathbf{L}}_n$ |
| $f_{\theta}(.) : \mathbb{R}^{d_f} \to \mathbb{R}^{d_{\mathcal{M}}}$ | Neural network with parameter set $\{\theta\}$ that generates the hidden state features before feeding input to Cusp Filtering. |
| $g_{\theta}(.) : \mathbb{R}^{d_{\mathcal{M}}} \to \mathbb{R}^{d_{\mathcal{C}}}$ | Neural network projector used in curvature encoding |

## 7.2 MORE ON PRELIMINARIES

### 7.2.1 ANALYSIS OF REAL-WORLD GRAPHS

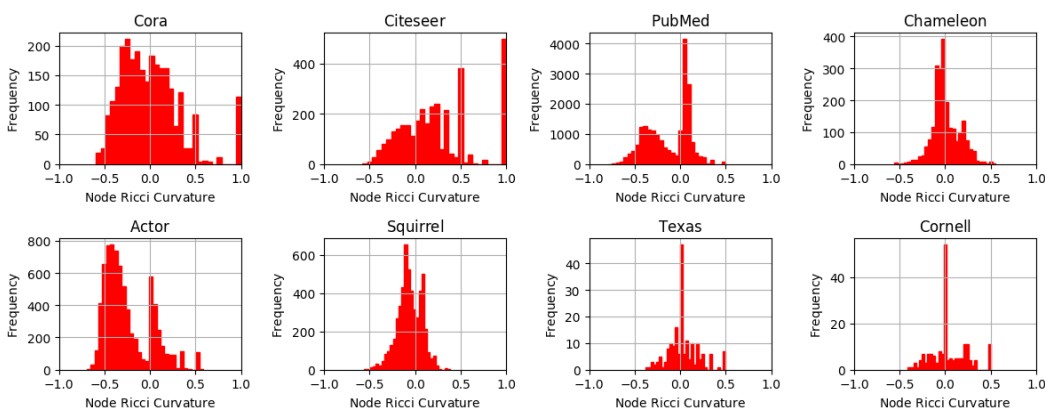

Figure 4: Distribution of Ollivier-Ricci curvatures $\widetilde{\kappa}(x, y)$ of **edges** across different datasets. The histograms illustrate the frequencies of edge-based Ollivier-Ricci curvature values for each dataset, *highlighting* the topological diversity in both homophilic and heterophilic settings, and hence the need of learning representations in manifolds with different curvatures.

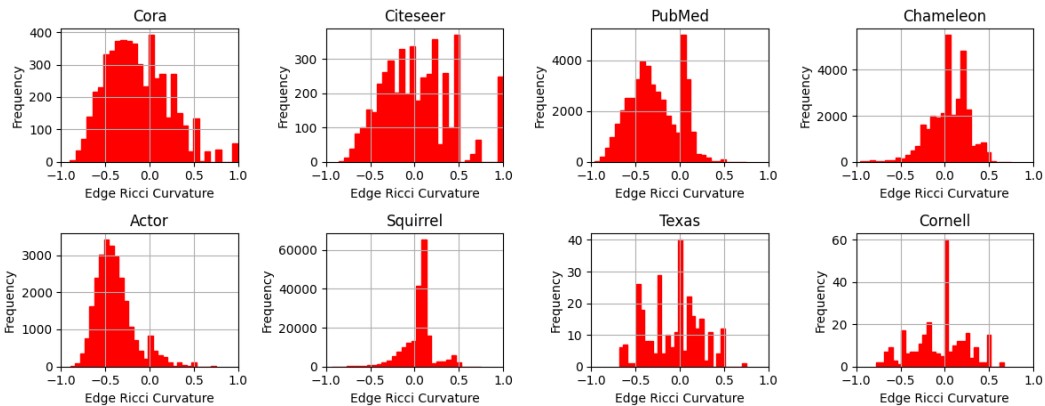

Figure 5: Distribution of Ollivier-Ricci curvatures $\widetilde{\kappa}(x)$ of **nodes** across different datasets.

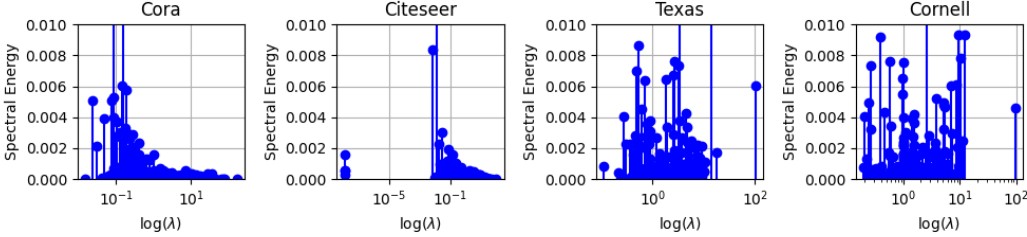

Figure 6: Distribution of **Spectral Energy** with the respective eigenvalues of the graph Laplacian. Spectral Energy, $\mathcal{E}_i = f_i^2 / \sum_j f_j^2$, where $f_j$ is the $j^{th}$ Fourier mode of the graph Laplacian. Low frequency implies *homophily* and high frequency components correspond to *heterophily*. These plots highlight the importance of capturing signals from different parts of the eigenspectrum for designing a GNN that works well across multiple tasks.

### 7.2.2 OLLIVIER-RICCI CURVATURE (ORC)

In an unweighted graph, the neighborhood of each node $x$, denoted as $\mathcal{N}(x)$, is assigned a probability distribution according to a lazy random walk formulation (Lin et al., 2011). Specifically, we define the distribution as follows:

$$m_z^\alpha(x) = \begin{cases} \alpha, & \text{if } z = x, \\ \frac{1-\alpha}{|\mathcal{N}(x)|}, & \text{if } z \in \mathcal{N}(x), \\ 0, & \text{otherwise.} \end{cases} \tag{9}$$

Here, $\alpha$ controls the probability that a random walk will remain at the current node, while the remaining probability mass $(1-\alpha)$ is uniformly distributed across the neighboring nodes. This formulation connects ORC with lazy random walks and influences the balance between local exploration and the likelihood of revisiting a node. In this work, we use $\alpha = 0.5$, meaning that equal probability mass is distributed between the node itself and its neighbors, striking a balance between *breadth-first* and *depth-first* search strategies. The choice of $\alpha$ is crucial and depends on the topology of the graph. A smaller $\alpha$ value encourages more local exploration, while a larger $\alpha$ favors revisiting nodes, thereby promoting a "lazy" walk. For our experiments, $\alpha = 0.5$ was chosen to reflect an equal probability mass distribution between the node and its neighbors.

■ **Computational Considerations.** Computing ORC can be computationally intensive due to the need to calculate the Wasserstein-1 distance ($W_1$), between the neighborhood distributions of connected nodes. In a discrete setting, this corresponds to solving a linear program. Typically, $W_1(m_x, m_y)$ between two nodes $x$ and $y$ is computed using the *Hungarian algorithm* (Kuhn, 1955), which has a cubic time complexity. However, this becomes prohibitively expensive as the graph size increases. Alternatively, the Wasserstein-1 distance can be approximated using the *Sinkhorn algorithm* (Sinkhorn & Knopp, 1967), which reduces the complexity to quadratic time. For this work, we employ the Sinkhorn approximation to compute ORC efficiently. Below, we provide an alternative to approximate ORC of an edge in linear time, in case of very large (million-scale) real-world graphs.

■ **Approximating ORC in Linear Time**. Even with the quadratic complexity of the Sinkhorn algorithm, scaling to large networks remains challenging. To address this, a linear-time combinatorial approximation of ORC can be employed, as suggested by Tian et al. (2023). This method approximates the Wasserstein distance by utilizing local structural information, making it much more computationally feasible. The approximation of ORC builds on classical bounds first introduced by Jost & Liu (2014). Let $\#(x,y)$ denote the number of triangles formed by the edge $(x,y)$, and define $a \wedge b = \min(a,b)$, $a \vee b = \max(a,b)$ and $d_x$ is the degree of node $x$. The following bounds on ORC can be derived for an edge $e = x, y$:

**Theorem 3** ( Jost & Liu (2014)). *For an unweighted graph, the Ollivier-Ricci curvature of an edge $e = x, y$ satisfies the following bounds:*

1. *Lower bound:*

$$\widetilde{\kappa}(x,y) \geq -\left(1 - \frac{1}{d_x} - \frac{1}{d_y} - \frac{\#(x,y)}{d_x \wedge d_y}\right)_+ - \left(1 - \frac{1}{d_x} - \frac{1}{d_y} - \frac{\#(x,y)}{d_x \vee d_y}\right)_+ + \frac{\#(x,y)}{d_x \vee d_y}.$$

2. *Upper bound:*

$$\widetilde{\kappa}(x,y) \leq \frac{\#(x,y)}{d_x \vee d_y}. \tag{10}$$

The ORC of an edge, can then be approximated as the arithmetic mean of these bounds:

$$\widehat{\kappa}(x,y) := \frac{1}{2}\left(\kappa^{\text{upper}}(x,y) + \kappa^{\text{lower}}(x,y)\right). \tag{11}$$

The proof of these bounds has been detailed in Tian et al. (2023). This approximation is computationally efficient, with linear-time complexity, and can be parallelized easily across edges, making it suitable for large-scale graphs. The computation relies solely on local structural information, such as the degree of the nodes and the number of triangles.

### 7.2.3 PRODUCT MANIFOLDS

Let $\mathcal{M}_1, \mathcal{M}_2, \ldots, \mathcal{M}_k$ denote a set of smooth manifolds. Their Cartesian product forms a product manifold, denoted by $\mathbb{P}$, such that $\mathbb{P} = \mathcal{M}_1 \times \mathcal{M}_2 \times \cdots \times \mathcal{M}_k$. Any point $p \in \mathbb{P}$ is characterized by its coordinates $p = (p_1, p_2, \ldots, p_k)$, where each $p_i$ corresponds to a point on the individual manifold $\mathcal{M}i$. Similarly, a tangent vector $v \in \mathcal{T}_p\mathbb{P}$ can be expressed as $v = (v_1, v_2, \ldots, v_k)$, where each $v_i \in \mathcal{T}_{p_i}\mathcal{M}_i$ represents the projection of $v$ in the tangent space of the respective component manifold $\mathcal{M}_i$. Optimization over manifolds requires the notion of taking steps along the manifold, which can be achieved by moving in the tangent space and mapping those movements back onto the manifold through the exponential map. The exponential map at a point $p \in \mathbb{P}$,

denoted $\exp_p : \mathcal{T}_p\mathbb{P} \to \mathbb{P}$, allows for this transfer. For product manifolds, the exponential map decomposes into individual component exponential maps. Specifically, given a tangent vector $v = (v_1, v_2, \ldots, v_k)$ at $p = (p_1, p_2, \ldots, p_k) \in \mathbb{P}$, the exponential map on $\mathbb{P}$ can be expressed as:

$$\exp_p(v) = (\exp_{p_1}(v_1), \exp_{p_2}(v_2), \ldots, \exp_{p_k}(v_k)) \tag{12}$$

### 7.2.4 KAPPA-STEREOGRAPHIC MODEL

The $\kappa$-stereographic model (Bachmann et al., 2020) unifies Hyperbolic and Spherical geometries under gyrovector formalism. This model leverages the framework of gyrovector spaces to represent all three constant curvature geometries—hyperbolic, Euclidean, and spherical—simultaneously. Additionally, it facilitates smooth transitions between these constant curvature geometries, enabling the joint learning of space curvatures alongside the embeddings. It is a smooth manifold $\mathcal{M}^{\kappa,d} = \{z \in \mathbb{R}^d| - \kappa||z||_2^2 < 1\}$, whose origin is $\mathbf{0} \in \mathbb{R}^d$, equipped with a Riemannian metric $g_z^\kappa = (\lambda_z^\kappa)^2\mathbf{I}$, where $\lambda_z^\kappa$ is given by $\lambda_z^\kappa = 2\left(1 + \kappa||z||_2^2\right)^{-1}$. The Riemannian operations under this model are elaborated in the table below:f

| Operation | Formalism in $\mathbb{E}^d$ | Unified formalism in $\kappa$-stereographic model ($\mathbb{H}^d$/$\mathbb{S}^d$) |
|---|---|---|
| Distance Metric | $d_\mathcal{M}^\kappa(\mathbf{x}, \mathbf{y}) = \|\mathbf{x} - \mathbf{y}\|_2$ | $d_\mathcal{M}^\kappa(\mathbf{x}, \mathbf{y}) = \frac{2}{\sqrt{|\kappa|}}\tan_\kappa^{-1}\left(\sqrt{|\kappa|}\ \|{-\mathbf{x}} \oplus_\kappa \mathbf{y}\|_2\right)$ |
| Exp. Map | $\exp_\mathbf{x}^\kappa(\mathbf{v}) = \mathbf{x} + \mathbf{v}$ | $\exp_\mathbf{x}^\kappa(\mathbf{v}) = \mathbf{x} \oplus_\kappa \left(\tan_\kappa\left(\sqrt{|\kappa|}\frac{\lambda_\mathbf{x}^\kappa\|\mathbf{v}\|_2}{2}\right)\frac{\mathbf{v}}{\sqrt{|\kappa|}\|\mathbf{v}\|_2}\right)$ |
| Log. Map | $\log_\mathbf{x}^\kappa(\mathbf{y}) = \mathbf{x} - \mathbf{y}$ | $\log_\mathbf{x}^\kappa(\mathbf{y}) = \frac{2}{\sqrt{|\kappa|}\lambda_\mathbf{x}^\kappa}\tan_\kappa^{-1}\left(\sqrt{|\kappa|}\ \|{-\mathbf{x}} \oplus_\kappa \mathbf{y}\|_2\right)\frac{-\mathbf{x}\oplus_\kappa\mathbf{y}}{\|{-\mathbf{x}}\oplus_\kappa\mathbf{y}\|_2}$ |
| Addition | $\mathbf{x} \oplus_\kappa \mathbf{y} = \mathbf{x} + \mathbf{y}$ | $\mathbf{x} \oplus_\kappa \mathbf{y} = \frac{\left(1+2\kappa\mathbf{x}^T\mathbf{y}+K\|\mathbf{y}\|^2\right)\mathbf{x}+\left(1-\kappa\|\mathbf{x}\|^2\right)\mathbf{y}}{1+2\kappa\mathbf{x}^T\mathbf{y}+\kappa^2\|\mathbf{x}\|^2\|\mathbf{v}\|^2}$ |

Table 7: Operations in Hyperbolic $\mathbb{H}^d$, Spherical $\mathbb{S}^d$ and Euclidean space $\mathbb{E}^d$.

■ $\kappa$-**right-matrix-multiplication**. Given a matrix $\mathbf{X} \in \mathbb{R}^{n \times d}$ holding $\kappa$-stereographic embeddings in its rows and weights $\mathbf{W} \in \mathbb{R}^{d \times e}$, the Euclidean right multiplication can be written row-wise as $(\mathbf{XW})_{i\bullet} = \mathbf{X}_{i\bullet}\mathbf{W}$. Then the $\kappa$-right-matrix-multiplication is defined row-wise as

$$(\mathbf{X} \otimes_\kappa \mathbf{W})_{i\bullet} = \exp_0^\kappa\left((\log_0^\kappa(\mathbf{X})\mathbf{W})_{i\bullet}\right) = \tan_\kappa\left(\alpha_i \tan_\kappa^{-1}(\|\mathbf{X}_{\bullet i}\|)\right)\frac{(\mathbf{XW})_{i\bullet}}{\|(\mathbf{XW})_{i\bullet}\|} \tag{13}$$

where $\alpha_i = \frac{\|(\mathbf{XW})_{i\bullet}\|}{\|\mathbf{X}_{i\bullet}\|}$ and $\exp_0^\kappa$ and $\log_0^\kappa$ denote the exponential and logarithmic map in the $\kappa$-stereo. model.

■ $\kappa$-**left-matrix-multiplication**. Given a matrix $\mathbf{X} \in \mathbb{R}^{n \times d}$ holding $\kappa$-stereographic embeddings in its rows and weights $\mathbf{A} \in \mathbb{R}^{n \times n}$, the $\kappa$-**left-matrix-multiplication** is defined row-wise as

$$(\mathbf{A} \boxtimes_\kappa \mathbf{X})_{i\bullet} := (\sum_j A_{ij}) \otimes_\kappa m_\kappa(\mathbf{X}_{1\bullet}, \cdots, \mathbf{X}_{n\bullet}; \mathbf{A}_{i\bullet}). \tag{14}$$

### 7.2.5 SPECTRAL GRAPH THEORY

Graph Fourier Transform (GFT) (Sandryhaila & Moura, 2013) lays the foundation for Graph Neural Networks (GNNs). A GFT is defined using a *reference* operator $\mathbf{R}$ which admits a spectral decomposition. Traditionally, in the case of GNNs, this reference operator has been the symmetric normalized Laplacian $\mathbf{L}_n = \mathbf{I} - \mathbf{A}_n$ (Kipf & Welling, 2016). The graph Fourier transform of a signal $\mathbf{f} \in \mathbb{R}^n$ is then defined as $\hat{\mathbf{f}} = \mathbf{U}^\top\mathbf{f} \in \mathbb{R}^n$, and its inverse as $\mathbf{f} = \mathbf{U}\hat{\mathbf{f}}$. A graph filter is an operator that acts independently on the entire eigenspace of a diagonalisable and symmetric reference operator R, by modulating their corresponding eigenvalues. A graph filter is defined via the graph filter function $g(.)$ operating on the reference operator as $g(\mathbf{R}) = \mathbf{U}g(\mathbf{\Lambda})\mathbf{U}^\top$.

### 7.3 MORE ON CUSP LAPLACIAN

Spectral graph theory has shown significant progress in relating geometric characteristics of graphs to properties of spectrum of graph Laplacians and related matrices. Several variants of the graph Laplacian matrices have been shown to capture specific inductive biases for different tasks (Ko et al., 2023; Belkin et al., 2008; Jacobson & Sorkine-Hornung, 2012).

*Proof of Definition 1.* Say the function $\psi : V \to \mathbb{R}$ is defined on the vertex set V of the graph. Suppose $\psi$ describes a temperature distribution across a graph, where $\psi(x)$ is the temperature at vertex $x$. According to Newton's law of cooling (He, 2024), the heat transferred from node $x$ to node $y$ is proportional to $\psi(x) - \psi(y)$

if nodes $x$ and $y$ are connected (if they are not connected, no heat is transferred). Consequently, the heat diffusion equation on the graph can be expressed as $\frac{d\psi}{dt} = -\beta \sum_y \mathbf{A}_{xy}(\psi(x) - \psi(y))$, where $\beta$ is a constant of proportionality and $\mathbf{A}$ denotes the adjacency matrix of the graph. Further insight can be gained by considering Fourier's law of thermal conductance (Liu, 1990), which states that heat flow is inversely proportional to the resistance to heat transfer. ORC measures the transportation cost ($\mathbf{W}_1(:,:)$) between the neighborhoods of two nodes, reflecting the effort required to transport mass between these neighborhoods (Bauer et al., 2011). We interpret this transportation cost as the *resistance* between nodes. The vital takeaway here is that $-$ *Heat flow between two nodes in a graph is influenced by the underlying Ollivier-Ricci curvature (ORC) distribution*. The diffusion rate is faster on an edge with positive curvature (low resistance), and slower on an edge with negative curvature (high resistance). Thus, the ratio $\mathcal{R}^{res}_{xy} = \frac{\mathbf{W}_1(m_x, m_y)}{d_{\mathcal{G}}(x,y)}$ represents the *resistance* from node $x$ to node $y$, i.e. $\frac{d\psi_{xy}}{dt} \propto \frac{1}{\mathcal{R}^{res}_{xy}}$. It can be observed that $\frac{1}{\mathcal{R}^{res}_{xy}} = \frac{d_{\mathcal{G}}(x,y)}{\mathbf{W}_1(m_x, m_y)} = \frac{1}{1 - \widetilde{\kappa}(x,y)}$ (From the definition of ORC) would tend to infinity when $\mathbf{W}_1(m_x, m_y) = 0$ (i.e. $\widetilde{\kappa}(x,y) = 1$). Thus, to ensure continuity, we create a new ratio as $\frac{1}{\mathcal{R}^{\bullet}_{xy}} = e^{-\mathcal{R}^{res}_{xy}} = e^{\frac{-1}{1-\widetilde{\kappa}(x,y)}}$. Thus, we can modify the above heat flow equation as:

$$\frac{d\psi}{dt} = -\bar{\beta} \sum_y \frac{\mathbf{A}_{xy}(\psi(x) - \psi(y))}{\mathcal{R}^{\bullet}_{xy}} \quad \text{(Inversely proportional to } \mathcal{R}^{\bullet}_{xy})$$

$$= -\bar{\beta} \sum_y \mathbf{A}_{xy}(\psi(x) - \psi(y)) \, e^{\frac{-1}{1-\widetilde{\kappa}(x,y)}} = -\bar{\beta} \left( \psi(x) \sum_y \mathbf{A}_{xy} - \sum_y \mathbf{A}_{xy}\psi(y) \right) e^{\frac{-1}{1-\widetilde{\kappa}(x,y)}}$$

$$= -\bar{\beta} \left( \psi(i)\mathbf{D}_{xx} - \sum_y \psi(y)\mathbf{A}_{xy} \right) e^{\frac{-1}{1-\widetilde{\kappa}(x,y)}} \quad (\because \mathbf{D}_{xx} = \sum_y \mathbf{A}_{xy})$$

$$= -\bar{\beta} \sum_y \left( \delta_{xy} \, e^{\frac{-1}{1-\widetilde{\kappa}(x,y)}} \mathbf{D}_{xx} - e^{\frac{-1}{1-\widetilde{\kappa}(x,y)}} \mathbf{A}_{xy} \right) \psi(y) \quad (\delta_{xy} \text{ is the Kronecker delta.})$$

$$= -\bar{\beta} \sum_y \left( e^{\frac{-1}{1-\widetilde{\kappa}(x,y)}} \mathbf{L}_{xy} \right) \psi(y) \quad (\mathbf{L}, \mathbf{D}, \mathbf{A} \text{ are Laplacian, Degree and Adjacency matrices.})$$

$$= -\bar{\beta} \left( \mathbf{L} \odot e^{\frac{-1}{1-\widetilde{\kappa}(x,y)}} \right) \psi = -\bar{\beta}\widetilde{\mathbf{L}}\psi \quad (\odot \text{ is the element-wise product})$$

This gives us the standard heat equation on graphs. Here, $\bar{\beta}$ is the updated constant of proportionality. $\widetilde{\mathbf{L}} = \widetilde{\mathbf{D}} - \widetilde{\mathbf{A}}$ is the **Cusp Laplacian** operator, where $\widetilde{\mathbf{D}}$ and $\widetilde{\mathbf{A}}$ are the updated Degree and Adjacency matrices, to represent that the graph is transformed under edge weights $w_{xy} = \frac{1}{\mathcal{R}^{\bullet}_{xy}} = e^{\frac{1}{1-\widetilde{\kappa}(x,y)}}$. Finally, our Cusp Laplacian operator can be written as ($x \sim y$ means $xy$ is an edge in the graph):

$$\widetilde{\mathbf{L}}\psi(x) = \sum_{y \sim x} \bar{w}_{xy} (\psi(x) - \psi(y)) = \sum_{y \sim x} e^{\frac{-1}{1-\widetilde{\kappa}(x,y)}} (\psi(x) - \psi(y)) \tag{15}$$

■ **Why is** $e^{-\mathcal{R}^{res}_{xy}} = e^{\frac{-1}{1-\widetilde{\kappa}(x,y)}}$ **the right choice?** To mathematically justify that $e^{-\frac{1}{1-\widetilde{\kappa}(x,y)}}$ is an appropriate choice, we must verify its properties:

1. **Asymptotics**. As $\widetilde{\kappa}(x,y) \to 1$, $e^{-\frac{1}{1-\widetilde{\kappa}(x,y)}} \to 0$, indicating that nodes with high positive curvature experience very fast heat diffusion (minimal resistance). Conversely, as $\widetilde{\kappa}(x,y) \to -1$, $e^{-\frac{1}{1-\widetilde{\kappa}(x,y)}} \to \frac{1}{\sqrt{e}}$, meaning that nodes with high negative curvature have slow heat diffusion (higher resistance).

2. **Continuity**. The exponential function is smooth and continuous, ensuring that even small changes in the curvature result in smooth changes in the heat flow dynamics, which is crucial for stable numerical simulations and theoretical consistency.

3. **Monotonicity**. For $\widetilde{\kappa}(x,y) > 0$, $e^{-\frac{1}{1-\widetilde{\kappa}(x,y)}}$ is a decreasing function with respect to $\widetilde{\kappa}(x,y)$. This means as curvature increases, the resistance decreases, aligning with the physical intuition of heat flow.

$\square$

### 7.3.1 RELEVENT THEOREMS FOR CUSP LAPLACIAN

**Theorem 4** (Positive Semidefiniteness of Cusp Laplacian). *The normalized Laplacian operator* $\widetilde{\mathbf{L}}$ *is positive semidefinite, i.e., for any real vector* $u \in \mathbb{R}^n$*, we have* $\mathbf{u}^T \widetilde{\mathbf{L}}_n \mathbf{u} \geq 0$.

*Proof.* We start by showing that the normalized **Cusp Laplacian**

$$\widetilde{\mathbf{L}}_{\mathrm{n}} = \mathbf{I} - \widetilde{\mathbf{D}}^{-1/2}\widetilde{\mathbf{A}}\widetilde{\mathbf{D}}^{-1/2} = \mathbf{I} - \widetilde{\mathbf{A}}_n \tag{16}$$

is positive semi-definite. Let $\mathbf{u}$ be any real vector of unit norm and $\mathbf{f} = \widetilde{\mathbf{D}}^{-1/2}\mathbf{u}$, then we have

$$\mathbf{u}^T\widetilde{\mathbf{L}}_{\mathrm{n}}\mathbf{u} = \mathbf{u}^T\mathbf{u} - \mathbf{u}^T\widetilde{\mathbf{D}}^{-1/2}\widetilde{\mathbf{A}}\widetilde{\mathbf{D}}^{-1/2}\mathbf{u} = \sum_{x=1}^{n}\mathbf{u}_x^2 - \sum_{x,y=1}^{n}\mathbf{f}_x\mathbf{f}_y\widetilde{\mathbf{A}}_{xy} \tag{17}$$

$$= \sum_{x=1}^{n}\widetilde{\mathbf{D}}_{xx}\mathbf{f}_x^2 - \sum_{x,y=1}^{n}\mathbf{f}_x\mathbf{f}_y\widetilde{\mathbf{A}}_{xy} = \frac{1}{2}\left(\sum_{x=1}^{n}\widetilde{\mathbf{D}}_{xx}\mathbf{f}_x^2 - 2\sum_{x,y=1}^{n}\mathbf{f}_x\mathbf{f}_y\widetilde{\mathbf{A}}_{xy} + \sum_{y=1}^{n}\widetilde{\mathbf{D}}_{yy}\mathbf{f}_y^2\right) \tag{18}$$

$$= \frac{1}{2}\sum_{x,y=1}^{n}\widetilde{\mathbf{A}}_{xy}(\mathbf{f}_x - \mathbf{f}_y)^2 = \frac{1}{2}\sum_{x,y=1}^{n}e^{\frac{-1}{1-\widetilde{\kappa}(x,y)}}\mathbf{A}_{xy}(\mathbf{f}_x - \mathbf{f}_y)^2, \tag{19}$$

where the last step follows from the definition of the degree. We know that $e^{\frac{-1}{1-\widetilde{\kappa}(x,y)}} > 0 \ \forall \kappa(x,y)$, hence our Cusp Laplacian is positive semidefinite. $\qquad\square$

**Theorem 5.** *The eigenvalues $\{\widetilde{\lambda}_i\}_{i=1}^{n}$ of the normalized Cusp Laplacian $\widetilde{\mathbf{L}}_n$ lie in the interval $[0,2]$.*

*Proof.* We begin by noting that Theorem 4 shows that the normalized **Cusp Laplacian** $\widetilde{\mathbf{L}}_{\mathrm{n}}$ has real, non-negative eigenvalues, meaning we need only to prove that the largest eigenvalue, denoted as $\lambda_n$, is less than or equal to 2. Before moving to that, we show that $0$ is indeed an eigenvalue of $\widetilde{\mathbf{L}}$ associated with the unit eigenvector $\boldsymbol{\tau}$ where $\boldsymbol{\tau} = \frac{\sqrt{\widetilde{\mathbf{D}}_{ii}}}{\sqrt{\sum_v \widetilde{\mathbf{D}}_{vv}}}$.

Let $\mathbf{1}$ be the all one vector. Then, a direct calculation reveals that

$$\widetilde{\mathbf{L}}_{\mathrm{sym}}\boldsymbol{\tau} = \boldsymbol{\tau} - \widetilde{\mathbf{D}}^{-1/2}\widetilde{\mathbf{A}}\widetilde{\mathbf{D}}^{-1/2}\boldsymbol{\tau} = \boldsymbol{\tau} - \widetilde{\mathbf{D}}^{-1/2}\widetilde{\mathbf{A}}\widetilde{\mathbf{D}}^{-1/2}\widetilde{\mathbf{D}}^{1/2}\mathbf{1} \times \frac{1}{\sqrt{\sum_v \widetilde{\mathbf{D}}_{vv}}} \tag{20}$$

$$= \boldsymbol{\tau} - \widetilde{\mathbf{D}}^{-1/2}\widetilde{\mathbf{A}}\mathbf{1} \times \frac{1}{\sqrt{\sum_v \widetilde{\mathbf{D}}_{vv}}} = \boldsymbol{\tau} - \widetilde{\mathbf{D}}^{-1/2}\widetilde{\mathbf{D}}\mathbf{1} \times \frac{1}{\sqrt{\sum_v \widetilde{\mathbf{D}}_{vv}}} \tag{21}$$

$$= \boldsymbol{\tau} - \widetilde{\mathbf{D}}^{1/2}\mathbf{1} \times \frac{1}{\sqrt{\sum_v \widetilde{\mathbf{D}}_{vv}}} = \boldsymbol{\tau} - \boldsymbol{\tau} = 0. \tag{22}$$

Combining this result with the positive semi-definite property of the Laplacian shows that $0$ is indeed the smallest eigenvalue of $\widetilde{\mathbf{L}}_{\mathrm{sym}}$ associated with the eigenvector $\boldsymbol{\tau}$. For the second part, using the Courant-Fischer theorem, we know that the largest eigenvalue can be expressed as:

$$\lambda_n = \max_{\mathbf{u}\neq 0}\frac{\mathbf{u}^\top\widetilde{\mathbf{L}}_{\mathrm{n}}\mathbf{u}}{\mathbf{u}^\top\mathbf{u}}.$$

Substituting the definition of the normalized Cusp Laplacian $\widetilde{\mathbf{L}}_{\mathrm{n}} = \mathbf{I} - \widetilde{\mathbf{A}}_{\mathrm{n}}$ into this expression, and letting $\mathbf{f} = \widetilde{\mathbf{D}}^{-1/2}\mathbf{u}$, we have:

$$\lambda_n = \max_{\mathbf{u}\neq 0}\frac{\mathbf{u}^\top\widetilde{\mathbf{D}}^{-1/2}\widetilde{\mathbf{L}}\widetilde{\mathbf{D}}^{-1/2}\mathbf{u}}{\mathbf{u}^\top\mathbf{u}} = \max_{\mathbf{f}\neq 0}\frac{\mathbf{f}^\top\widetilde{\mathbf{L}}\mathbf{f}}{\mathbf{f}^\top\widetilde{\mathbf{D}}\mathbf{f}}.$$

The degree matrix, can be expressed in the quadratic form as $\mathbf{f}^\top\widetilde{\mathbf{D}}\mathbf{f} = \sum_{x=1}^{n}\widetilde{\mathbf{D}}_{xx}|\mathbf{f}_x|^2$.

For the numerator involving $\widetilde{\mathbf{L}}$, we expand the quadratic form:

$$\mathbf{f}^\top\widetilde{\mathbf{L}}\mathbf{f} = \frac{1}{2}\sum_{x,y=1}^{n}\widetilde{\mathbf{A}}_{xy}(\mathbf{f}_x - \mathbf{f}_y)^2 = \frac{1}{2}\sum_{x,y=1}^{n}e^{\frac{-1}{1-\widetilde{\kappa}(x,y)}}\mathbf{A}_{xy}(\mathbf{f}_x - \mathbf{f}_y)^2 \tag{23}$$

$$\leq \sum_{x,y=1}^{n}e^{\frac{-1}{1-\widetilde{\kappa}(x,y)}}\mathbf{A}_{xy}(\mathbf{f}_x + \mathbf{f}_y)^2 \leq 2\sum_{x=1}^{n}|\mathbf{f}_x|^2\left(\sum_{y=1}^{n}\mathbf{A}_{xy}\right) = 2\sum_{x=1}^{n}\mathbf{D}_{xx}|\mathbf{f}(x)|^2. \tag{24}$$

The last inequality follows from the fact that $e^{\frac{-1}{1-\widetilde{\kappa}(x,y)}} \to \frac{1}{\sqrt{e}}$ as $\widetilde{\kappa}(x,y) \to -1$ implies that it is always $< 1$. Thus, we can conclude that, $\mathbf{f}^\top\widetilde{\mathbf{L}}\mathbf{f} \leq 2\mathbf{f}^\top\widetilde{\mathbf{D}}\mathbf{f}$, and the Rayleigh quotient is bounded $\frac{\mathbf{f}^\top\widetilde{\mathbf{L}}\mathbf{f}}{\mathbf{f}^\top\widetilde{\mathbf{D}}\mathbf{f}} \leq 2$. This shows that the largest eigenvalue of the normalized Cusp Laplacian $\widetilde{\mathbf{L}}_{\mathrm{n}}$ is bounded by 2, completing the proof that the eigenvalues of $\widetilde{\mathbf{L}}_{\mathrm{n}}$ are contained within the interval $[0,2]$. $\qquad\square$

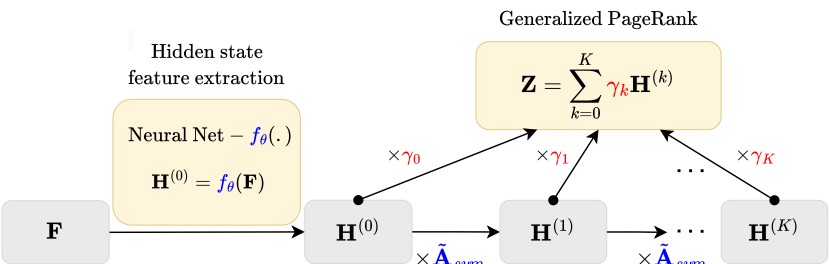

Figure 7: Architecture of GPRGNN.

## 7.4 GENERALISED PAGERANKS AND GPRGNN

■ **Equivalence of the GPR method and polynomial graph filtering.** If we truncate the infinite series in the GPR definition at some integer $K$, $\sum_{k=0}^{K} \gamma_k \widetilde{\mathbf{A}}_n^k$ becomes a polynomial graph filter of degree $K$. Consequently, optimizing the GPR weights is tantamount to optimizing the polynomial graph filter. It is important to note that any graph filter can be approximated using a polynomial graph filter, enabling the GPR method to handle a wide variety of node label patterns. Additionally, increasing $K$ enhances the approximation of the optimal graph filter. This again illustrates the advantage of large-step propagation.

■ **GPRGNN architecture.** GPR-GNN initially derives hidden state features for each node and subsequently employs GPR to disseminate them. The GPR-GNN procedure can be represented as:

$$\hat{\mathbf{P}} = \text{softmax}(\mathbf{Z}), \ \mathbf{Z} = \sum_{k=0}^{K} \gamma_k \mathbf{H}^{(k)}, \ \mathbf{H}^{(k)} = \widetilde{\mathbf{A}}_{\text{sym}} \mathbf{H}^{(k-1)}, \ \mathbf{H}_{i:}^{(0)} = f_\theta(\mathbf{X}_{i:}), \tag{25}$$

Here, $f_\theta(.)$ denotes a neural network parametrized by $\{\theta\}$, which produces the hidden state features $\mathbf{H}^{(0)}$. The GPR weights $\gamma_k$ are optimized alongside $\{\theta\}$ in an end-to-end manner. The GPR-GNN model is straightforward to interpret: As previously mentioned, GPR-GNN is capable of adaptively managing the contribution of each propagation step to fit the node label pattern. Analyzing the trained GPR weights also aids in understanding the topological properties of a graph, such as identifying the optimal polynomial graph filter.

### 7.4.1 PROOF OF THEOREM 1

We first state the formal version of Theorem 1

**Theorem 6** (Formal version of Theorem 1). *Assume the graph $G$ is connected. Let $\lambda_1 \geq \lambda_2 \geq ... \geq \lambda_n$ and $\widetilde{\lambda}_1 \leq \widetilde{\lambda}_2 \leq ... \leq \widetilde{\lambda}_n$ be the eigenvalues of $\widetilde{\mathbf{A}}_n$ and $\widetilde{\mathbf{L}}_n$, respectively. If $\gamma_l \geq 0 \ \forall l \in \{0, 1, ..., L\}$, $\sum_{l=0}^{L} \gamma_l = 1$ and $\exists l' > 0$ such that $\gamma_{l'} > 0$, then $\left| \frac{g_{\gamma,L}(\lambda_i)}{g_{\gamma,L}(\lambda_1)} \right| < 1 \ \forall i \geq 2$. Also, if $\gamma_l = (-\alpha)^l, \alpha \in (0, 1)$ and $L \rightarrow \infty$, then $\left| \frac{\lim_{L \rightarrow \infty} g_{\gamma,L}(\lambda_i)}{\lim_{L \rightarrow \infty} g_{\gamma,L}(\lambda_1)} \right| > 1 \ \forall i \geq 2$.*

1. Note that $\left| \frac{g_{\gamma,L}(\lambda_i)}{g_{\gamma,L}(\lambda_1)} \right| < 1 \ \forall i \geq 2$ implies the **low-pass case** as after applying the graph filter $g_{\gamma,L}$, the lowest frequency component (correspond to $\lambda_1$) further dominates.

2. **Unfiltered case**. Recall that in the unfiltered case, we do not multiply with $\widetilde{\mathbf{A}}_n$. It can also be viewed as multiplying the identity matrix $I$, where the eigenvalue ratio is $\frac{|\lambda_i|^0}{|\lambda_1|^0} = 1$. Hence $g_{\gamma,L}$ acts like a low pass filter in this case.

3. In contrast, $\left| \frac{\lim_{L \rightarrow \infty} g_{\gamma,L}(\lambda_i)}{\lim_{L \rightarrow \infty} g_{\gamma,L}(\lambda_1)} \right| > 1 \ \forall i \geq 2$ implies that after applying the graph filter, the lowest frequency component (correspond to $\lambda_1$) no longer dominates. This corresponds to the **high pass filter** case.

*Proof.* We start with the **low pass filter result**. From Theorem 5, we know that $0 \leq \widetilde{\lambda}_1 \leq \widetilde{\lambda}_2 \leq ... \leq \widetilde{\lambda}_n \leq 2$. Given the spectrum of $\widetilde{\mathbf{A}}_n$, we know that $-\widetilde{\mathbf{A}}_n$ has spectrum negatives of $\widetilde{\mathbf{A}}_n$, and $\mathbf{I} - \widetilde{\mathbf{A}}_n$ adds one to each eigenvalue of $-\widetilde{\mathbf{A}}_n$. Hence, $1 \geq \lambda_1 \geq \lambda_2 \geq ... \geq \lambda_n \geq -1$ follows directly. Now, we know that $\lambda_1 = 1$ and $|\lambda_i| < 1, \forall i \geq 2$. Further, we have assumed that $g_{\gamma,L}(\lambda_1) = \sum_{l=0}^{L} \gamma_l = 1$. Hence, proving Theorem 6 is equivalent to show

$$|g_{\gamma,L}(\lambda_i)| < 1 \ \forall i \geq 2.$$

This is obvious since $g_{\gamma,L}(\lambda) = \sum_{l=0}^{L} \gamma_l \lambda^l$ is a polynomial of order $L$ with nonnegative coefficients. It is easy to check that $\forall l \geq 1$, $|\lambda|^l < 1, \forall |\lambda| < 1$. Combine with the fact that all $\gamma_k$'s are nonnegative we have

$$|g_{\gamma,L}(\lambda_i)| \leq \sum_{l=0}^{L} \gamma_l |\lambda^l| = \sum_{l=0}^{L} \gamma_l |\lambda|^l \overset{\text{(a)}}{\leq} \sum_{l=0}^{L} \gamma_l = 1.$$

Finally, note that the only possibility that the equality holds is $\gamma_l = \delta_{0,l}$ since $\forall l \geq 1$, $|\lambda|^l < 1, \forall |\lambda| < 1$. However, by assumption $\sum_{l=0}^{L} \gamma_l = 1$ and $\exists l' > 0$ such that $\gamma_{l'} > 0$ we know that this is impossible. Hence (a) is a strict inequality $<$.

For the **high pass filter result**, it can be observed that:

$$\lim_{L \to \infty} g_{\gamma,L}(\lambda) = \lim_{L \to \infty} \sum_{l=0}^{L} \gamma_l \lambda^l = \lim_{L \to \infty} \sum_{l=0}^{L} (-\alpha\lambda)^l = \frac{1}{1 + \alpha\lambda},$$

where the last step is due to the fact that $\alpha \in (0,1)$ and thus $\lim_{L \to \infty}(-\alpha\lambda)^L = 0, \forall |\lambda| \leq 1$. Thus we have

$$\left| \frac{\lim_{L \to \infty} g_{\gamma,L}(\lambda_i)}{\lim_{L \to \infty} g_{\gamma,L}(\lambda_1)} \right| = \left| \frac{1 + \alpha}{1 + \alpha\lambda_i} \right| \overset{\text{(b)}}{>} 1 \ \forall i \geq 2.$$

The strict inequalities (b) is from the fact that $|\lambda_i| < 1, \forall i \geq 2$. Notably, $\sup_{\lambda \in [1,-1)} \frac{1}{1+\alpha\lambda}$ happens at the boundary $\lambda = -1$, which corresponds the the bipartite graph. It further shows that the graph filter with respect to the choice $\gamma_l = (-\alpha)^l$ emphasizes high frequency components and thus it is indeed acting as a high pass filter. □

### 7.4.2 WHY GPRGNN AS THE BACKBONE OF CUSP?

In this section, we elaborate on why GPR is an ideal backbone when compared to other Spectral GNNs.

1. **Adaptive Filter Design**. GPR learns the filter coefficients directly, allowing the spectral response to adapt to the task and dataset. This flexibility is critical for modeling both homophilic and heterophilic graphs.

2. **Universality**. Unlike fixed low-pass filters like ChebNet, which excel primarily in homophilic settings, GPR's learnable filters enable it to balance low-pass and high-pass components, making it suitable for both homophilic and heterophilic graphs. This is one of the main goals of our paper - to achieve superior performance on homophilic and heterophilic tasks. Fixed polynomial filters in ChebNet and Bernstein-based methods approximate spectral responses up to a fixed order, limiting their ability to model complex spectral properties.

3. **GPRGNN escapes oversmoothing**. GPR weights are adaptively learnable, which allows GPR-GNN to avoid over-smoothing and trade node and topology feature informativeness. See Section 4 of Chien et al. (2020) for more theoretical analysis on the same and proofs, which is beyond the scope of this work. GPR not only mitigates feature over-smoothing, but also works on highly diverse node label patterns (See Section 4 and 5 of Chien et al. (2020)).

4. **Capturing node features and graph topology**. In many important graph data processing applications, the acquired information includes both node features and observations of the graph topology. GPRGNN jointly optimizes node feature and topological information extraction, regardless of the extent to which the node labels are homophilic or heterophilic.

5. **Filter Bank Construction**. Using GPR based spectral filters, helps us to effectively construct a filter bank where each adaptive filter contributes to a specific spectral profile, enabling the model to aggregate information across different spectral bands. This approach captures diverse patterns in node features and topology, unlike ChebNet or Bernstein-based methods, which rely on fixed polynomial approximations and lack such flexibility.

### 7.5 THEOREMS FOR CURVATURE ENCODING

**Theorem 7** (Bochner's Theorem). *(Moeller et al., 2016) A continuous, translation-invariant kernel $\mathcal{K}(\mathbf{x}, \mathbf{y}) = \Psi(\mathbf{x} - \mathbf{y})$ on $\mathbb{R}^d$ is positive definite if and only if there exists a non-negative measure on $\mathbb{R}$ such that $\Psi$ is the Fourier transform of the measure.*

### 7.5.1 PROOF OF DEFINITION 2

**Proof.** Using the Bochner's theorem stated above, our curvature kernel $\mathcal{K}_{\mathbb{R}}$ has the expression:

$$\mathcal{K}_{\mathbb{R}}(\widetilde{\kappa}_a, \widetilde{\kappa}_b) = \Psi_{\mathbb{R}}(\widetilde{\kappa}_a, \widetilde{\kappa}_a) = \int_{\mathbb{R}} e^{i\omega(\widetilde{\kappa}_a - \widetilde{\kappa}_b)} p(\omega) d\omega = \mathbb{E}_{\omega}[\xi_{\omega}(\widetilde{\kappa}_a)\xi_{\omega}(\widetilde{\kappa}_b)^*], \tag{26}$$

where $\xi_\omega(\widetilde{\kappa}) = e^{i\omega\widetilde{\kappa}}$. Since kernel $\mathcal{K}_\mathbb{R}$ and measure $p(\omega)$ are real, we extract the real part of (26):

$$\mathcal{K}_\mathbb{R}(\widetilde{\kappa}_a, \widetilde{\kappa}_b) = \mathbb{E}_\omega\big[\cos(\omega(\widetilde{\kappa}_a - \widetilde{\kappa}_b))\big] = \mathbb{E}_\omega\big[\cos(\omega\widetilde{\kappa}_a)\cos(\omega\widetilde{\kappa}_b) + \sin(\omega\widetilde{\kappa}_a)\sin(\omega\widetilde{\kappa}_b)\big]. \qquad (27)$$

The above formulation suggests approximating the expectation by the Monte Carlo integral, i.e. $\mathcal{K}_\mathbb{R}(\widetilde{\kappa}_a, \widetilde{\kappa}_b) \approx \frac{1}{d_\mathcal{C}}\sum_{i=1}^{d_\mathcal{C}} \cos(\omega_i\widetilde{\kappa}_a)\cos(\omega_i\widetilde{\kappa}_b) + \sin(\omega_i\widetilde{\kappa}_a)\sin(\omega_i\widetilde{\kappa}_b)$, with $\omega_1, \ldots, \omega_{d_\mathcal{C}} \overset{\text{i.i.d}}{\sim} p(\omega)$. Therefore, we propose the finite dimensional functional mapping to $\mathbb{R}^{d_\mathcal{C}}$ as:

$$\Phi_{\mathbb{R}^{d_\mathcal{C}}}(\widetilde{\kappa}) = \sqrt{\frac{1}{d_\mathcal{C}}}\Big[\cos(\omega_1\widetilde{\kappa}), \sin(\omega_1\widetilde{\kappa}), \ldots, \cos(\omega_{d_\mathcal{C}}\widetilde{\kappa}), \sin(\omega_{d_\mathcal{C}}\widetilde{\kappa})\Big] \qquad (28)$$

The unknown probability distribution $p(\omega)$ is estimated using the inverse cumulative distribution function (CDF) transformation as in Xu et al. (2020). Since our GNN is operating in the mixed-curvature space, we must map our defined curvature kernel based representations to the product manifold. We do so using the exponential map, for a node $x$ with ORC curvature $\widetilde{\kappa}(x)$ as:

$$\Phi_{\mathbb{P}^{d_\mathcal{C}}}(\widetilde{\kappa}(x)) = g_\theta\Big(\|_{q=1}^{\mathcal{Q}}\mathbf{exp}_\mathbf{0}^{\kappa_{(q)}}(\Phi_{\mathbb{R}^{d_\mathcal{C}}}(\widetilde{\kappa}(x)))\Big) \qquad (29)$$

$$= g_\theta\Big(\|_{q=1}^{\mathcal{Q}}\Phi_{\mathcal{M}_q^{\kappa_{(q)},d_{(q)}}}(\widetilde{\kappa}(x))\Big) \qquad (30)$$

where $\mathbf{exp}_\mathbf{0}^{\kappa_{(q)}} : \mathbb{R}^{d_\mathcal{C}} \to \mathcal{M}_q^{\kappa_{(q)},d_{(q)}}$ denotes the exponential map on the $q^{th}$ component manifold with curvature $\kappa_{(q)}$, $\|$ is the concatenation operator and $g_\theta : \mathbb{P}^{d_f} \to \mathbb{P}^{d_\mathcal{C}}$ is a Riemannian projector. We need $g_\theta$ because we maintain a single product manifold for CUSP with total dimension $d_f$. So, upon taking the exponential map with respect to this product manifold, we are required to project the curvature embeddings to the required dimension $d_\mathcal{C}$. $\qquad\square$

### 7.5.2 PROOF OF THEOREM 2

**Proof.** We begin by recalling that in Euclidean space, the curvature kernel $\mathcal{K}_\mathbb{R}$ is:

$$\mathcal{K}_\mathbb{R}(\widetilde{\kappa}_a, \widetilde{\kappa}_b) = \big\langle \Phi_{\mathbb{R}^{d_\mathcal{C}}}(\widetilde{\kappa}_a), \Phi_{\mathbb{R}^{d_\mathcal{C}}}(\widetilde{\kappa}_b)\big\rangle = \Psi_\mathbb{R}(\widetilde{\kappa}_a - \widetilde{\kappa}_b).$$

The key property here is translation invariance:

$$\mathcal{K}_\mathbb{R}(\widetilde{\kappa}_a + \widetilde{c}, \widetilde{\kappa}_b + \widetilde{c}) = \mathcal{K}_\mathbb{R}(\widetilde{\kappa}_a, \widetilde{\kappa}_b) = \Psi_\mathbb{R}(\widetilde{\kappa}_a - \widetilde{\kappa}_b).$$

Next, we move to the product manifold $\mathbb{P}^{d_c}$, which consists of multiple components of different curvatures, such as hyperbolic, spherical, and Euclidean spaces.

For each component manifold $\mathcal{M}_q^{\kappa_{(q)},d_{(q)}}$ with curvature $\kappa_{(q)}$, the stereographic inner product $\langle .,.\rangle_\mathbf{x}^\kappa : \mathcal{T}_x\mathcal{M}_\kappa^n \times \mathcal{T}_x\mathcal{M}_\kappa^n \to \mathbb{R}$, is defined on the tangent plane of the Riemannian manifold as:

$$\langle \mathbf{u}, \mathbf{v}\rangle_\mathbf{x}^\kappa = \mathbf{u}^T\mathbf{g}_\mathbf{x}^\kappa\mathbf{v} = (\lambda_\mathbf{x}^\kappa)^2\langle \mathbf{u}, \mathbf{v}\rangle,$$

where the conformal factor $\lambda_\mathbf{x}^\kappa$ is defined as:

$$\lambda_\mathbf{x}^\kappa = \frac{2}{1 + \kappa\|\mathbf{x}\|_2^2}.$$

This conformal factor modulates the stereographic projection in the curved space, and it ensures that distances are mapped correctly in the manifold space. Now, consider the inner product between the curvature embeddings $\Phi_{\mathbb{P}^{d_\mathcal{C}}}(\widetilde{\kappa}_a)$ and $\Phi_{\mathbb{P}^{d_\mathcal{C}}}(\widetilde{\kappa}_b)$ in the mixed-curvature space.

$$\mathcal{K}_\mathbb{P}(\widetilde{\kappa}_a, \widetilde{\kappa}_b) = \sum_{q=1}^{\mathcal{Q}} \langle \Phi_{\mathcal{M}_q^{\kappa_{(q)},d_{(q)}}}(\widetilde{\kappa}_a), \Phi_{\mathcal{M}_q^{\kappa_{(q)},d_{(q)}}}(\widetilde{\kappa}_b)\rangle_{\kappa_{(q)}},$$

where each component manifold $\mathcal{M}_q^{\kappa_{(q)},d_{(q)}}$ contributes to the overall inner product in the product manifold $\mathbb{P}^{d_\mathcal{C}}$. Using the stereographic inner product in each component, we can write:

$$\mathcal{K}_\mathbb{P}(\widetilde{\kappa}_a, \widetilde{\kappa}_b) = \sum_{q=1}^{\mathcal{Q}} \left(\lambda_\mathbf{x}^{\kappa_{(q)}}\right)^2\langle \Phi_{\mathbb{R}^{d_\mathcal{C}}}(\widetilde{\kappa}_a), \Phi_{\mathbb{R}^{d_\mathcal{C}}}(\widetilde{\kappa}_b)\rangle.$$

We now need to show that translation invariance holds in the mixed-curvature product manifold. Since the conformal factor $\lambda_\mathbf{x}^\kappa$ depends only on the norm $\|\mathbf{x}\|_2$, any translation by a constant $\widetilde{c}$ does not affect the relative difference between curvature embeddings. Specifically, for any constant shift $\widetilde{c}$:

$$\mathcal{K}_\mathbb{P}(\widetilde{\kappa}_a + \widetilde{c}, \widetilde{\kappa}_b + \widetilde{c}) = \sum_{q=1}^{\mathcal{Q}} \left(\lambda_\mathbf{x}^{\kappa_{(q)}}\right)^2\Psi_\mathbb{R}((\widetilde{\kappa}_a + \widetilde{c}) - (\widetilde{\kappa}_b + \widetilde{c})) = \Psi_\mathbb{P}(\widetilde{\kappa}_a - \widetilde{\kappa}_b).$$

Thus, the kernel in the mixed-curvature space remains invariant to translation, completing the proof. $\qquad\square$

### 7.6 EXPERIMENTATION

#### 7.6.1 DATASETS

The performance of CUSP is evaluated over eight benchmark datasets for two primary tasks: Node Classification (NC) and Link Prediction (LP). These datasets encompass both homophilic and heterophilic domains. Detailed descriptions of each dataset are provided below.

1. *Citation Networks*. **Cora**, **PubMed**, and **Citeseer** (Sen et al., 2008; Yang et al., 2016) are citation networks in which nodes symbolize research papers, and edges denote citation links between them. Each node is labeled with its subject category. This dataset is commonly utilized for node classification because of its pronounced *homophilic* properties.

2. *Wikipedia graphs*. **Chameleon** and **Squirrel** (Rozemberczki et al., 2021) are *heterophilic* graphs derived from Wikipedia articles. Nodes represent articles, and edges represent hyperlinks between them. Node labels correspond to website traffic levels.

3. *Actor Co-occurrence Network*. **Actor** (Tang et al., 2009) is a heterophilic graph dataset where nodes depict actors and edges signify co-occurrences on the same Wikipedia page. The node labels correspond to the professional background of the actors.

4. *Webpage graphs*. **Texas** and **Cornell**[4] are parts of the WebKB dataset, and are sparsely connected heterophilic graphs. Here, nodes represent web pages, and edges represent hyperlinks between them. Labels reflect different types of webpages.

#### 7.6.2 BASELINES

This part offers an in-depth discussion of the baseline models against which CUSP is compared. We classify the baseline models into three categories: **Spatial**, **Riemannian**, and **Spectral** methods, which each address a distinct facet of graph neural network architectures.

■ **Spatial baselines**. The first kind of baselines includes the traditional spatial methods, which directly operate on the node features and their immediate neighborhoods.

1. **GCN** (Kipf & Welling, 2016). Graph Convolutional Networks (GCNs) represent one of the foundational graph neural networks that utilize spectral graph convolution in the spatial domain. They derive node embeddings by combining features from neighboring nodes via a linear combination involving the adjacency matrix and the nodes' features.

2. **GAT** (Veličković et al., 2017). Graph Attention Network (GAT) introduces attention mechanisms to graph neural networks. Each node assigns learnable attention weights to its neighbors and aggregates their features based on these weights.

3. **GraphSAGE** (Hamilton et al., 2017). GraphSAGE is an inductive technique designed to learn node embeddings by sampling and aggregating features from a *fixed* set of neighboring nodes, instead of processing the entire graph. This method enables GraphSAGE to create embeddings for nodes not encountered during training by using efficient neighborhood sampling.

■ **Riemannian Baselines**. Riemannian models function within non-Euclidean spaces (such as hyperbolic or spherical manifolds) and are tailored for graph data characterized by intricate geometric properties (e.g. hierarchical or cyclic structures).

1. **HGCN** (Chami et al., 2019). Hyperbolic Graph Convolutional Networks utilize *hyperbolic geometry* to represent the hierarchical and tree-like characteristics of graphs. This approach extends GCN to hyperbolic space by introducing a hyperbolic variant of the convolutional operation. It is especially suitable for datasets that exhibit hierarchical or tree-like configurations.

2. **HGAT** (Zhang et al., 2021b). Hyperbolic Graph Attention Network (HGAT) extends Graph Attention Networks (GAT) into hyperbolic space by integrating attention mechanisms with hyperbolic geometry, and calculates the attention weights among the nodes in the hyperbolic space to enhance the aggregation of features.

3. $\kappa$**GCN** (Bachmann et al., 2020). $\kappa$GCN allows for learning the curvature of each node in a graph and generalizes GCN to operate in mixed-curvature spaces. The curvature parameter $\kappa$ determines whether a node lies in hyperbolic, spherical, or Euclidean space. By learning curvature adaptively, $\kappa$GCN offers flexibility in modeling graphs with regions of different geometries, providing a better fit for graphs with complex structures.

---

[4]http://www.cs.cmu.edu/afs/cs.cmu.edu/project/theo-11/www/wwkb

4. $\mathcal{Q}$**GCN** (Xiong et al., 2022). Pseudo-Riemannian GCN extends a GCN to a pseudo-Riemannian manifold, enabling functionality in mixed-curvature spaces. This network is capable of modeling graph regions with both positive and negative curvature.

5. **SelfMGNN** (Sun et al., 2022). SelfMGNN generates embeddings within a mixed-curvature space through self-supervision. It dynamically allocates varied curvatures to different regions of the graph by utilizing a mixed-curvature embedding space. This approach incorporates both self-supervision and mixed-curvature learning to improve performance on heterogeneous graphs.

■ **Spectral Baselines**. These techniques utilize the eigenvalues of either the graph Laplacian or adjacency matrix to establish convolutional filters that function in the frequency domain.

1. **ChebyNet** (Defferrard et al., 2016). ChebyNet implements spectral convolutions through a polynomial approximation of the graph Laplacian, sidestepping the expensive process of eigenvalue decomposition. Instead, it approximates the convolution using Chebyshev polynomials. This approach allows ChebyNet to execute localized graph convolutions efficiently, making it well-suited for handling larger graphs.

2. **BernNet** (He et al., 2021). BernNet employs Bernstein polynomials to approximate graph filters, providing flexible management over the filter's frequency response. This method extends polynomial-based graph filters and is adaptable to various frequency elements in graphs.

3. **GPRGNN** (Chien et al., 2020). The Generalized PageRank Graph Neural Network (GPRGNN) builds upon the Personalized PageRank (PPR) approach, integrating it into the framework of graph neural networks. It propagates node features through the graph by using a weighted sum of adjacency matrix powers, dynamically adjusting to both homophilic and heterophilic graphs.

4. **FiGURe** (Ekbote et al., 2024). FiGURe employs adaptive filters to capture various sections of the graph spectrum, enabling it to learn both high-pass and low-pass filters specific to the task. It dynamically selects the optimal filter bank to accurately represent the graph's architecture.

### 7.6.3 EXPERIMENTAL RESULTS FOR LINK PREDICTION

| Baseline | Cora | Citeseer | PubMed | Chameleon | Actor | Squirrel | Texas | Cornell | Av. Δ Gain |
|---|---|---|---|---|---|---|---|---|---|
| GCN | $88.54_{\pm 0.51}$ | $85.42_{\pm 0.89}$ | $91.31_{\pm 0.73}$ | $86.07_{\pm 0.64}$ | $85.12_{\pm 0.78}$ | $90.01_{\pm 0.15}$ | $69.08_{\pm 0.99}$ | $73.09_{\pm 0.92}$ | 9.58 |
| GAT | $85.45_{\pm 0.66}$ | $87.23_{\pm 0.11}$ | $87.65_{\pm 0.04}$ | $88.99_{\pm 0.13}$ | $87.33_{\pm 0.08}$ | $90.23_{\pm 0.14}$ | $68.79_{\pm 0.72}$ | $75.12_{\pm 0.77}$ | 9.31 |
| SAGE | $87.12_{\pm 0.82}$ | $90.71_{\pm 0.65}$ | $90.09_{\pm 0.90}$ | $90.01_{\pm 0.58}$ | $86.06_{\pm 0.73}$ | $91.02_{\pm 0.61}$ | $76.54_{\pm 0.69}$ | $77.98_{\pm 0.88}$ | 6.97 |
| HGCN | $91.63_{\pm 0.55}$ | $94.13_{\pm 0.67}$ | $91.04_{\pm 0.79}$ | $91.45_{\pm 0.62}$ | $90.01_{\pm 0.80}$ | $92.34_{\pm 0.01}$ | $69.99_{\pm 0.84}$ | $74.03_{\pm 0.57}$ | 6.34 |
| HGAT | $90.43_{\pm 0.03}$ | $91.02_{\pm 0.16}$ | $88.99_{\pm 0.89}$ | $89.77_{\pm 0.02}$ | $90.99_{\pm 0.01}$ | $89.22_{\pm 0.04}$ | $71.58_{\pm 0.89}$ | $72.03_{\pm 0.22}$ | 7.66 |
| $\kappa$GCN | $92.04_{\pm 0.70}$ | $93.33_{\pm 0.57}$ | $92.45_{\pm 0.85}$ | $92.03_{\pm 0.63}$ | $90.45_{\pm 0.88}$ | $91.35_{\pm 0.60}$ | $76.09_{\pm 0.76}$ | $73.05_{\pm 0.71}$ | 5.56 |
| $\mathcal{Q}$GCN | $92.17_{\pm 0.79}$ | $92.75_{\pm 0.52}$ | $92.16_{\pm 0.09}$ | $91.67_{\pm 0.05}$ | $91.07_{\pm 0.06}$ | $90.98_{\pm 0.92}$ | $75.44_{\pm 0.10}$ | $73.89_{\pm 0.26}$ | 5.65 |
| SelfMGNN | $93.12_{\pm 0.04}$ | $92.99_{\pm 0.91}$ | $90.99_{\pm 0.17}$ | $93.51_{\pm 0.14}$ | $91.98_{\pm 0.19}$ | $95.01_{\pm 0.16}$ | $74.51_{\pm 0.62}$ | $78.99_{\pm 0.81}$ | 4.28 |
| ChebyNet | $88.23_{\pm 0.85}$ | $89.22_{\pm 0.06}$ | $86.54_{\pm 0.29}$ | $90.01_{\pm 0.23}$ | $88.09_{\pm 0.44}$ | $92.13_{\pm 0.57}$ | $73.45_{\pm 0.01}$ | $79.01_{\pm 0.18}$ | 7.33 |
| BernNet | $86.34_{\pm 0.13}$ | $87.09_{\pm 0.60}$ | $85.34_{\pm 0.82}$ | $87.15_{\pm 0.37}$ | $87.22_{\pm 0.15}$ | $91.22_{\pm 0.55}$ | $77.65_{\pm 0.87}$ | $78.34_{\pm 0.19}$ | 8.12 |
| GPRGNN | $91.16_{\pm 0.72}$ | $93.05_{\pm 0.81}$ | $92.03_{\pm 0.01}$ | $91.22_{\pm 0.16}$ | $89.76_{\pm 0.62}$ | $92.34_{\pm 0.23}$ | $76.05_{\pm 0.18}$ | $80.04_{\pm 0.12}$ | 4.96 |
| FiGURe | $91.98_{\pm 0.69}$ | $94.33_{\pm 0.15}$ | $92.67_{\pm 0.83}$ | $93.09_{\pm 0.31}$ | $90.11_{\pm 0.29}$ | $95.43_{\pm 0.65}$ | $76.99_{\pm 0.16}$ | $80.12_{\pm 0.58}$ | 3.82 |
| CUSP | $\mathbf{95.08_{\pm 0.13}}$ | $\mathbf{96.88_{\pm 0.65}}$ | $\mathbf{96.01_{\pm 0.01}}$ | $\mathbf{97.66_{\pm 0.33}}$ | $\mathbf{96.04_{\pm 0.38}}$ | $\mathbf{97.17_{\pm 0.11}}$ | $\mathbf{81.23_{\pm 0.14}}$ | $\mathbf{85.23_{\pm 0.05}}$ | 0 |
| Imp. Δ | 1.96 | 2.55 | 3.34 | 4.15 | 4.06 | 1.74 | 3.58 | 5.11 | |

Table 8: Performance comparision of CUSP with baselines for LP task (Mean AUC Score $\pm$ 95% confidence interval). **First**, Second and Third best performing models are highlighted.

| Dataset | CUSP | $\text{CUSP}_{euc}$ | $\text{CUSP}_{lap}$ | $\text{CUSP}_{enc}$ | $\text{CUSP}_{pool}$ | $\text{CUSP}_{fil}$ |
|---|---|---|---|---|---|---|
| Cora | $\mathbf{95.08_{\pm 0.13}}$ | $92.45_{\pm 0.25}$ | $93.21_{\pm 0.42}$ | $93.78_{\pm 0.33}$ | $92.13_{\pm 0.40}$ | $93.02_{\pm 0.27}$ |
| Citeseer | $\mathbf{96.88_{\pm 0.65}}$ | $94.03_{\pm 0.30}$ | $95.34_{\pm 0.22}$ | $94.08_{\pm 0.29}$ | $94.01_{\pm 0.31}$ | $94.56_{\pm 0.26}$ |
| PubMed | $\mathbf{96.01_{\pm 0.01}}$ | $93.52_{\pm 0.60}$ | $94.81_{\pm 0.40}$ | $94.92_{\pm 0.38}$ | $94.11_{\pm 0.35}$ | $94.16_{\pm 0.39}$ |
| Chameleon | $\mathbf{97.66_{\pm 0.33}}$ | $95.02_{\pm 0.44}$ | $96.23_{\pm 0.52}$ | $96.13_{\pm 0.28}$ | $95.13_{\pm 0.40}$ | $95.67_{\pm 0.57}$ |
| Actor | $\mathbf{96.04_{\pm 0.38}}$ | $91.45_{\pm 0.55}$ | $93.99_{\pm 0.32}$ | $92.81_{\pm 0.42}$ | $91.98_{\pm 0.47}$ | $92.13_{\pm 0.49}$ |
| Squirrel | $\mathbf{97.17_{\pm 0.11}}$ | $93.14_{\pm 0.22}$ | $95.56_{\pm 0.37}$ | $94.33_{\pm 0.43}$ | $93.75_{\pm 0.32}$ | $92.89_{\pm 0.14}$ |
| Texas | $\mathbf{81.23_{\pm 0.14}}$ | $78.45_{\pm 0.36}$ | $79.88_{\pm 0.52}$ | $80.12_{\pm 0.42}$ | $79.23_{\pm 0.37}$ | $77.81_{\pm 0.54}$ |
| Cornell | $\mathbf{85.23_{\pm 0.05}}$ | $82.89_{\pm 0.32}$ | $83.91_{\pm 0.39}$ | $84.23_{\pm 0.37}$ | $82.31_{\pm 0.48}$ | $82.45_{\pm 0.42}$ |
| Avg. Δ Gain | 0 | 3.0437 | 1.5462 | 1.8625 | 2.8312 | 2.8262 |

Table 9: Ablation study (LP) results. $\text{CUSP}_{euc}$ is the Euclidean variant, $\text{CUSP}_{lap}$ uses the traditional Laplacian, $\text{CUSP}_{enc}$ gets rid of curvature encoding, $\text{CUSP}_{pool}$ replaces Cusp pooling with concatenation, and $\text{CUSP}_{fil}$ uses a single filter instead of a filter bank. **Av. Δ Gain** represents the average gain of CUSP over the ablation model in that column, averaged across the different datasets.

| CUSP | Cora | Citeseer | PubMed | Chameleon | Actor | Squirrel | Texas | Cornell |
|---|---|---|---|---|---|---|---|---|
| $\mathbb{H}^{24} \times \mathbb{S}^{24}$ | $93.10_{\pm 0.22}$ | $94.33_{\pm 0.35}$ | $95.41_{\pm 0.25}$ | $97.28_{\pm 0.45}$ | $95.25_{\pm 0.44}$ | $96.05_{\pm 0.51}$ | $75.43_{\pm 0.31}$ | $76.95_{\pm 0.38}$ |
| $(\mathbb{H}^8)^2 \times (\mathbb{S}^8)^2 \times \mathbb{E}^{16}$ | $93.00_{\pm 0.27}$ | $95.11_{\pm 0.28}$ | $94.52_{\pm 0.34}$ | $96.66_{\pm 0.39}$ | $95.34_{\pm 0.38}$ | $95.40_{\pm 0.57}$ | $80.95_{\pm 0.42}$ | $84.80_{\pm 0.43}$ |
| $\mathbb{H}^8 \times \mathbb{S}^8 \times \mathbb{E}^{32}$ | $93.20_{\pm 0.24}$ | $93.25_{\pm 0.30}$ | $95.65_{\pm 0.30}$ | $97.11_{\pm 0.37}$ | $94.92_{\pm 0.40}$ | $95.51_{\pm 0.45}$ | $\mathbf{81.23_{\pm 0.14}}$ | $\mathbf{85.23_{\pm 0.05}}$ |
| $\mathbb{H}^{16} \times (\mathbb{S}^{16})^2$ | $93.50_{\pm 0.25}$ | $94.12_{\pm 0.33}$ | $95.32_{\pm 0.45}$ | $\mathbf{97.66_{\pm 0.33}}$ | $95.20_{\pm 0.34}$ | $\mathbf{97.17_{\pm 0.11}}$ | $80.30_{\pm 0.45}$ | $78.02_{\pm 0.47}$ |
| $(\mathbb{H}^{16})^2 \times \mathbb{E}^{16}$ | $93.25_{\pm 0.21}$ | $93.05_{\pm 0.32}$ | $\mathbf{96.01_{\pm 0.01}}$ | $96.80_{\pm 0.40}$ | $\mathbf{96.04_{\pm 0.38}}$ | $96.15_{\pm 0.55}$ | $79.50_{\pm 0.38}$ | $84.60_{\pm 0.40}$ |
| $\mathbb{H}^{24} \times \mathbb{E}^{24}$ | $92.70_{\pm 0.30}$ | $94.71_{\pm 0.35}$ | $95.45_{\pm 0.42}$ | $96.51_{\pm 0.47}$ | $95.00_{\pm 0.37}$ | $88.95_{\pm 0.53}$ | $79.70_{\pm 0.54}$ | $81.20_{\pm 0.60}$ |
| $\mathbb{S}^{24} \times \mathbb{E}^{24}$ | $88.50_{\pm 0.39}$ | $91.23_{\pm 0.35}$ | $89.99_{\pm 0.44}$ | $91.12_{\pm 0.43}$ | $79.90_{\pm 0.52}$ | $91.20_{\pm 0.51}$ | $79.08_{\pm 0.52}$ | $83.75_{\pm 0.48}$ |
| $\mathbb{H}^{16} \times \mathbb{S}^{16} \times \mathbb{E}^{16}$ | $\mathbf{95.08_{\pm 0.13}}$ | $\mathbf{96.88_{\pm 0.65}}$ | $95.55_{\pm 0.41}$ | $94.44_{\pm 0.34}$ | $95.55_{\pm 0.35}$ | $96.25_{\pm 0.38}$ | $81.07_{\pm 0.29}$ | $84.60_{\pm 0.30}$ |
| $(\mathbb{S}^8)^2 \times \mathbb{E}^{32}$ | $88.05_{\pm 0.40}$ | $90.35_{\pm 0.42}$ | $86.30_{\pm 0.48}$ | $89.34_{\pm 0.45}$ | $93.33_{\pm 0.54}$ | $90.25_{\pm 0.48}$ | $80.10_{\pm 0.43}$ | $82.01_{\pm 0.55}$ |
| $(\mathbb{H}^{16})^3$ | $89.10_{\pm 0.44}$ | $91.01_{\pm 0.50}$ | $93.93_{\pm 0.55}$ | $96.19_{\pm 0.50}$ | $95.25_{\pm 0.49}$ | $92.30_{\pm 0.52}$ | $77.77_{\pm 0.53}$ | $79.44_{\pm 0.58}$ |

Table 10: Performance comparison of CUSP with different manifold signatures for Link Prediction (LP). Best performing *signatures* are in **Bold**, and cases with a large decline in performance because of manifold mismatch are in Blue.

| Dataset | Signature |
|---|---|
| Cora | $\mathbb{H}^{16}(-0.21, 0.31) \times \mathbb{S}^{16}(+0.49, 0.38) \times \mathbb{E}^{16}(0, 0.31)$ |
| Citeseer | $\mathbb{H}^{16}(-0.78, 0.29) \times \mathbb{S}^{16}(+0.55, 0.39) \times \mathbb{E}^{16}(0, 0.32)$ |
| PubMed | $\mathbb{H}^{16}(-0.76, 0.56) \times \mathbb{H}^{16}(-0.28, 0.41) \times \mathbb{E}^{16}(0, 0.03)$ |
| Chameleon | $\mathbb{H}^{16}(-0.34, 0.09) \times \mathbb{S}^{16}(+0.71, 0.25) \times \mathbb{S}^{16}(+0.55, 0.34)$ |
| Actor | $\mathbb{H}^{16}(-0.77, 0.17) \times \mathbb{H}^{16}(-0.39, 0.42) \times \mathbb{E}^{16}(0, 0.41)$ |
| Squirrel | $\mathbb{H}^{16}(-0.17, 0.23) \times \mathbb{S}^{16}(+0.54, 0.38) \times \mathbb{S}^{16}(+0.38, 0.39)$ |
| Texas | $\mathbb{H}^8(-0.38, 0.31) \times \mathbb{S}^8(+0.18, 0.19) \times \mathbb{E}^{32}(0, 0.50)$ |
| Cornell | $\mathbb{H}^8(-0.41, 0.19) \times \mathbb{S}^8(+0.09, 0.26) \times \mathbb{E}^{32}(0, 0.55)$ |

Table 11: Learning results of CUSP on Link Prediction (LP) task for the best performing product signature. Format of entries − *manifold type* $^{(\text{dim})}$ (learnt curvature, learnt manifold weight).

| Dataset | I | $\mathbf{Z}^{(1)}$ | $\mathbf{Z}^{(2)}$ | $\mathbf{Z}^{(3)}$ | $\mathbf{Z}^{(4)}$ | $\mathbf{Z}^{(5)}$ | $\mathbf{Z}^{(6)}$ | $\mathbf{Z}^{(7)}$ | $\mathbf{Z}^{(8)}$ | $\mathbf{Z}^{(9)}$ | $\mathbf{Z}^{(10)}$ |
|---|---|---|---|---|---|---|---|---|---|---|---|
| Cora | **0.1125** | 0.2393 | 0.0520 | 0.0504 | 0.0676 | 0.0272 | 0.1531 | 0.1251 | 0.0939 | 0.0095 | 0.0694 |
| Citeseer | **0.2131** | 0.0150 | 0.2195 | 0.0283 | 0.0922 | 0.1530 | 0.0350 | 0.0320 | 0.0282 | 0.1254 | 0.0582 |
| PubMed | 0.0279 | 0.1793 | 0.0285 | 0.0296 | 0.0563 | **0.3870** | 0.0603 | 0.0324 | 0.0612 | 0.0271 | 0.1104 |
| Chameleon | 0.0601 | 0.1136 | 0.1694 | 0.0924 | **0.1329** | 0.1605 | 0.1026 | 0.0157 | 0.0475 | 0.0344 | 0.0709 |
| Actor | 0.1230 | 0.0321 | 0.0324 | 0.1147 | 0.1287 | **0.3700** | 0.0136 | 0.0389 | 0.0604 | 0.0691 | 0.0172 |
| Squirrel | 0.0182 | 0.0289 | 0.1056 | **0.2099** | 0.0209 | 0.0355 | 0.0854 | 0.2159 | 0.0475 | 0.1042 | 0.1279 |
| Texas | 0.0890 | 0.1983 | 0.0179 | **0.4570** | 0.0035 | 0.1429 | 0.0177 | 0.0087 | 0.0474 | 0.0131 | 0.0046 |
| Cornell | 0.0886 | 0.2026 | 0.0181 | **0.4460** | 0.0034 | 0.1472 | 0.0183 | 0.0089 | 0.0487 | 0.0134 | 0.0048 |

Table 12: Learned filter weights (Link Prediction) for the top-performing split, distinguishing between homophilic (favoring low-pass filters) and heterophilic (favoring high-pass filters). **First**, second, and third highest filter weights are highlighted.

### 7.6.4 ESTIMATING PRODUCT MANIFOLD SIGNATURE

In our model, the mixed-curvature product manifold $\mathbb{P}^{dc}$ is essential for representing the geometric structure of the data. The curvature configuration needed for each dataset depends on the intrinsic geometry of its graph. To generalize across various datasets, we aim to determine the optimal signature of the product manifold, specifically the proportions of hyperbolic, spherical, and Euclidean components. This estimation is based on analyzing the Ollivier-Ricci curvature (ORC) distribution as a heuristic. See Figures 4 and 5 in Appendix 7.2.1 for the ORC distribution of multiple datasets. For datasets with many positively curved edges, we select a Spherical component, and for those with negatively curved edges, we choose a Hyperbolic component. For example, the curvature distribution of PubMed's edges in Figure 4 shows two significant peaks around $-0.45$ and $+0.25$. Given this distribution, we opt for Spherical and Hyperbolic components when evaluating CUSP on PubMed. Empirically, the best performance for PubMed is achieved with the signature $(\mathbb{H}^{16})^2 \times \mathbb{E}^{16}$. We initialize the curvatures of PubMed's component manifolds with these prominent values: $\mathbb{H}$ with $-0.45$ and $\mathbb{S}$ with $+0.25$. We select two hyperbolic manifolds to capture different curvature ranges.

An overview of this simple idea is provided in Algorithm 1. By systematically analyzing the curvature distribution, our heuristic-based algorithm identifies the manifold signature that best represents the dataset's underlying geometric structure. We heuristically cluster the curvature distribution and identify the centroid curvatures without altering their order or frequencies. The use of predefined dimensions allows for flexibility based on experimental settings. Since optimal dimension allocations can vary and are complex to analyze, we manually set the dimensions of the component manifolds as a hyperparameter. This ensures fair and uniform comparison across multiple datasets, as different datasets may perform best with different configurations. We do not claim that this algorithm finds the best possible, optimal combination of component manifolds, rather, it *estimates* a potential signature that might be a good fit for a particular dataset.

---

**Algorithm 1** Product manifold signature estimation and curvature initialisation

---

**Require:** • Edge curvature histogram $\mathcal{C} = \{(\kappa_i, f_i)\}_{i=1}^N$
   • Threshold $\epsilon$ to distinguish between curved and flat regions
   • Maximum number of Hyperbolic ($\mathcal{H}_{\max}$) and Spherical ($\mathcal{S}_{\max}$) components.
   • Total product manifold dimension $d_{\mathcal{M}}$
   • (Optional) Preferred component manifold dimensions $d_{(h)}^{\text{pre}}, d_{(s)}^{\text{pre}}, d_{(e)}^{\text{pre}}$

**Ensure:** Product manifold signature $\mathbb{P}^{d_{\mathcal{M}}} = \times_{q=1}^{\mathcal{Q}} \mathcal{M}_q^{\kappa_{(q)}, d_{(q)}}$

1: Normalize frequencies: $f_i' = \frac{f_i}{\sum_{j=1}^N f_j}$
2: Construct weighted curvature set: $\mathcal{C}' = \{(\kappa_i, f_i')\}_{i=1}^N$
3: Determine optimal number of clusters $K$ using methods like the elbow method, constrained by $K \leq \mathcal{H}_{\max} + \mathcal{S}_{\max} + 1$ ▷ There can be only 1 Euclidean component
4: Cluster $\mathcal{C}'$ into $K$ clusters using weighted clustering (e.g., weighted K-means)
5: Initialize empty lists $\mathcal{H}, \mathcal{S}, \mathcal{E}$
6: **for** each cluster $c$ in clusters **do**
7:    Compute cluster centroid curvature $\kappa_c = \frac{\sum_{(\kappa_i, f_i') \in c} \kappa_i}{|c|}$
8:    Compute total frequency weight $w_c = \sum_{(\kappa_i, f_i') \in c} f_i'$
9:    **if** $\kappa_c < -\epsilon$ **and** $|\mathcal{H}| \leq \mathcal{H}_{\max}$ **then** ▷ Negative curvature
10:       Assign manifold component: $\mathcal{M}_q = \mathbb{H}^{\kappa_c}$ ▷ Curvature initialization
11:       Add $(\mathcal{M}q, w_c)$ to $\mathcal{H}$
12:    **else if** $\kappa_c > \epsilon$ **and** $|\mathcal{S}| \leq \mathcal{S}_{\max}$ **then** ▷ Positive curvature
13:       Assign manifold component: $\mathcal{M}_q = \mathbb{S}^{\kappa_c}$ ▷ Curvature initialization
14:       Add $(\mathcal{M}q, w_c)$ to $\mathcal{S}$
15:    **else**
16:       Assign manifold component: $\mathcal{M}_q = \mathbb{E}$ ▷ Approximate zero curvature, i.e. $\kappa_c \in [-\epsilon, \epsilon]$
17:       Add $(\mathcal{M}q, w_c)$ to $\mathcal{E}$
18:    **end if**
19: **end for**
20: **if** Predefined dimensions $d_{(h)}^{\text{pre}}, d_{(s)}^{\text{pre}}, d_{(e)}^{\text{pre}}$ are provided **then**
21:    Assign dimensions $d_{(q)}$ to each component $q$ as per predefined values ▷ Dimension assignment
22: **else**
23:    Set total number of components $\mathcal{Q} = |\mathcal{H}| + |\mathcal{S}| + |\mathcal{E}|$ ▷ Dimension assignment
24:    Allocate dimensions $d_{(q)}$ to each component $q$: $d_{(q)} = \left\lfloor d_{\mathcal{M}} \times \frac{w_q}{\sum_{p=1}^{\mathcal{Q}} w_p} \right\rfloor$ ▷ Proportional to weights
25:    Adjust $d_{(q)}$ to ensure $\sum_{q=1}^{\mathcal{Q}} d_{(q)} = d_{\mathcal{M}}$
26: **end if**
27: Formulate manifold signature:

$$\mathbb{P}^{d_{\mathcal{M}}} = \left( \times_{h=1}^{|\mathcal{H}|} \mathbb{H}_h^{\kappa_{(h)}, d_{(h)}} \right) \times \left( \times_{s=1}^{|\mathcal{S}|} \mathbb{S}_s^{\kappa_{(s)}, d_{(s)}} \right) \times \mathbb{E}^{d_{(e)}}$$

---

### 7.6.5 MORE EXPERIMENTAL SETTINGS

| Hyperparameter | Tuning Configurations | Description |
|---|---|---|
| $L$ | $\{5, 10, 15, 20, 25\}$ | Total number of graph filters. |
| $\delta$ | $\{0.2, 0.5, 0.7\}$ | Neighbourhood weight distribution parameter for ORC |
| $\alpha$ | $\{0.1, 0.3, 0.5, 0.9\}$ | Alpha (initialization) parameter for GPR propagation |
| $d^{\mathcal{C}}$ | $\{8, 16, 32, 64\}$ | Total dimension of curvature embeddings. |
| $d^{\mathcal{M}}$ | $\{32, 48, 64, 128, 256\}$ | Total dimension of the product manifold. |
| dropout | $\{0.2, 0.3, 0.5\}$ | Dropout rate |
| epochs | $\{50, 100, 300\}$ | Number of training epochs |
| lr | $\{1e-4, 4e-3, 0.001, 0.01\}$ | Learning rate |
| weight_decay | $\{0, 1e-4, 5e-4\}$ | Weight decay |

Table 13: Hyperparameter configurations used in the experiments for all baselines. Some of the hyperparameters are specific to CUSP. We highlight the final configuration of CUSP for NC in Red.

