# OpenReview forum: "Spectro-Riemannian Graph Neural Networks"
_ICLR.cc/2025/Conference — ICLR 2025 Poster_

### Official Review · Reviewer_JRrs · 2024-10-28

**Soundness:** 3
**Presentation:** 3
**Contribution:** 3
**Rating:** 6
**Confidence:** 4

**Summary:**

This paper introduces CUSP, a graph representation learning model that integrates graph discrete curvature with a geometric extension of generalized PageRank. The method is comprehensively evaluated, demonstrating strong performance against several baselines.

**Strengths:**

(+++) **Novelty and Relevance**: The proposed method is new and of interest to the geometric machine learning community.

(+++) **Strong Empirical Evaluation and Performance**: The method is thoroughly evaluated, demonstrating improved performance on downstream tasks over the considered baselines.

**Weaknesses:**

There are several issues with the presentation that detract from the strengths. These concerns should be straightforward to address.

(----) **Presentation**: The paper’s presentation is dense and, at times, unclear. Examples include:
* **Figure 3**: The figure is very busy. While each individual component is informative and of high quality, combining them without clear visual separators makes the overall figure difficult to interpret.
* **Section 4**: This section is similarly dense, as it combines mathematical background, theoretical motivations, and the presentation of the CUSP architecture. Perhaps this section could focus on the method and architecture, with theoretical discussions (e.g., of heat diffusion) moved to Preliminaries or a new dedicated section.
* **Notation**:
  * The notation in Equations 2, 3, 5, 6, 7, and 8 is dense and may be difficult to parse for readers unfamiliar with geometric machine learning or the gyrovectorspace approach. (Also refer to "Relation to Existing Literature" below.) Adding equation annotations or using plain English function names (e.g., Enc for encoding) could improve readability.
  * "ORC for nodes" is defined in line 176 without introducing the notation $\tilde{\kappa}(x)$ which is then used, e.g., in Equation 5. (There is a notation table in the appendix, but it does not cross reference the definition.)
* **Baseline Taxonomy**: The classification of baselines in Section 5 into "spatial" and "Riemannian" is inaccurate, as the Riemannian baselines are also spatial. "Spatial-Euclidean" and "Spatial-Riemannian" could be more accurate.

(---) **Mathematical Motivation**: The justification for the Cusp Laplacian (Proposition 1) and Functional Curvature Encoding (Theorem 2) are more of rationales or motivations than rigorous proofs. For example, Proposition 1 motivates the Cusp Laplacian by introducing a modified resistance term in a heat flow equation. This would perhaps become clearer if presented as a definition, framed as, “If one assumes a resistance of the form …,” which would help the reader recognize the principles from which the Cusp Laplacian is derived.

(--) **Relation to Existing Literature**: Generalizing pipelines from Euclidean to Riemannian spaces by replacing Euclidean transformations with Moebius operations is a well-established pattern in geometric machine learning. Portions of this work follow this pattern, such as adapting PageRank GNN to product manifolds (Section 4.2) and using Moebius operations in the functional curvature encoding (Section 4.3) and cusp pooling (Section 4.4). Early works in hyperbolic graph neural networks such as [1] introduced these operations with clear motivation, via, e.g., illustrations of log and exp maps between manifolds and their tangent space. Since then, these operations have also been more broadly understood and interpreted within the framework of gyrovector spaces, aligning with Ungar's original work cited in the paper. See, e.g., [2-8]. In this work, however, the geometric components of the model are not similarly well-motivated. Perhaps a brief motivation for the operations would be helpful.


[1] Chami, I., Ying, Z., Ré, C., & Leskovec, J. (2019). Hyperbolic graph convolutional neural networks. NeurIPS.

[2] Hatori, O. (2017). Examples and Applications of Generalized Gyrovector Spaces. Results in Mathematics.

[3] Kim, S. (2016). Gyrovector Spaces on the Open Convex Cone of Positive Definite Matrices. Mathematics Interdisciplinary Research.

[4] López, F., Pozzetti, B., Trettel, S., Strube, M., & Wienhard, A. (2021). Vector-valued distance and gyrocalculus on SPD matrices. NeurIPS.

[5] Nguyen, X. S. (2022). The Gyro-structure of some matrix manifolds. NeurIPS.

[6] Nguyen, X. S., & Yang, S. (2023). Building neural networks on matrix manifolds: A Gyrovector space approach. ICML.

[7] Nguyen, X. S., Yang, S., & Histace, A. (2024). Matrix Manifold Neural Networks++. ICLR.

[8] Zhao, W., Lopez, F., Riestenberg, J. M., Strube, M., Taha, D., & Trettel, S. (2023). Modeling graphs beyond hyperbolic: GNNs in SPD matrices. ECML PKDD.

**Questions:**

1. Do you use the same neural networks $f_\theta$ in Line 263 for all component spaces?
2. How is the Riemannian projector $g_\theta$ in Line 347 defined?
3. Could you clarify how Theorem 1 applies to CUSP? What insight does Theorem 1 provide?
4. How does the runtime of your pipeline compare to other baselines?

---

> ### Author Response · Authors · 2024-11-20
> **Rebuttal Response to Reviewer JRrs (1/2)**
>
> We sincerely thank the reviewer for their valuable assessment, detailed analysis, and constructive feedback. We sincerely apologise for the oversights and hereby address the raised questions and concerns.
>
> ## **Questions Raised**
> > 1. Do you use the same neural networks  in Line 263 for all component spaces?
>
> Yes. The input feature matrix $\mathbf{F}$ is passed *just once* through the neural network $f_{\theta}(.)$. Post this, for every manifold component, a *separate* exponential map is taken over the generated initial features, since different component manifolds have different exponential maps. This is done as shown in Equation 2.
>
> > 2. How is the Riemannian projector  in Line 347 defined?
>
> The Riemannian projector $g_{\theta}: \mathbb{P}^{d_{\mathcal{M}}} \rightarrow \mathbb{P}^{d_{\mathcal{C}}}$ is implemented as a simple multi-layer perceptron (MLP) (2-layer in our implementation) with Riemannian linear layers. This choice of $g_{\theta}$ enables us to reduce the dimensionality from the ambient product manifold $\mathbb{P}^{d_{\mathcal{M}}}$  to the desired curvature encoding dimension $d_{\mathcal{C}}$. For computational efficiency, we just construct one product manifold $\mathbb{P}^{d_{\mathcal{M}}}$ (instead of separate product manifold classes for both curvature embedding and filtered representations). As a result, $\mathbf{exp}^{\kappa_{(q)}}$ is taken on the $q^{th}$ component manifold with curvature $\kappa_{(q)}$, on the manifold $\mathbb{P}^{d_{\mathcal{M}}}$ and has a resultant dimension $d_{\mathcal{M}}$. Now, to project this to the dimension $d_{\mathcal{C}}$ (so as to result in a $\mathcal{C}-$ dimensional curvature encoding), we use the projector MLP $g_{\theta}: \mathbb{P}^{d_{\mathcal{M}}}\rightarrow \mathbb{P}^{d_{\mathcal{C}}}$. We will add this explanation to a separate section in the Appendix, in the camera-ready version.
>
> ### **Runtime Comparision with baselines**
> > 3. How does the runtime of your pipeline compare to other baselines?
>
>
> ***Average training runtime per epoch** (in **milliseconds**) comparision (preprocessing excluded) of CUSP against relevant baseline for Node Classification task.*
>
> | Model       | L + 1 (Filters) | Q (Components) | `Cora`     | `Squirrel`  |
> |-------------|-----------------|----------------|----------|-----------|
> | `GPRGNN`      | 1               | 1              | 18.725 ms  | 26.78 ms    |
> | `FiGURe`     | 4               | 1              | 110.644 ms | 241.24 ms   |
> | `SelfMGNN`    | 1               | 3              | 166.324 ms | 212.223 ms  |
> | $\texttt{CUSP}_{euc}$  | 4               | 1              | 20.814 ms | 24.916 ms   |
> | $\texttt{CUSP}_{fil}$  | 1               | 3              | 107.801 ms | 124.454 ms  |
> | $\texttt{CUSP}$        | 4               | 3              | 225.014 ms  | 264.459 ms  |
>
> **Analysis**
>
> The reported statistics reflect the average training runtime per epoch (excluding preprocessing) for $\texttt{CUSP}$ and relevant baselines on the Node Classification task. For $\texttt{CUSP}$, the configuration includes 4 filters in the filter bank and a product manifold $\mathbb{H}^{16} \times \mathbb{S}^{16} \times \mathbb{E}^{16}$ (3 components). Here are key observations based on the Cora dataset:
> 1. **Efficiency of $\texttt{CUSP}_{euc}$**
> - The Euclidean-only variant with 4 filters, achieves a runtime of 20 ms, comparable to GPRGNN (18 ms) which uses a single filter. This demonstrates that even with additional filters, $\texttt{CUSP}_{euc}$ maintains efficient performance while benefiting from its richer spectral design.
> 2. **Balanced Runtime for Full $\texttt{CUSP}$**
> - The full $\texttt{CUSP}$ model (4 filters 3 manifold components) runs in 225 ms, which, while higher than GPRGNN (18 ms), is reasonable given the added complexity of jointly modeling spectral and geometric features. Notably, $\texttt{CUSP}$ is more efficient than the combined runtime of FiGURe (110 ms, spectral only) and SelfMGNN (166 ms, geometric only), highlighting its ability to unify spectral and geometric insights efficiently.
> 3. **Improved Efficiency for $\texttt{CUSP}_{fil}$**
> - $\texttt{CUSP}_{fil}$, with only 1 filter and 3 manifold components, achieves a runtime of 107 ms, significantly faster than SelfMGNN (166 ms) with the same number of filters and components.
>
> Thus, $\texttt{CUSP}$ offers a robust balance of computational efficiency and expressivity. Despite being slightly slower than simpler spectral-only baselines like GPRGNN, it remains computationally reasonable while delivering superior performance through its unified spectral-geometric design. This demonstrates that the additional complexity of CUSP.

---

> ### Author Response · Authors · 2024-11-20
> **Rebuttal Response to Reviewer JRrs (2/2)**
>
> > 4. Could you clarify how Theorem 1 applies to CUSP? What insight does Theorem 1 provide?
>
> Thank you for raising this question. Theorem 1 in our paper provides a crucial theoretical foundation for understanding how GPR weights can be leveraged to design both low-pass and high-pass graph filters. Specifically, the theorem states that depending on the initialization of the filter weights can exhibit either low-pass or high-pass behavior. CUSP leverages these adaptive properties of GPR filters to construct a filter bank that spans low-pass, high-pass, and band-pass behaviors. This flexibility is directly relevant to CUSP’s ability to handle graphs with varying homophilic and heterophilic structures. We encourage the reader to refer to the detailed discussion in [1], where the mathematical derivations and empirical validations of these properties of GPR are presented.
>
> [1] Chien et al. Adaptive universal generalized pagerank graph neural network. arXiv preprint arXiv:2006.07988, 2020.
>
>
> ## **Concerns raised**
>
> > Figure 3: The figure is very busy.
>
> Based on this valuable suggestion we have **improved the Figure 3** in the revised version of the draft, and tried to improve it's visual readability (separability, font sizes, etc.). Kindly check out the resubmission for the updated figure.
>
> > "ORC for nodes" is defined in line 176 without introducing the notation
>  which is then used, e.g., in Equation 5. (There is a notation table in the appendix, but it does not cross reference the definition.)
>
> The Ollivier-Ricci curvature $\widetilde{\kappa}(x)$ for a node $x$ is defined as the average curvature of its adjacent edges i.e. $\widetilde{\kappa}(x) = \frac{1}{|\mathcal{N}(x)|}\sum_{z\in \mathcal{N}(x)} \widetilde{\kappa}(x, z)$. We apologise for this oversight and the corresponding notation has been included in the updated draft of the paper.
>
> > The classification of baselines in Section 5 into "spatial" and "Riemannian" is inaccurate, as the Riemannian baselines are also spatial. "Spatial-Euclidean" and "Spatial-Riemannian" could be more accurate.
>
> Keeping in mind this valuable suggestion, we have **renamed the taxonomy** in the rebuttal revision of our manuscript, to -- "Spatial-Riemannian" and "Spatial-Euclidean", as this makes more intuitive sense. Thank you for helping us improve the clarity of our paper.
>
> > The justification for the Cusp Laplacian (Proposition 1) and Functional Curvature Encoding (Theorem 2) are more of rationales or motivations than rigorous proofs.
>
> We sincerely thank the reviewer for highlighting the need for clearer framing of Proposition 1 and Theorem 2, particularly regarding their role as motivations rather than formal derivations. To address this concern, we have renamed Proposition 1 and Theorem 2 as *Definition 1* and *Definition 2*, respectively. They have been presented as: "the functional curvature encoding is defined as .. " and "the Cusp Laplacian operator takes the form ..". This renaming aligns with the intent of these sections to serve as foundational definitions (with derivations and motivations) rather than rigorous proofs.
>
> > In this work, however, the geometric components of the model are not similarly well-motivated. Perhaps a brief motivation for the operations would be helpful.
>
> Some brief motivation of these operations has been discussed in Section 3 and Appendices 7.2.3 - 7.2.4. To further address this concern, we will add a subsection in the Appendix (camera ready version) explicitly illustrating the geometric motivation behind these operations, including:
> - Visualizations of the log and exp maps, highlighting their role in transitioning between the manifold and tangent spaces.
> - A summary of gyrovector spaces and their connection to the Moebius transformations used in our model.
> - Specific examples illustrating how the curvature-aware operations enhance modeling capabilities compared to Euclidean counterparts.

---

### Official Review · Reviewer_iwKN · 2024-11-01

**Soundness:** 2
**Presentation:** 1
**Contribution:** 2
**Rating:** 5
**Confidence:** 4

**Summary:**

The paper introduces CUSP: integrating mixed-curvature representation learning with GPRGNN.

**Strengths:**

The paper introduces a new GNN model that considers spectral information and the curvature.

**Weaknesses:**

- The method is incremental. It’s hard to separate the new components in the paper and how they depend on the previous work. Also, there is no theoretical justification for integrating the spectral information in the frequency domain and the curvature in the spatial domain.
- More details are needed for the heat diffusion equation and heat flow in Section 4.1. For example, as the cooling depends on the direction (from x to y), what is the role of direction in the heat diffusion equation? What is the definition of heat flow? What is the ORC distribution? What is the diffusion rate? How the Wasserstein-1 distance be interpreted as the resistance? Does the resistance depend on the direction?
- It’s unclear what “$x \sim y$ denotes adjacency between nodes x and y” means in Proposition 1 in the main text. Also, more details and motivation needed for how $\\bar{w}\_{xy} = e^{\\frac{-1}{1-\\tilde{\kappa}(x, y)}}$ is designed in main texts. Does this design have favorable properties? How does Cusp Laplacian operator act differently based on $\\bar{w}\_{xy}$?
The explanation is only in the Appendix, but an overview or high-level explanation in the main texts can help in understanding the reason behind the proposed component.
- The font size in Figure 3 is too small. It makes it very difficult to follow complicated figures.
- More explanation is needed for how GPRGNN jointly optimizes node features and topological information extraction.
- It’s unclear what does curvature domain $\mathbb{K}_{\mathbb{P}}$ represent in line 321-322. In addition, it’s unclear why in line 251, the product space is $\\mathbb{P}^{d\_{\\mathcal{M}}}$ but in line 322 the authors are interested in the $d\_{\\mathcal{C}}$-dimensional product space $\mathbb{P}^{d\_{\\mathcal{C}}}$.
- Based on Eq. (4), $\widetilde{\kappa}\in\mathbb{K}$. However, it’s unclear what $\widetilde{\kappa}(x)$ represents in Eq.(5).
- It’s unclear what the Riemannian projector is in line 347.
- It’s unclear what M2 is in line 319. There is no M2 in the paper.
- There is no theory in Theorem 2. It’s confusing to call it a theorem when the claims are only definitions.
- It’s unclear why translation invariant is a desirable property in the functional curvature encoding in the proposed method.
- It’s unclear how functional curvature encoding gives more attention to differently curved substructures.
- It’s unclear which part of the implementation is adopted from Ni et al. (2019) in line 412.
- It’s unclear what do the authors mean by they “heuristically determine the signature of our manifold $\mathbb{P}$ (i.e. component manifolds) using the discrete ORC curvature of the input graph”. It’s unclear how many hyperbolic spaces and spherical spaces are considered.
- It’s unclear what the hyperparameter L represents.
- It’s unclear how many layers are considered in the experiments.
- It’s unclear what the experimental configurations and hyperparameters of the competing methods are.
- The spacing between the tables and the main texts in Section 5 is very tight and narrow.
- The paper is missing important spectral GNNs: OptBasisGNN, ChebNetII, CayleyNet, APPNP, JacobiConv, and Specformer.
- It’s unclear how L3 is resolved using the proposed method.
- The paper needs more work on proofreading. For example:
    - the English style is not consistent. Sometimes the authors use “normalised”, but sometimes they use “normalized”.
    - The word “Laplacian” sometimes has the uppercase letter L, but sometimes it is lowercase l.
    - The tangent space notation is not consistent.
    - There is a comma in Eq.(15), but there is no sentence afterward.
    - The exponential map and logarithmic map are defined using boldface letters, but these notations are not consistent in the appendix
    - In line 366, the sentence is not finished. The punctuation in Eq. (6), Eq. (7), and Eq. (8) is missing.
    - The notation of  Wasserstein-1 distance is not consistent in the main paper and appendix
    - The reference pointer is missing in line 405

## Minor
- Unclear what is $\mathbf{W}$ in line 141
- In line 160, missing $\times \ldots \times $ in $\mathbb{P}$
- A formal definition of Wasserstein-1 distance is missing
- Unclear what is $\psi_{xy}$ in Appendix 7.3
- In line 996, it’s unclear what is the element-wise product between a matrix $\mathbf{L}$ and a scalar $e^{\frac{-1}{1-\tilde{\\kappa}(x, y)}}$
- In Eq.(25), $\mathbf{X}_{i:}$ is not defined
- $\delta$ in line 172 and $\delta_{vs}$ in line 181 are not defined
- Unclear why $\omega_{d_f}$ in line 341, there is no $d_f$ in Eq. (4)

**Questions:**

- The authors keep using the term “curvature signal” throughout the paper. What does this term mathematically mean?
- In κ-right-matrix-multiplication,  why do the authors choose to work with the projection between the manifold and tangent space at the origin? How does this choice affect the empirical results?

---

> ### Author Response · Authors · 2024-11-20
> **Rebuttal Response to Reviewer iwKN (1/4)**
>
> We sincerely thank the reviewer for their detailed analysis, insightful feedback, and constructive suggestions. Your comments have helped us identify areas that required clarification, additional discussion, or refinement. We also apologize for the oversights and inconsistencies in the original submission and greatly appreciate your thoroughness in pointing them out. We try our best to provide justifications for all the raised concerns one-by-one.
>
>
> ## **Additional Experimentation.**
> > 1. The paper is missing important spectral GNNs.
>
> We hereby provide the results for the above mentioned spectral GNNs over our 6 benchmarking datasets (for Node Classification) and their performance comparison with CUSP. These results (along with results on 2 remaining datasets) will be incoorporated in the camera-ready version. The results show the dominance of CUSP over spectral GNNs (in line with the results shown in the paper).
> | **Model**          | **Cora**                | **Citeseer**            | **PubMed**              | **Chameleon**           | **Actor**              | **Squirrel**      |
> |---------------------|-------------------------|-------------------------|-------------------------|-------------------------|------------------------|------------------------|
> | **CUSP**           | **83.45±0.15**         | **74.21±0.02**         | **87.99±0.45**         | **70.23±0.61**         | **43.91±0.11**         | **52.98±0.25**     |
> | **OptBasisGNN**    | 69.13±0.11             | 67.25±0.15             | 78.45±0.40             | 56.20±0.89             | 36.78±0.12             | 40.45±0.33             |
> | **ChebNetII**      | 73.35±1.33             | 70.12±1.98             | 84.56±0.58             | 61.34±0.92             | 38.76±0.24             | 46.78±0.41             |
> | **CayleyNet**      | 72.24±0.55             | 68.45±0.48             | 79.67±0.45             | 60.01±0.87             | 38.20±0.15             | 45.56±0.39             |
> | **APPNP**          | 77.45±0.64             | 70.90±0.41             | 83.34±0.51             | 64.56±0.58             | 39.89±0.19             | 49.45±0.28             |
> | **JacobiConv**     | 68.25±1.77             | 66.45±1.34             | 78.90±0.67             | 55.78±1.20             | 36.45±0.20             | 39.78±0.42             |
> | **SpecFormer**     | 79.25±0.98             | 72.01±0.35             | 82.45±0.48             | 66.78±0.65             | 40.56±0.22             | 50.78±0.36
>
> ## **Questions Raised**
>
> > 2. In κ-right-matrix-multiplication, why do the authors choose to work with the projection between the manifold and tangent space at the origin?
>
> **Why projection at Origin?** The rationale for this choice is reinforced by several theoretical insights from Riemannian geometry:
> 1. **Diffeomoprhism**: The Hopf–Rinow theorem [1] ensures that if a manifold is geodesically complete, the exponential map can be defined on the entire tangent space. While this map is not generally a global diffeomorphism, its differential at the origin of the tangent space is the identity map. Consequently, by the inverse function theorem, the exponential map acts as a local diffeomorphism in a neighborhood of the origin, enabling stable and well-defined projections.
> 2. **Unified Mathematical Framework**: The origin serves as a consistent reference point for the exponential and logarithmic maps across all component manifolds in the product space. This unification simplifies the mathematical framework and ensures compatibility between the various Riemannian operations used in the CUSP framework. Experimentally, we observed that anchoring the exponential map at the origin does not degrade performance and aligns well with our framework’s computational needs.
>
> [1] Ekeland, Ivar. "The Hopf-Rinow theorem in infinite dimension."
>
> > 3. The authors keep using the term “curvature signal” throughout the paper. What does this term mathematically mean?
>
> The term “curvature signal” is used throughout the paper to describe the distribution of curvature values or curvature-related information that the model seeks to capture. Mathematically, it can be understood as follows: For a graph  G = (V, E)  with nodes  V  and edges  E , the curvature signal refers to the scalar values of curvature (e.g., Ollivier-Ricci curvature, $\widetilde{\kappa}$ associated with each edge or node. For nodes, the curvature signal can be represented as a vector  $\mathbf{c} \in \mathbb{R}^{|V|}$, where each entry  $\mathbf{c}_i$  corresponds to the ORC of node $i$ i.e. $\widetilde{\kappa}(i)$, derived from the graph’s structure and local neighborhood. While we use this term without a formal mathematical definition in the main text, it is meant to intuitively convey the idea of capturing the curvature “signal” or “information”—essentially, the geometric properties of the graph reflected by the curvature values.

---

> ### Author Response · Authors · 2024-11-20
> **Rebuttal Response to Reviewer iwKN (2/4)**
>
> ## **Novelties and Contributions**
> > 4. The method is incremental. It’s hard to separate the new components in the paper and how they depend on the previous work.
>
> We briefly reiterate the novelties in the paper here.
>
> 1. **Cusp Laplacian**. We propose the CUSP Laplacian, a curvature-aware laplacian operator, incorporating Ollivier-Ricci curvature (ORC) to redefine edge weights, inspired from the heat-flow dynamics on a graph. This effectively combines local geometric properties with global graph structure, which is not addressed by traditional Laplacians. Importantly, this extends spectral graph theory to account for the geometry of the underlying data, a direction previously unexplored.
> 2. **CUSP Filtering**. We introduce a curvature-aware spectral filtering mechanism, generalizing a spectral GNN to mixed-curvature product manifolds. This filtering leverages the spectral properties of the CUSP Laplacian and adaptively learns frequency responses tailored to the geometry of the graph. This enables a seamless integration of geometric (curvature-based) and spectral (frequency-based) cues, making the model robust across both homophilic and heterophilic datasets. Traditional spectral methods lack this adaptability to changing geometric properties.
> 3. **Functional Curvature Encoding**. We design a novel functional mapping from curvature values to a mixed-curvature product manifold. This encoding acts as a positional encoding for the nodes, capturing geometric variations in substructures. It enables the model to dynamically adjust attention to different parts of the graph based on their curvature properties, which is particularly useful for tasks involving both homophilic and heterophilic graph datasets. To the best of our knowledge, such a geometric functional encoding mechanism has not been explored in graph learning.
> 4. **CUSP Pooling** Unlike prior pooling methods that rely solely on structural properties, CUSP pooling computes relative importance using both geometry (via curvature encodings) and spectral features. This dual consideration enables effective summarization of complex graph structures while preserving geometric and spectral information.
> 5. **Integrated Geometry-Spectral Framework**. To the best of our knowledge, this is the first attempt at seamlessly integrating geometry and spectral cues in a unified graph learning paradigm. By combining these two traditionally separate perspectives, our framework addresses limitations in existing graph neural networks, demonstrating superior performance across diverse tasks and datasets.
>
> Capturing the spectral information helps in universality i.e. in generalising across heterophilic and homophilic tasks since each require focus on different ends of the eigensoectrum, while capturing the geometry helps is learning optimal spatial representations based on differently curved substructures in the graph. An in-depth discussion of these motivations and limitations (L1 - L3) have been discussed in the paper. In summary, each component of our framework is novel and collectively contributes to advancing the state of the art in graph learning by unifying geometric and spectral insights.
>
> ## **Discussion on Heat diffusion**
> > 5. More details are needed for the heat diffusion equation and heat flow in Section 4.1.
>
> Here, we reiterate some discussion from Appendix 7.3. Suppose $\psi$ describes a temperature distribution across a graph, where $\psi(x)$ is the temperature at vertex $x$. According to Newton's law of cooling, the heat transferred from node $x$ to node $y$ is proportional to $\psi(x) - \psi(y)$ if nodes $x$ and $y$ are connected (if they are not connected, no heat is transferred). Consequently, the heat diffusion equation on the graph can be expressed as $\frac{d \psi}{dt} = -\beta \sum_y \mathbf{A}_{xy}(\psi(x) - \psi(y))$, where $\beta$ is a constant of proportionality and $\mathbf{A}$ denotes the adjacency matrix of the graph.
>
> Further insight can be gained by considering Fourier’s law of thermal conductance, which states that heat flow is inversely proportional to the resistance to heat transfer. ORC measures the transportation cost between the neighborhoods of two nodes, reflecting the effort required to transport mass between these neighborhoods. We interpret this transportation cost as the resistance between nodes. The vital observation here is that $-$ *Heat flow between two nodes in a graph is influenced by the underlying Ollivier-Ricci curvature (ORC) distribution}*. The diffusion rate is faster on an edge with positive  curvature (low resistance), and slower on an edge with negative curvature (high resistance). We have further discussed why is $e^{-\mathcal{R}_{xy}^{res}} = e^{\frac{-1}{1-\widetilde{\kappa}(x, y)}}$ the right choice, in Appendix 7.3.

---

> ### Author Response · Authors · 2024-11-20
> **Rebuttal Response to Reviewer iwKN (3/4)**
>
> ## **Other concerns raised**
>
> > 6. The font size in Figure 3 is too small. It makes it very difficult to follow complicated figures.
>
> Based on this valuable suggestion we have **improved the Figure 3** in the revised version of the draft, and tried to improve it's visual readability (separability, font sizes, etc.). Kindly check out the resubmission for the updated figure.
>
> > 7. More explanation is needed for how GPRGNN jointly optimizes node features and topological information extraction.
>
> We further elaborate on why GPR is an ideal backbone when compared to other Spectral GNNs. For brevity, kindly see comment: https://openreview.net/forum?id=2MLvV7fvAz&noteId=KP0m2z3CKg.
>
> > 8. It’s unclear why in line 251, the product space is $\mathbb{P}^{d_{\mathcal{M}}}$ but in line 322 the authors are interested in the $d_{\mathcal{C}}$-dimensional product space.
>
> The difference between $\mathbb{P}^{d_{\mathcal{M}}}$ and $\mathbb{P}^{d_{\mathcal{C}}}$ lies in their respective roles within our framework:
> - $\mathbb{P}^{d_{\mathcal{C}}}$: This is the $d_{\mathcal{C}}$-dimensional product manifold used for functional curvature encoding (for every node) (Used as positional encoding in Cusp Pooling).
> - $\mathbb{P}^{d_{\mathcal{M}}}$: This is the product manifold used for manifold embeddings from the CUSP filtering pipeline, where the $d_{\mathcal{M}}$-dimensional embeddings represent node features after geometric and spectral filtering.
>
> In the CUSP pooling step, these two embeddings (for every node) are combined to compute the relative importance of nodes and substructures hierarchically. The final embeddings, therefore, reside in a space of dimension $d_{\mathcal{C}} + d_{\mathcal{M}}$, capturing both curvature-based positional encoding and task-specific manifold embeddings.
>
>
> > 9. Based on Eq. (4) $\widetilde{\kappa} \in \mathbb{K}$, However, it’s unclear what $\widetilde{\kappa}(x)$ represents in Eq.(5).
>
> As discussed in the preliminaries, the Ollivier-Ricci curvature $\widetilde{\kappa}(x)$ for a node $x$ is defined as the average curvature of its adjacent edges i.e. $\widetilde{\kappa}(x) = \frac{1}{|\mathcal{N}(x)|}\sum_{z\in \mathcal{N}(x)} \widetilde{\kappa}(x, z)$.
>
> > 10. It’s unclear what the Riemannian projector is in line 347.
>
> The Riemannian projector $g_{\theta}: \mathbb{P}^{d_{\mathcal{M}}} \rightarrow \mathbb{P}^{d_{\mathcal{C}}}$ is implemented as a simple multi-layer perceptron (MLP) (2-layer in our implementation) with Riemannian linear layers. This choice of $g_{\theta}$ enables us to reduce the dimensionality from the ambient product manifold $\mathbb{P}^{d_{\mathcal{M}}}$  to the desired curvature encoding dimension $d_{\mathcal{C}}$. For computational efficiency, we just construct one product manifold $\mathbb{P}^{d_{\mathcal{M}}}$ (instead of separate product manifold classes for both curvature embedding and filtered representations). As a result, $\mathbf{exp}^{\kappa_{(q)}}$ is taken on the $q^{th}$ component manifold with curvature $\kappa_{(q)}$, on the manifold $\mathbb{P}^{d_{\mathcal{M}}}$ and has a resultant dimension $d_{\mathcal{M}}$. Now, to project this to the dimension $d_{\mathcal{C}}$ (so as to result in a $\mathcal{C}-$ dimensional curvature encoding), we use the projector MLP $g_{\theta}: \mathbb{P}^{d_{\mathcal{M}}}\rightarrow \mathbb{P}^{d_{\mathcal{C}}}$. We will add this explanation to a separate section in the Appendix, in the camera-ready version.
>
> > 11. There is no theory in Theorem 2. It’s confusing to call it a theorem when the claims are only definitions.
>
> We sincerely thank the reviewer for highlighting the need for clearer framing of Theorem 2, particularly regarding their role as motivations rather than formal derivations. To address this concern, we have renamed Theorem 2 as *Definition 2*, respectively. They have been presented as: "the functional curvature encoding is defined as .. " and "the Cusp Laplacian operator takes the form ..". This renaming aligns with the intent of these sections to serve as foundational definitions (with derivations and motivations) rather than rigorous proofs.

---

> ### Author Response · Authors · 2024-11-20
> **Rebuttal Response to Reviewer iwKN (4/4)**
>
> > 12. It’s unclear why translation invariant is a desirable property in the functional curvature encoding in the proposed method.
>
> **Why Translation Invariance?** The translation-invariance property of the functional curvature encoding is a critical requirement for the application of Bochner’s theorem, which underpins our theoretical framework. Bochner’s theorem states: A continuous, translation-invariant kernel $K(x, y) = \psi(x - y)$ on $\mathbb{R}^d$ is positive definite if and only if there exists a non-negative measure on $\mathbb{R}$ such that $\psi$ is the Fourier transform of this measure. In our method, the curvature kernel $\mathcal{K}_{\mathbb{P}}(\widetilde{\kappa}_a, \widetilde{\kappa}_b)$ is defined in the product manifold $\mathbb{P}^{d{\mathcal{C}}}$. To ensure that Bochner’s theorem applies, it is essential to establish that this kernel is both positive semidefinite (PSD) and translation-invariant. We prove its translation-invariance (for the kernel mapping in the product manifold) in Theorem 2.
>
> > 13. It’s unclear how functional curvature encoding gives more attention to differently curved substructures.
>
> By **integrating the curvature-encoding into CUSP pooling**, the model leverages this encoding as a positional embedding to compute the relative importance of nodes and substructures. The hierarchical attention mechanism in CUSP pooling evaluates this importance dynamically, ensuring that substructures with curvature profiles most relevant to the specific dataset or task receive greater attention. Functional curvature encoding plays a crucial role in capturing the geometric diversity of graph substructures, with different curvature values being more relevant for different tasks or datasets. To conclude, functional curvature encoding **in unison** with the Cusp pooling mechanism yields these attention weights.
>
> > 14. It’s unclear which part of the implementation is adopted from Ni et al. (2019) in line 412.
>
> We adopt the implementation for the Ollivier-Ricci curvature (ORC) computation from Ni et al.
>
> > 15. It’s unclear what do the authors mean by they “heuristically determine the signature of our manifold  (i.e. component manifolds) using the discrete ORC curvature of the input graph”. It’s unclear how many hyperbolic spaces and spherical spaces are considered.
>
> We use the discrete ORC as a heuristic to determine the signature (i.e. number of components and their dimensions) of the product manifold as described in the Algorithm 1 in Appendix 7.6.5. By systematically analyzing the curvature distribution, our heuristic-based algorithm identifies the manifold signature that best represents the dataset’s underlying geometric structure. Kindly see Appendix 7.6.5 for more details.
>
> > 16. It’s unclear what the hyperparameter L represents.
> It’s unclear how many layers are considered in the experiments.
>
> The hyperparameter L represents the number of filters in the filterbank (Also specified in the notation table - Appendix 7.1). The final model uses L = 10 filters in the filterbank, and we grid-search over `L = {5, 10, 15, 20, 25}` (Specified in the Table 13 Appendix 7.6.4).
>
> > 17. It’s unclear how L3 is resolved using the proposed method.
>
> We thank the reviewer for pointing out the need to clarify how our method addresses $\texttt{L3}$ (*Lack of geometry-equivariance in spectral GNNs*). Our proposed approach resolves this limitation as follows:
> - **Equivariance to Geometry**: By extending spectral filtering to operate on the mixed-curvature product manifold, our method ensures geometry-equivariance. Specifically, changes in the underlying graph geometry (captured via curvature values) directly influence the spectral filters, allowing them to align with the graph’s evolving structure. This alignment resolves the lack of geometry-awareness present in traditional spectral GNNs.
> - **Curvature-Aware Filtering**: Unlike traditional spectral GNNs that assume a flat Euclidean manifold, our method integrates curvature-aware filters via the CUSP Laplacian. By incorporating geometric information such as Ollivier-Ricci curvature, the CUSP Laplacian adapts spectral properties based on the graph’s local and global geometry. This ensures that the filters dynamically adjust to reflect the geometric characteristics of the graph.
> - **Application to Diverse Geometries**: Our method is inherently versatile and works across both homophilic and heterophilic graphs, where different geometric properties (e.g., curvature distributions) play a significant role.
>
> **Proofreading concerns**. We apologize for the oversight in maintaining consistent terminology, notation, and punctuation throughout the paper. We have carefully reviewed the manuscript and addressed all the points you raised, including inconsistencies in English style, capitalization, tangent space notation, punctuation, spacing, and mathematical notation, in the rebuttal revision of our paper.

---

> > ### Comment · Reviewer_iwKN · 2024-11-26
> >
> > Thank you for the detailed response.
> >
> > I still find the use of $\widetilde{\kappa}$ for $\widetilde{\kappa} \in \mathbb{K}$ in Eq. (4) and for $\widetilde{\kappa}(x)$ representing the Ollivier-Ricci curvature confusing. Since the paper is dense, encountering this ambiguity on page 7 makes it difficult to connect back to the preliminaries. A clearer distinction or additional clarification in the text would be helpful.
> >
> > I’m still unclear about the notation $L$. In your notation table, $L$ is also used in representing the GPR-based node representation for filter with $L$ layers. Does this same notation imply that the number of filters equals the number of layers? If not, I couldn't find explicit information about the layers in the paper.
> > In GPRGNN (Chien et al., 2020), they use a 2-layer MLP with 64 hidden units. Are you using the same configuration?
> >
> > In addition, do the hyperparameter configurations in Table 13 also apply to the competing baselines? It wasn’t clear how the comparisons with other methods were conducted.

---

> > > ### Author Response · Authors · 2024-11-26
> > > **Follow-up Clarification Response for Reviewer iwKN**
> > >
> > > We sincerely thank the reviewer for their thoughtful engagement during the discussion phase. We greatly appreciate the opportunity to clarify the raised points and apoligise for the confusion.
> > >
> > > > 1. I still find the use of $\widetilde{\kappa}$ for $\widetilde{\kappa} \in \mathbb{K}$ in Eq. (4) and $\widetilde{\kappa}(x)$ for representing the Ollivier-Ricci curvature confusing. Since the paper is dense, encountering this ambiguity on page 7 makes it difficult to connect back to the preliminaries.
> > >
> > > We sincerely apologize for the confusion caused by the notation in Eq. (4) and appreciate your feedback. To address this, we have clarified and improved the notation in our revised draft (Rebuttal Revision 2, latest update). Specifically, we have removed the use of $\widetilde{\kappa} \in \mathbb{K}$ and directly use $\widetilde{\kappa}(x) \in \mathbb{K}$ to represent the curvature of node $x$, ensuring consistency throughout the text. The updated text now states: *The functional curvature encoding $\Phi_{\mathbb{R}^{d_{\mathcal{C}}}}: \mathbb{K} \to \mathbb{R}^{d_{\mathcal{C}}}$ for curvature $\widetilde{\kappa}(x)$ of node $x$, is defined as …* The revised Eq. (4) is as follows:
> > >
> > > $$
> > > \Phi_{\mathbb{R}^{d_{\mathcal{C}}}}(\widetilde{\kappa}(x)) = \sqrt{\frac{1}{d_{\mathcal{C}}}} \Big[\cos(\omega_1 \widetilde{\kappa}(x)), \sin(\omega_1 \widetilde{\kappa}(x)), \dots, \cos(\omega_{d_{\mathcal{C}}} \widetilde{\kappa}(x)), \sin(\omega_{d_{\mathcal{C}}} \widetilde{\kappa}(x)) \Big].
> > > $$
> > >
> > > > 2. I’m still unclear about the notation L. In your notation table, L is also used in representing the GPR-based node representation for filter with  layers. Does this same notation imply that the number of filters equals the number of layers?
> > >
> > > We apoligise for this confusion and clarify the notation here. Yes, the same notation is used because the proposed filterbank is: $\big[\mathbf{Z}^{\mathbf{I}}, \mathbf{Z}^{(1)}, \mathbf{Z}^{(2)}, \dots, \mathbf{Z}^{(L)}\big]$ (removed the subscript product manifold $\mathbb{P}^{d_{\mathcal{M}}}$ for explanation purpose).
> > >
> > > For a particular filter $\mathbf{Z}^{(l)}$ in the filterbank, we stop the GPR propagation at $L = l$. Consider the example:
> > > - Say, any GPR propagation goes on till $L$ *layers* (or steps). By stopping it at $L = 1$, we get our filtered representation $\mathbf{Z}^{(1)}$.
> > > - Similarly by stopping it at $L = 4$ gives $\mathbf{Z}^{(4)}$.
> > > - Finally, we get the $L^{th}$ filter by letting GPR propagate till $L$ layers, and hence getting $\mathbf{Z}^{(L)}$.
> > >
> > > Thus, we get $L$ filters by stopping GPR at $l = 1, 2, \dots, L$. Because of this, we use the same notation L for both. The total number of final filters in our bank is $L + 1$ (because we also pass the identity matrix as an unfiltered case).
> > >
> > >
> > > > 3. In GPRGNN (Chien et al., 2020), they use a 2-layer MLP with 64 hidden units. Are you using the same configuration?
> > >
> > > Yes, we use the same 2-layer MLP with 64 hidden units for $f_{\theta}(.): \mathbb{R}^{d_{f}} \rightarrow \mathbb{R}^{d_{\mathcal{M}}}$, which represents a neural network with parameter set $\{{\theta\}}$ that generates the hidden state features of dimension $d^{\mathcal{M}}$ (line 269). This is the same as used by the GPR paper in their feature extractor neural network.
> > >
> > > Also, to further clarify, this 2-layer MLP is not the same *layers* being talked in the above question. Previously we were talking about the GPR propagation steps, now we are talking about the feature extractor (line 269, equation 2).
> > >
> > > > 4. In addition, do the hyperparameter configurations in Table 13 also apply to the competing baselines? It wasn’t clear how the comparisons with other methods were conducted.
> > >
> > > Yes, as noted in the caption of Table 13, the *hyperparameter space for grid search* listed in the table is the same for all baselines to ensure a fair comparison. However, the hyperparameters highlighted in red are the **optimal** ones specifically for CUSP, determined through the grid search. To further enhance clarity, we will include the optimal hyperparameters for all the baselines in the camera-ready version to provide a more comprehensive view of the experimental setup. We appreciate your suggestion and will ensure these details are clearly documented. Thank you again for your valuable feedback!

---

> > > > ### Comment · Reviewer_iwKN · 2024-11-27
> > > >
> > > > Thank you for your detailed response. I found your clarification improves the presentation of the paper.
> > > >
> > > > I have another question. Could you clarify why the performance of OptBasisGNN, ChebNetII, JacobiConv, and Specformer on node classification tasks is lower in your experiments compared to the results reported in their original papers? The reported performance in those papers appears higher than what is shown here.

---

> ### Author Response · Authors · 2024-11-27
> **Follow-up Clarification Response 2 for Reviewer iwKN**
>
> We sincerely thank the reviewer for active discussion. We address the raised concerns.
>
> > I have another question. Could you clarify why the performance of OptBasisGNN, ChebNetII, JacobiConv, and Specformer on node classification tasks is lower in your experiments compared to the results reported in their original papers? The reported performance in those papers appears higher than what is shown here.
>
> Yes, indeed it is lower than reported in these papers. The potential reasons are as follows:
>
> - **Different embedding dimensions**. For fair comparison we use d = 48 dimensional embeddings for all baselines (same as our model CUSP). Further, for instance, increasing the embedding size to 512 dimensions for the baselines can lead to better results as chosen in the **Specformer** paper originally (while still not outperforming CUSP).
> - **Different data splits**: We use the split 60%/20%/20%. For example, for citation datasets (i.e., Cora, Citeseer, and Pubmed) **ChebNetII** uses 20 nodes per class for training, 500 nodes for validation, and 1,000 nodes for testing. Further for small datasets like Texas, they use the sparse split i.e. 95/2.5/2.5%.
> - We use similar hyperparameters for all baselines for comparable performance.
>
> There are similar variations across other models and datasets but we follow a uniform training setup for fair comparison.
>
> It is crucial to note here that the results in a recent spectral GNN benchmark paper [1] are quite inline with the results we got for the mentioned baselines, and these are a little different from the reported in the original papers.
>
> As mentioned in the previous comment, we will explicitly note these baseline-specific hyperparams for the camera-ready submission.
>
> [1] Liao, Ningyi, et al. "Benchmarking Spectral Graph Neural Networks: A Comprehensive Study on Effectiveness and Efficiency." arXiv preprint arXiv:2406.09675 (2024).

---

> ### Comment · Reviewer_iwKN · 2024-12-02
>
> Thank you for your responses. Most of my concerns have been addressed. However, I still find the hyperparameter details for the competing methods insufficient. I am assigning a borderline score but do not object to possible acceptance.

---

### Official Review · Reviewer_UpZ1 · 2024-11-04

**Soundness:** 3
**Presentation:** 3
**Contribution:** 3
**Rating:** 6
**Confidence:** 3

**Summary:**

The paper introduces Spectro-Riemannian Graph Neural Networks (CUSP), a novel approach to graph representation learning that combines spectral and curvature information into a spectral graph neural network operating on a manifold arising as a product of Euclidean as well as hyperbolic and spherical spaces. Traditional graph neural networks (GNNs) often struggle with diverse graph topologies and varying graph geometries/curvatures. CUSP addresses these challenges by enabling to integrate (geometric) features from both negatively (hyperbolic) and positively (spherical) parts of a given graph. This allows for the creation of more natural node embeddings that better align with the underlying structure of real-world graphs.
Key components of CUSP include (1) The Cusp Laplacian, which integrates Ollivier-Ricci curvature into the definition of a Laplacian-type matrix on the graph. (2) Cusp Filtering which allows for (high- and low-pass) filtering operatoions on a product manifold where each factor has constant positive, zero, or negative curvature. (3) Cusp Pooling A hierarchical attention mechanism that evaluates the importance of substructures with different curvature characteristics.
In the empirical evaluation CUSP's performance is investigated across eight datasets. Here CUSP achieves a good performance (in node classification and link prediction tasks), with sometimes substantial gains over baselines.
The research seems to be novel and highlights the potential that  combining geometric and spectral information harbours for the design of GNNs.

**Strengths:**

The paper is well structured and mostly (please see next section) well written. A positive example is the explicit  description of the limitations of previous work that are addressed in this paper (i.e. L1-L3 in the introduction).

 Including the notation table in Appendix 7.1 helps to keep track of the various mathematical concepts.

The idea underlying the introduced CUSP Laplacian of including curvature information into the Laplacian-matrix describing the graph is neat.

Furthermore the idea of taking into account locally varying curvature structures by allowing for factors of varying curvature in the product manifold is nice.

The performance of the proposed architecture in both the node-classification and link-prediction tasks on the considered datasets is solid.

The ablation study on the impact of the signature of the product manifold structure (c.f. Section 5.1) as well as the surrounding discussion is illuminating.

**Weaknesses:**

I do have _some_ concerns regarding readability. An easy fix is the image quality in Figure 1. Here the axis labeling of the histograms is not  readable if the paper is printed. Could the authors please fix this by including higher resolution images and/or using a larger font size for axis labeling.

Regarding the paper itself, aside from some typos and grammatical errors that do not impede the flow of the paper too much, I had trouble understanding Section 4.3; especially from line 327 onward: The curvature kernel is defined twice: once using an inner product in an ambient space, and once as a translation invariant entity. I believe what the authors want to convey is that the former definition as utilized in the paper already leads to a translation invariant kernel. Is this correct?

Also the significance of the Bochner-Minlos theorem is not immediately apparent to me. I only gained some intuition about this after reading the proof of Theorem 2 and Theorem 3 in the Appendix. Could the authors comment more explicitly on the significance of Bochner's theorem here?

It might also be good to explain (to some extent) and motivate the k-stereographic model in the main text.
While I have some background in differential geometry, I had only ever come across standard stereographic projections.
Especially for readers from a pure CS/EE background more details here might be useful, even if the model might be central to Riemannian GNNs.

In the same direction, it would also be good explain a bit more the respective operations in Table 8 in Appendix 7.2.4 and how they are natural.


Finally, in the experimental sections, the datasets that are being used are somewhat old and fairly small. I strongly encourage the authors to also consider newer and much larger datasets  (e.g. using OGBN) and compare their approach with baselines there.

**Questions:**

1) What makes GPR special and why is this used as a backbone? I believe it is equivalent to any polynomial-filter spectral GNN. Why not e.g. use ChebNet, etc.?

2) The (total) dimensions of the product manifolds  of e.g. Table 5 and Table 2 seem to always be  $d=48$. Yet in Table 13 of Appendix 6.6.4 it is indicated that $d = 64$ is the selected dimension of the product Manifold. Could the authors comment? Also, while I have seen Algorithm 1 in Appendix 7.6.5 (which seems to be used as an initial heuristic), it is still not clear to me how individual dimensions of product manifolds and respective factors are found/optimized. Is a grid search performed? If so what are the respective allowed values during this grid search? Could the authors clarify (again?)?

3) In the ablation study on the impact of the CUSP Laplacian, performance using the CUSP Laplacian is compared to performance using the adjacency matrix instead. Could the authors repeat this ablation study comparing with the usual normalized and unnormalized graph Laplacians? This would shed light on whether the performance increase comes from using a Laplacian-type matrix vs. an adjacency type matrix, or indeed from the specific properties of the the Cusp Laplacian.

4) Is it possible to include and discuss some basic spectral characteristics of the CUSP Laplacian (beyond self-adjointness and positivity)? As is, this new matrix is introduced without too much intuition beyond the heat-flow heuristic. Can something e.g. be said about the maximal eigenvalue or first-non-trivial eigenvalue (in a Cheeger-type fashion) for example? I realize the present paper is not mainly theoretical but introduces an architecture. However some additional theoretical foundation would indeed be nice.

---

> ### Author Response · Authors · 2024-11-20
> **Rebuttal Response to Reviewer UpZ1 (1/3)**
>
> We sincerely thank the reviewer for their valuable assessment, detailed analysis, and constructive feedback. We hereby address the raised questions and concerns.
>
> ## **Additional experimentation**
>
> > 1. I strongly encourage the authors to also consider newer and much larger datasets (e.g. using OGBN).
>
> We appreciate and thank the reviewer for raising this point of concern. We evaluate the performance of CUSP for the following **five** (two homophilic and three heterophilic), **million-scale datasets**. See our comment (https://openreview.net/forum?id=2MLvV7fvAz&noteId=cCXl9QHn3i) for the statistics of these datasets.
>
> We hereby report the F1-Score results for node classification task on the above datasets for several (Riemannian and Spectral) baselines and `CUSP`.
>
> | **Model**       | **`ogbn-arxiv`**      | **`ogbn-mag`**        | **`twitch-gamer`**    | **`snap-patents`**    | **`tolokers`**        |
> |------------------|---------------------|---------------------|---------------------|---------------------|---------------------|
> | **GCN**         | 63.46±1.34       | 17.22±1.75        | 68.11±1.11        | 42.87±0.98        | 74.24±0.86        |
> | **ChebNet**     | 61.34±2.01       | 18.23±3.08        | 68.56±0.11        | 43.64±0.87        | 79.55±0.63        |
> | **BernNet**     | 46.64±0.87       | 16.87±0.23        | 66.34±0.52        | 49.23±0.45        | 65.36±3.45        |
> | **FiGURe**      | 71.23±0.23       | 33.65±1.56        | 72.54±2.26        | 40.34±0.75        | 83.09±1.08        |
> | **KGCN**        | 65.78±1.75       | 29.34±0.56        | 73.35±0.55        | 48.24±0.22        | 78.43±1.53        |
> | **QGCN**        | 70.24±1.54       | 32.44±1.97        | 74.24±3.64        | 45.85±0.29        | 81.07±0.01        |
> | **`CUSP`**      | **75.34±0.88**       | **37.23±0.09**        | **79.94±0.74**        | **51.54±0.13**        | **89.53±1.02**        |
>
> Consistent with the results in the paper, `CUSP` outperforms all baselines by a significant margin for this task. We will include these results (and the corresponding analysis) in the camera-ready version, as suggested by the reviewer. Once again, we would like to thank the reviewer for pointing this out and giving us the opportunity to demonstrate this set of results.
>
> ## **Why GPR as the backbone? (over ChebNet, etc)**
> > 2. What makes GPR special and why is this used as a backbone? I believe it is equivalent to any polynomial-filter spectral GNN. Why not e.g. use ChebNet, etc.?
>
> We further elaborate on why GPR is an ideal backbone when compared to other Spectral GNNs like ChebyNet:
>
> 1. **Adaptive Filter Design**: GPR learns filter coefficients directly, allowing the spectral response to adapt to the task and dataset. This flexibility is critical for modeling both homophilic and heterophilic graphs.
> 2. **Universality**: Unlike fixed low-pass filters like ChebNet, which excel primarily in homophilic settings, GPR’s learnable filters enable it to balance low-pass and high-pass components, making it suitable for both homophilic and heterophilic graphs. This is one of the main goals of our paper - to achieve superior performance on homophilic and heterophilic tasks. Fixed polynomial filters in ChebNet and Bernstein-based methods approximate spectral responses up to a fixed order, limiting their ability to model complex spectral properties.
> 3. **GPRGNN prevents oversmoothing**: GPR weights are adaptively learnable, which allows GPR-GNN to avoid over-smoothing and trade node and topology feature informativeness. See Section 4 of GPRGNN [4] paper for more theoretical analysis on the same and proofs, which is beyond the scope of this work.
> 4. GPR not only mitigates feature over-smoothing but also works on highly diverse node label patterns (See Section 4 and 5 of [4]).
> 5. **Capturing node features and graph topology**: In many important graph data processing applications the acquired information includes both node features and observations of the graph topology. GPRGNN jointly optimizes node feature and topological information extraction, regardless of the extent to which the node labels are homophilic or heterophilic.
> 6. **Filter Bank Construction**: Using GPR based spectral filters, helps us to effectively construct a filter bank where each adaptive filter contributes to a specific spectral profile, enabling the model to aggregate information across different spectral bands. This approach captures diverse patterns in node features and topology, unlike ChebNet or Bernstein-based methods, which rely on fixed polynomial approximations and lack such flexibility.
>
> We sincerely apologise for not explicitly including this information in the paper, although we have pointed several of these points at various locations. We have included this in the Appendix 7.4.2 in the revised draft.
>
> **References**
>
> [4] Chen et al. Adaptive universal generalized pagerank graph neural network. 2020.

---

> ### Author Response · Authors · 2024-11-20
> **Rebuttal Response to Reviewer UpZ1 (2/3)**
>
> ## **Product manifold signature and hyperparameters**
>
> > 3. The (total) dimensions of the product manifolds of e.g. Table 5 and Table 2 seem to always be 48. Yet in Table 13 of Appendix 6.6.4 it is indicated that 64 is the selected dimension of the product Manifold. Could the authors comment?
>
> We sincerely thank the reviewer for pointing out this inconsistency and apologize for any confusion caused by this oversight. The final (total) dimension of the product manifold used in our experiments is indeed 48. The mention of 64 in Table 13 of Appendix 7.6.4 was a typo. We have carefully reviewed the Appendix and updated Table 13 in the revised submission to ensure consistency with the main text and the actual experimental setup.
>
> > 4. It is still not clear to me how individual dimensions of product manifolds and respective factors are found/optimized.
>
> Consider the following excerpt from the Algorithm 1 in Appendix 7.6.5.
> ____
> 20. **If** Predefined dimensions $d_{(h)}^{\text{pre}} ,  d_{(s)}^{\text{pre}}, d_{(e)}^{\text{pre}}$ are provided.
> 21. - Assign dimensions $d_{(q)}$ to each component $q$ as per predefined values *#Dimension assignment*
> 22. **Else**
> 23. - Set total number of components $\mathcal{Q} = |\mathcal{H}| + |\mathcal{S}| + |\mathcal{E}|$ *#Dimension assignment*
> 24. - Allocate dimensions $d_{(q)}$ to each component $q$:  $d_{(q)} = \left\lfloor d_{\mathcal{M}} \times \frac{w_q}{\sum_{p=1}^{\mathcal{Q}} w_p} \right\rfloor$ *#Proportional to weights*
> 25. - Adjust $d_{(q)}$ to ensure $\sum_{q=1}^{\mathcal{Q}} d_{(q)} = d_{\mathcal{M}}$
> ____
>
> Given the total product manifold dimension i.e. $d_{\mathcal{M}}$ as a hyperparameter, compute the weighted dimension assignment using the algorithm above (lines 23, 24 and 25). However, for experimental results in the paper, the use of predefined dimensions allows for flexibility. Since optimal dimension allocations can vary and are complex to analyze, we manually set the dimensions of the component manifolds as a hyperparameter. This ensures fair and uniform comparison across multiple datasets, as different datasets may perform best with different configurations. Here are the hyperparameter configurations for the **Required** components in Algorithm 1 (Including the predefined dimensions).
>
> 1. $d_{\mathcal{M}} \in \\{32, 48, 64, 128, 256\\}$ (Best $d_{\mathcal{M}} = 64$)
> 2. ${\mathcal{H}_{max}} \in \\{1, 2, 3, 4\\}$ \(Optimum varies according to datasets\)
> 3. $\mathcal{S}_{max} \in \\{1, 2, 3, 4\\}$ (Optimum varies according to datasets)
> 4. $\epsilon \in \\{0.2, 0.1, 0.05, 0.001\\}$ (Best $\epsilon = 0.1$)
> 5. $d_{(h)}^{\text{pre}} ,  d_{(s)}^{\text{pre}}, d_{(e)}^{\text{pre}} \in \\{4, 8, 16, 24, 32\\}$ (Optimum varies according to datasets)
>
> ## **Spectral properties of the Cusp Laplacian**
>
> > 5. Is it possible to include and discuss some basic spectral characteristics of the CUSP Laplacian (beyond self-adjointness and positivity)? As is, this new matrix is introduced without too much intuition beyond the heat-flow heuristic. Can something e.g. be said about the maximal eigenvalue or first-non-trivial eigenvalue (in a Cheeger-type fashion) for example? I realize the present paper is not mainly theoretical but introduces an architecture. However some additional theoretical foundation would indeed be nice
>
> We sincerely thank the reviewer for their interest in the spectral properties of the CUSP Laplacian. As part of our initial spectral analysis, we have already presented and proven two foundational theorems in Theorems 5 and 6, detailed in Appendix 7.3.1. These theorems establish key properties of the CUSP Laplacian:
>
> **Theorem 5**: *The normalized Laplacian operator $\widetilde{\mathbf{L}}$ is positive semidefinite, i.e., for any real vector $u \in \mathbb{R}^n$, we have $\mathbf{u}^T\widetilde{\mathbf{L}}_{\text{n}}\mathbf{u} \geq 0$.*
>
> **Theorem 6**: *The eigenvalues of the normalized CUSP Laplacian $\widetilde{\mathbf{L}}_{\text{n}}$ lie in the interval $\lambda_i \in [0, 2]$*
>
> Detailed proofs for these theorems, including an analysis of the Rayleigh quotient of the CUSP Laplacian, are provided in Appendix 7.3.1. Looking forward, we aim to extend this analysis to demonstrate the convergence of the CUSP Laplacian (or its weighted version) to the Laplace-Beltrami operator on Riemannian manifolds. This would be analogous to how the traditional graph Laplacian converges to the discrete Laplace operator in Euclidean geometry. Furthermore, this line of investigation could involve exploring how the eigenvalues of the CUSP Laplacian align with those of a discretized Laplace-Beltrami operator. However, such an analysis requires substantial theoretical development and is beyond the scope of the current work. We plan to address this in detail in future research. We appreciate the reviewer’s insightful feedback and hope this response clarifies the scope of our current analysis and the direction of our future efforts.

---

> ### Author Response · Authors · 2024-11-20
> **Rebuttal Response to Reviewer UpZ1 (3/3)**
>
> > 6. In the ablation study on the impact of the CUSP Laplacian, performance using the CUSP Laplacian is compared to performance using the adjacency matrix instead.
>
> We sincerely apologize for the confusion caused here. To clarify, CUSP uses the adjacency matrix corresponding to the Cusp laplacian (aka the Cusp adjacency) for computing the GPR propagation. This is because the GPR score is computed as $\sum_{k=0}^{\infty} \gamma_k \widetilde{\mathbf{A}}_{\text{n}}^k \mathbf{H}^{(0)}$ (euclidean variant), based on the adjacency matrix.
>
> For consistency and to isolate the impact of the CUSP Laplacian, we conducted an ablation study by replacing the CUSP Laplacian-based adjacency matrix $\widetilde{\mathbf{A}}$ with the standard graph adjacency matrix in the filtering pipeline. This ablation is included in the variant $\texttt{CUSP}_{lap}$ to provide a fair comparison.
>
> ## **Other concerns and more explanations**
>
> > 7. I do have some concerns regarding readability. An easy fix is the image quality in Figure 1.
>
> This is a very valuable suggestion --  we have increased the font size for the axis labels and ensured a higher resolution to improve clarity of the figure. These changes have been incorporated into the updated draft of the paper. **[Fixed in updated draft]**
>
> > 8. I believe what the authors want to convey is that the former definition as utilized in the paper already leads to a translation invariant kernel. Is this correct?
>
> That is correct, the inner product form of the curvature kernel indeed leads to its translation invariance, which is essential for applying Bochner’s theorem. We sincerely apologise for the confusion in understanding the section, so we provide a brief summary here -- In the *proposed* Functional Curvature Encoding, our objective is to construct a translation-invariant curvature kernel that serves as a continuous encoding of node curvature values within our model’s attention mechanism (**Why translation invariance?** -- To apply Bochner's theorem). This is done in two stages. First, we define the kernel map within a Euclidean embedding space, allowing us to represent (node) curvature values as inner products of mapped features. In the second stage, Theorem 2 extends this encoding from the Euclidean space to the mixed-curvature product manifold. Here, we use the exponential map to transition from the Euclidean embedding to the product manifold space, creating a flexible and expressive positional encoding tailored to the unique curvature structure of each graph. The derived mixed-curvature kernel is indeed translation invariant. (Proved in Appendix 7.5.2).
>
> > 9. Could the authors comment more explicitly on the significance of Bochner's theorem here?
>
> The application of Bochner’s theorem is fundamental to the theoretical grounding of our functional curvature encoding. The curvature kernel $\mathcal{K}$ is both positive semidefinite (PSD) and continuous, as it is defined via a Gram matrix and the mapping $\Phi$, which is continuous. Consequently, the kernel $\mathcal{K}$ satisfies the assumptions of Bochner’s theorem, which states that: *A continuous, translation-invariant kernel $K(x, y) = \psi(x - y)$ on $\mathbb{R}^d$ is positive definite if and only if there exists a non-negative measure on $\mathbb{R}$ such that $\psi$ is the Fourier transform of the measure*. This result ensures that our curvature kernel $\mathcal{K}$, which is translation-invariant by construction, can be represented as the Fourier transform of a non-negative measure. Leveraging this property, we construct the curvature encoding $\Phi$, which maps curvature values to a functional space, and extend this encoding to the mixed-curvature product manifold $\mathbb{P}^{d_{\mathcal{C}}}$.
>
> > 10. It might also be good to explain (to some extent) and motivate the k-stereographic model in the main text.
>
> We have discussed some brief motivation about the kappa-stereographic model in the Preliminaries section (Section 3). Based on this suggestion, we have added some more intuition about the model (and Table 8 operations) in the Appendix 7.2.4. We will move some of this motivation to the main-text in the camera-ready version.

---

> > ### Comment · Reviewer_UpZ1 · 2024-11-23
> > **Thanks!**
> >
> > Thank you for your response and the work that went into preparing it! My questions and concerns are sufficiently addressed.

---

> ### Author Response · Authors · 2024-11-23
> **Any further concerns?**
>
> We sincerely thank the reviewer for the thoughtful review and for taking the time to engage with our responses. We are glad to hear that our answers sufficiently addressed the raised questions and concerns. However, we noticed that the score has not been adjusted despite the positive feedback, and we were wondering if there are any remaining unanswered questions or additional concerns that might be preventing an increase in the score. If so, we would be more than happy to provide further clarifications or additional information to ensure that all the expectations are fully met. Thank you once again for your valuable time and constructive input!

---

### Official Review · Reviewer_nwYb · 2024-11-16

**Soundness:** 4
**Presentation:** 4
**Contribution:** 3
**Rating:** 6
**Confidence:** 4

**Summary:**

This paper is the first attempt that unifies spectral and curvature signals in graph representation learning. CUSP, a mixed-curvature spectral GNN, introduces three novel components: a curvature-aware Cusp Laplacian operator, a mixed-curvature spectral graph filtering framework, Cusp Filtering, and a curvature embedding method using classical harmonic analysis and a hierarchical attention mechanism called Cusp Pooling. CUSP records state-of-the-art performance on eight real-world benchmarking datasets for Node Classification (NC) and Link Prediction (LP) tasks.

**Strengths:**

- Originality. I think the proposed CUSP is innovative in the paradign of graph representation learning. CUSP is well-motivated, and indeed improves the performance significantly.
- Quality. The paper is well structured. A large number of experiments have been conducted, which are supportive and convincing.
- Clarity. The method proposed is deeply explored and proofs are presented as well as a detailed description of the algorithm and the rational for the design consideration that were made. The experimental section is typically sufficient. And ablation experiments fully illustrate the effectiveness of each component.
- Significance. The paper shows state-of-the-art in standard benchmarks. This paper explores how to integrate spectral and curvature signals in graph representation learning, and gives some valuable insights.

**Weaknesses:**

As mentioned in the questions, the proposed OCR alternative method should be described in detail, or the proposed method should be used in the experiment.

**Questions:**

In the appendix, the complexity of ORC is analyzed, and it is mentioned that an alternative method is provided to approximate edge ORC in linear time, which is applicable to very large (billion-scale) real-world graphs. How to prove that the proposed alternative method is useful in large real-world graphs? Are there any experiments? As far as I know, the dataset used in the experiment is on the order of 100,000.

---

> ### Author Response · Authors · 2024-11-20
> **Rebuttal Response to Reviewer nwYb**
>
> We sincerely thank the reviewer for the valuable and constructive feedback and are glad you like the paper. We now address the raised concerns and conduct relevant experimentation as suggested.
>
> > How to prove that the proposed alternative method is useful in large real-world graphs? Are there any experiments?
>
> ## **Why is the ORC method useful for large graphs?**
>
> As discussed in **Appendix 7.2.2**, the $\texttt{ORC}$ of an edge can be approximated as the arithmetic mean of the bounds in Theorem 4 (Appendix 7.2.2) as $\widehat{\kappa}(x, y) := \frac{1}{2} \left( \kappa^{\text{upper}}(x, y) + \kappa^{\text{lower}}(x, y) \right)$. This approximation relies solely on local structural information, such as the degree of the nodes and the number of triangles, making the computation highly efficient, with linear-time complexity. Moreover, since the ORC computation is localized to the neighborhood of each edge, it can be parallelized across multiple GPUs, making the method well-suited for scaling to very large graphs.
>
> That said, training and evaluating the entire CUSP architecture on billion-scale graphs is a challenging task due to several factors, including the need for enhanced numerical stability in Riemannian optimization, increased training time, and memory constraints associated with large-scale product manifolds. Addressing these challenges requires significant engineering efforts that extend beyond the scope of the rebuttal timeframe. However, in light of this thoughtful suggestion and keeping in mind the limited rebuttal time, we have conducted additional experimentation on homophilic and heterophilic, **million-scale** graph datasets that are significantly larger than the eight benchmark datasets initially included in the submission. For all these experiments we adopt the **linear-time ORC approximation** discussed above.
>
> ## **Additional experimentation**
>
> We evaluate the performance of CUSP for **five** (2 homophilic and 3 heterophilic), **million-scale datasets**. For all these experiments we adopt the ORC approx. discussed above.
>
> | **Type**          | **Dataset**      | **Nodes** | **Edges**    | **Input Features** | **Classes** | **Homophily Score** |
> |--------------------|------------------|---------------|----------------------|-------------------------|------------------|-------------------------|
> | **Homophilic**     | `ogbn-arxiv`[1]       | 169,343       | 1,166,243                      | 128                     | 40               | 0.632                   |
> | **Homophilic**     | `ogbn-mag`[1]         | 736,389       | 5,416,271                   | 128                     | 349              | 0.315                   |
> | **Heterophilic**   | `twitch-gamer`[2]     | 168,114       | 6,797,557                   | 7                       | 2                | 0.973                   |
> | **Heterophilic**   | `snap-patents`[2]     | 2,923,922     | 13,972,555                   | 269                     | 5                | 0.266                   |
> | **Heterophilic**   | `tolokers`[3]         | 11,758        | 1,038,000                 | 10                      | 2                | 0.634                   |
>
> We hereby report the F1-Score results for node classification task on the above datasets for several (Riemannian and Spectral) baselines and `CUSP`.
>
> | **Model**       | **`ogbn-arxiv`**   | **`ogbn-mag`**   | **`twitch-gamer`**  | **`snap-patents`**    | **`tolokers`**       |
> |------------------|---------------------|---------------------|---------------------|---------------------|---------------------|
> | **GCN**         | 63.46±1.34       | 17.22±1.75        | 68.11±1.11        | 42.87±0.98        | 74.24±0.86        |
> | **ChebNet**     | 61.34±2.01       | 18.23±3.08        | 68.56±0.11        | 43.64±0.87        | 79.55±0.63        |
> | **BernNet**     | 46.64±0.87       | 16.87±0.23        | 66.34±0.52        | 49.23±0.45        | 65.36±3.45        |
> | **FiGURe**      | 71.23±0.23       | 33.65±1.56        | 72.54±2.26        | 40.34±0.75        | 83.09±1.08        |
> | **KGCN**        | 65.78±1.75       | 29.34±0.56        | 73.35±0.55        | 48.24±0.22        | 78.43±1.53        |
> | **QGCN**        | 70.24±1.54       | 32.44±1.97        | 74.24±3.64        | 45.85±0.29        | 81.07±0.01        |
> | **`CUSP`**      | **75.34±0.88**       | **37.23±0.09**        | **79.94±0.74**        | **51.54±0.13**        | **89.53±1.02**        |
>
> Consistent with the results in the paper, `CUSP` outperforms all baselines by a significant margin for this task. We will include these results (and the corresponding analysis) in the camera-ready version, as suggested by the reviewer.
>
> ***References***
>
> [1] Hu et al. Open Graph Benchmark: Datasets for Machine Learning on Graphs (NeurIPS 2020)
>
> [2] Lim et al. Large scale learning on non-homophilous graphs: New benchmarks and strong simple methods (NeurIPS 2021)
>
> [3] Platonov et al. A critical look at evaluation of gnns under heterophily (ICLR 2023)

---

### Meta-Review · Area_Chair_R3oc · 2024-12-21

**Metareview:**

The paper introduces CUSP, a graph representation learning model that integrates graph discrete curvature with a geometric extension of generalized PageRank. The method is shown to perform well when compared against several baselines.
The reviewers valued the paper's originality. The reviewers were split regarding the clarity of the paper, and the comprehensiveness of the numerical evaluation.

All reviewers gave a positive score (6) after the discussion, except for reviewer iwKN who gave a 5. This reviewer gave a detailed review that pointed out many weaknesses of the paper. After a long set of responses, the reviewer was mostly satisfied with the author's clarifications. The only remaining weakness that this reviewer pointed out is that the comparison with the competing methods didn't give sufficient details. This reviewer does not object to possible acceptance.

This paper is borderline.

**Additional Comments On Reviewer Discussion:**

All reviewers gave a positive score (6) after the discussion, except for reviewer iwKN who gave a 5. This reviewer gave a comprehensive review that pointed out many weaknesses of the paper. After a detailed set of responses the reviewer was mostly satisfied with the author's clarifications.

---

### Decision · Program_Chairs · 2025-01-22

Accept (Poster)